# Order and stochasticity in the folding of individual *Drosophila* genomes

Sergey V. Ulianov[1,2,16], Vlada V. Zakharova[1,2,3,16], Aleksandra A. Galitsyna [4,16], Pavel I. Kos[5,16],
Kirill E. Polovnikov[4,6], Ilya M. Flyamer [7], Elena A. Mikhaleva[8], Ekaterina E. Khrameeva [4], Diego Germini[3],
Mariya D. Logacheva[4], Alexey A. Gavrilov[1,9], Alexander S. Gorsky[10,11], Sergey K. Nechaev[12,13],
Mikhail S. Gelfand[4,10], Yegor S. Vassetzky[3,14], Alexander V. Chertovich[5,15], Yuri Y. Shevelyov [8] &
Sergey V. Razin [1,2 ✉]

Mammalian and *Drosophila* genomes are partitioned into topologically associating domains (TADs). Although this partitioning has been reported to be functionally relevant, it is unclear whether TADs represent true physical units located at the same genomic positions in each cell nucleus or emerge as an average of numerous alternative chromatin folding patterns in a cell population. Here, we use a single-nucleus Hi-C technique to construct high-resolution Hi-C maps in individual *Drosophila* genomes. These maps demonstrate chromatin compartmentalization at the megabase scale and partitioning of the genome into non-hierarchical TADs at the scale of 100 kb, which closely resembles the TAD profile in the bulk in situ Hi-C data. Over 40% of TAD boundaries are conserved between individual nuclei and possess a high level of active epigenetic marks. Polymer simulations demonstrate that chromatin folding is best described by the random walk model within TADs and is most suitably approximated by a crumpled globule build of Gaussian blobs at longer distances. We observe prominent cell-to-cell variability in the long-range contacts between either active genome loci or between Polycomb-bound regions, suggesting an important contribution of stochastic processes to the formation of the *Drosophila* 3D genome.

[1] Institute of Gene Biology, Russian Academy of Sciences, Moscow, Russia. [2] Faculty of Biology, M.V. Lomonosov Moscow State University, Moscow, Russia. [3] UMR9018, CNRS, Université Paris-Sud Paris-Saclay, Institut Gustave Roussy, Villejuif, France. [4] Skolkovo Institute of Science and Technology, Moscow, Russia. [5] Faculty of Physics, M.V. Lomonosov Moscow State University, Moscow, Russia. [6] Institute for Medical Engineering and Science, Massachusetts Institute of Technology, Cambridge, MA 02139, USA. [7] MRC Human Genetics Unit, Institute of Genetics and Molecular Medicine, University of Edinburgh, Edinburgh, UK. [8] Institute of Molecular Genetics, National Research Centre "Kurchatov Institute", Moscow, Russia. [9] Center for Precision Genome Editing and Genetic Technologies for Biomedicine, Institute of Gene Biology, Russian Academy of Sciences, Moscow, Russia. [10] Institute for Information Transmission Problems (the Kharkevich Institute), Russian Academy of Sciences, Moscow, Russia. [11] Moscow Institute for Physics and Technology, Dolgoprudnyi, Russia. [12] Interdisciplinary Scientific Center Poncelet (CNRS UMI 2615), Moscow, Russia. [13] P.N. Lebedev Physical Institute, Russian Academy of Sciences, Moscow, Russia. [14] Koltzov Institute of Developmental Biology, Russian Academy of Sciences, Moscow, Russia. [15] Semenov Federal Research Center for Chemical Physics, Moscow, Russia. [16]These authors contributed equally: Sergey V. Ulianov, Vlada V. Zakharova, Aleksandra A. Galitsyna, Pavel I. Kos. ✉email: sergey.v.razin@usa.net

The principles of higher-order chromatin folding in the eukaryotic cell nucleus have been disclosed thanks to the development of chromosome conformation capture techniques, or C-methods[1,2]. High-throughput chromosome conformation capture (Hi-C) studies demonstrated that chromosomal territories were partitioned into partially insulated topologically associating domains (TADs)[3–5]. TADs likely coincide with functional domains of the genome[6–8], although the results concerning the role of TADs in the transcriptional control are still conflicting[6,9–12]. Analysis performed at low resolution suggested that active and repressed TADs were spatially segregated within A and B chromatin compartments[13,14]. However, high-resolution studies demonstrated that the genome was partitioned into relatively small compartmental domains bearing distinct chromatin marks and comparable in sizes with TADs[15]. In mammals, the formation of TADs by active DNA loop extrusion partially overrides the profile of compartmental domains[15,16]. Of note, TADs identified in studies of cell populations are highly hierarchical (i.e., comprising smaller subdomains, some of which are represented by DNA loops[5,17]).

Partitioning of the genome into TADs is relatively stable across cell types of the same species[3,4]. The recent data suggest that mammalian TADs are formed by active DNA loop extrusion[18,19]. The boundaries of mammalian TADs frequently contain convergent binding sites for the insulator protein CTCF that are thought to block the progression of loop extrusion[19–21]. Contribution of DNA loop extrusion in the assembly of *Drosophila* TADs has not been demonstrated yet[22]; thus, *Drosophila* TADs might represent pure compartmental domains[23]. Large TADs in the *Drosophila* genome are mostly inactive and are separated by transcribed regions characterized by the presence of a set of active histone marks, including hyperacetylated histones[5,24]. Some insulator/architectural proteins are also overrepresented in *Drosophila* TAD boundaries[24–26], but their contribution to the formation of these boundaries has not been directly tested. The results of computer simulations suggest that *Drosophila* TADs are assembled by the condensation of nucleosomes of inactive chromatin[24].

The current view of genome folding is based on the population Hi-C data that present integrated interaction maps of millions of individual cells. It is not clear, however, whether and to what extent the 3D genome organization in individual cells differs from this population average. Even the existence of TADs in individual cells may be questioned. Indeed, the DNA loop extrusion model considers TADs as a population average representing a superimposition of various extruded DNA loops in individual cells[18]. Heterogeneity in patterns of epigenetic modifications and transcriptomes in single cells of the same population was shown by different single-cell techniques, such as single-cell RNA-seq[27], ATAC-seq[28], and DNA-methylation analysis[29]. Studies performed using FISH demonstrated that the relative positions of individual genomic loci varied significantly in individual cells[30]. The first single-cell Hi-C study captured a low number of unique contacts per individual cell[31] and allowed only the demonstration of a significant variability of DNA path at the level of a chromosome territory. Improved single-cell Hi-C protocols[32,33] allowed to achieve single-cell Hi-C maps with a resolution of up to 40 kb per individual cell[32,34] and investigate local and global chromatin spatial variability in mammalian cells, driven by various factors, including cell cycle progression[33]. Of note, TAD profiles directly annotated in individual cells demonstrated prominent variability in individual mouse cells[32]. The possible contribution of stochastic fluctuations of captured contacts in sparse single-cell Hi-C matrices into this apparent variability was not analyzed[32]. More comprehensive observations were made when super-resolution microscopy (Hi-M, 3D-SIM) coupled with high-throughput hybridization was used to analyze chromatin folding in individual cells at a kilobase-scale resolution. These studies demonstrated chromosome partitioning into TADs in individual mammalian cells and confirmed a trend for colocalization of CTCF and cohesin at TAD boundaries, although the positions of boundaries again demonstrated significant cell-to-cell variability[35]. Condensed chromatin domains coinciding with population TADs were also observed in *Drosophila* cells[36,37]. In accordance with previous observations made in cell population Hi-C studies[24], the obtained results suggested that partitioning of the *Drosophila* genome into TADs was driven by the stochastic contacts of chromosome regions with similar epigenetic states at different folding levels[38].

Although studies performed using FISH and multiplex hybridization allowed to construct chromatin interaction maps with a very high resolution[35], they cannot provide genome-wide information. Here, we present single-nucleus Hi-C (snHi-C) maps of individual *Drosophila* cells with a 10-kb resolution. These maps allow direct annotation of TADs that appear to be non-hierarchical and are remarkably reproducible between individual cells. TAD boundaries conserved in different cells of the population bear a high level of active chromatin marks supporting the idea that active chromatin might be among determinants of TAD boundaries in *Drosophila*[24].

## Results

**High-resolution single-nucleus Hi-C reveals distinct TADs in *Drosophila* genome.** To investigate the nature of TADs in single cells and to characterize individual cell variability in *Drosophila* 3D genome organization, we performed single-nucleus Hi-C (snHi-C)[32] (Fig. 1a) in 88 asynchronously growing *Drosophila* male Dm-BG3c2 (BG3) cells (Supplementary Fig. 1a) in parallel with the bulk BG3 in situ Hi-C analysis and obtained 2–5 million paired-end reads per single-cell library (for the data processing workflow, see Supplementary Fig. 1b). To select the libraries for deep sequencing, we subsampled the snHi-C data to estimate the expected number of unique contacts that could be extracted from the data (Supplementary Fig. 2a; also see "Methods"). Twenty libraries were additionally sequenced with 16.7–36.5 million paired-end reads, and we extracted 8032–107,823 unique contacts per cell (Supplementary Table 1). We developed a custom *pairtools*-based approach termed ORBITA (One Read-Based Interaction Annotation) (Fig. 1b) to eliminate artificial contacts generated by spontaneous template switches of the Phi29 DNA-polymerase[39,40] (Fig. 1c, d) during the whole-genome amplification (WGA) step (see "Methods"). In contrast to the *hiclib*[32,41] (see "Methods") annotations showing up to 20 contacts per restriction fragment (RF) in a single nucleus, ORBITA detects one or two unique contacts per RF (Fig. 1d, Supplementary Fig. 2b, c). We tested ORBITA by analyzing previously published snHi-C data from murine oocytes[32] and found that ORBITA allowed us to filter out artificial junctions in this dataset (Supplementary Fig. 3a). Notably, *hiclib* and ORBITA detect a similar number of contacts per RF in single-cell Hi-C data obtained without the usage of Phi29 DNA-polymerase[33] (Supplementary Fig. 3b). Thus, ORBITA efficiently filters out artificial Phi29 DNA-polymerase-produced DNA chimeras from snHi-C libraries.

We then constructed snHi-C maps with a resolution of up to 10 kb (Fig. 1e). In single nuclei, the dependence of the contact probability on the genomic distance, $P_c(s)$, has a shape comparable to that observed in the bulk BG3 in situ Hi-C regardless of the number of captured contacts (Fig. 1f), indicating that the key steps of the snHi-C protocol such as fixation, DNA fragmentation, and in situ ligation were performed successfully. To estimate the overall quality of the snHi-C libraries, we first

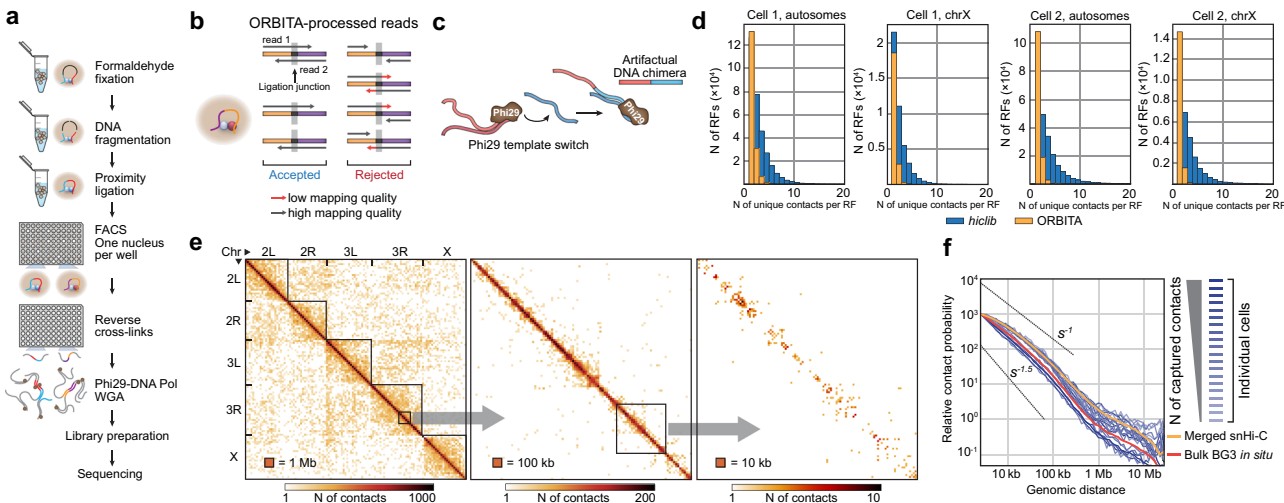

**Fig. 1 ORBITA-processed Hi-C data from single *Drosophila* nuclei. a** Single-nucleus Hi-C protocol scheme (see "Methods" for details). **b** Workflow of ORBITA function for detection of unique Hi-C contacts. ORBITA processes only chimeric reads with good mapping quality containing ligation junction marked by the cleavage site for restriction enzyme used for the snHi-C map construction. **c** Scheme of an artefactual DNA chimera formation by Phi29-DNA-polymerase. **d** Number of unique contacts per restriction fragment (RF) captured by ORBITA (orange) and *hiclib* (blue) for autosomes and the X chromosome. BG3 is a diploid male cell line; accordingly, in a single nucleus, each RF from autosomes and the X chromosome could establish no more than four and two unique contacts, respectively. Cell 1, autosomes: $n = 148{,}415$ and $159{,}060$ for ORBITA and *hiclib*, respectively; ChrX: $n = 22{,}016$ and $26{,}674$ for ORBITA and *hiclib*, respectively. Cell 2: autosomes: $n = 113{,}988$ and $119{,}066$ for ORBITA and *hiclib*, respectively; ChrX: $n = 16{,}384$ and $19{,}429$ for ORBITA and *hiclib*, respectively. **e** Visualization of a single-nucleus Hi-C map at 1-Mb, 100-kb, and 10-kb resolution for the cell with 107,823 captured unique contacts. **f** Dependence of the contact probability $P_c(s)$ on the genomic distance s for single nuclei (shades of blue reflect the number of unique contacts captured in individual nuclei), merged snHi-C data (orange), and bulk in situ BG3 Hi-C data (red). Black lines show slopes for $P_c(s) = s^{-1.5}$ and $P_c(s) = s^{-1}$.

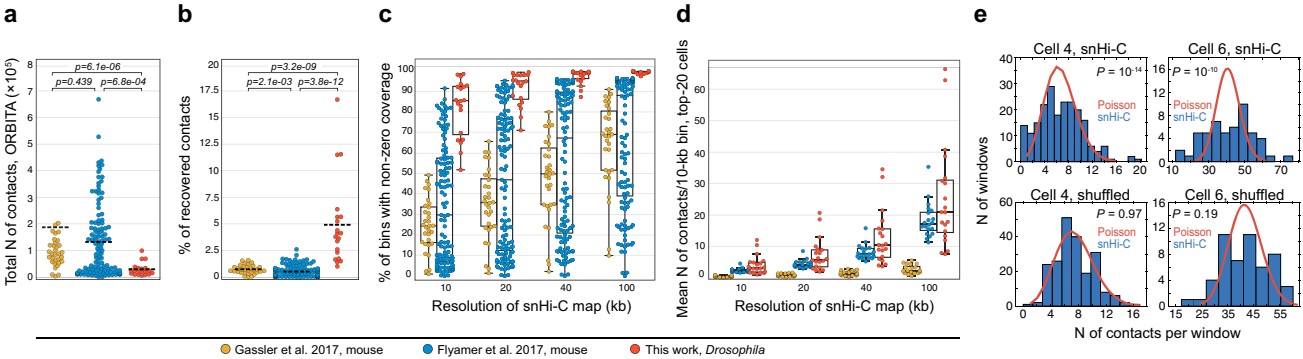

**Fig. 2 snHi-C datasets in *Drosophila* represent a major portion of the genome and are not random matrices. a** Number of ORBITA-captured contacts per individual nuclei obtained for *Drosophila* in the current work, compared with mouse oocytes from Flyamer et al.[32] and G2 zygotes pronuclei from Gassler et al.[34]. **$p < 0.01$ using the Mann–Whitney two-sided test. $n = 20$, 120, and 32 nuclei for *Drosophila* in the current work, mouse oocytes from Flyamer et al.[32] and G2 zygotes pronuclei from Gassler et al.[34], respectively (the same is true for (**b**) and (**c**)). **b** Percentage of recovered contacts out of the total possible for *Drosophila* in the current work, compared with mouse oocytes from Flyamer et al.[32] and G2 zygotes pronuclei from Gassler et al.[34]. P-values are calculated using the Mann–Whitney two-sided test. **c** Percentage of bins with non-zero coverage for autosomes and sex chromosome of *Drosophila*, murine oocytes, and G2 zygote pronuclei. Boxplots represent the median, interquartile range, maximum and minimum. **d** Mean number of contacts per 10-kb genomic bin in top-20 cells in the current work, compared with mouse oocytes from Flyamer et al.[32] and G2 zygotes pronuclei from Gassler et al.[34]. Boxplots represent the median, interquartile range, maximum and minimum. **e** Distributions of the number of contacts in windows of fixed size (100 kb for the Cell 4, and 400 kb for the Cell 6; chr2R) in snHi-C data and shuffled maps for two individual cells (blue bars). The red curve shows the Poisson distribution expected for an entirely random matrix with the same number of contacts. P-values were estimated by the goodness of fit test. $n = 211$ and 52 windows for the cell 4 and for the cell 6, respectively.

calculated the number of captured contacts per cell. On average, we extracted 33,291 unique contacts from individual nuclei that represented 5% of the theoretical maximum number of contacts and corresponded to four contacts per 10-kb genomic bin (see "Methods"); in the best cell, 17% of contacts were recovered (Fig. 2a, b, Supplementary Table 1). Relying on the number of captured contacts, we then estimated the proportion of the genome available for the downstream analysis. At 10-kb

resolution, ~82% of the genome on average was covered with contacts in each individual cell, and 67% of genomic bins established more than 1 contact (Fig. 2c). Notably, in the previously published mouse snHi-C datasets, ~0.6% of theoretically possible contacts were detected on average (Fig. 2b). Because the top-20 mouse snHi-C libraries from Flyamer et al.[32] demonstrated a comparable genome coverage with contacts and a number of contacts per 10-kb genomic bin (Fig. 2d), we could

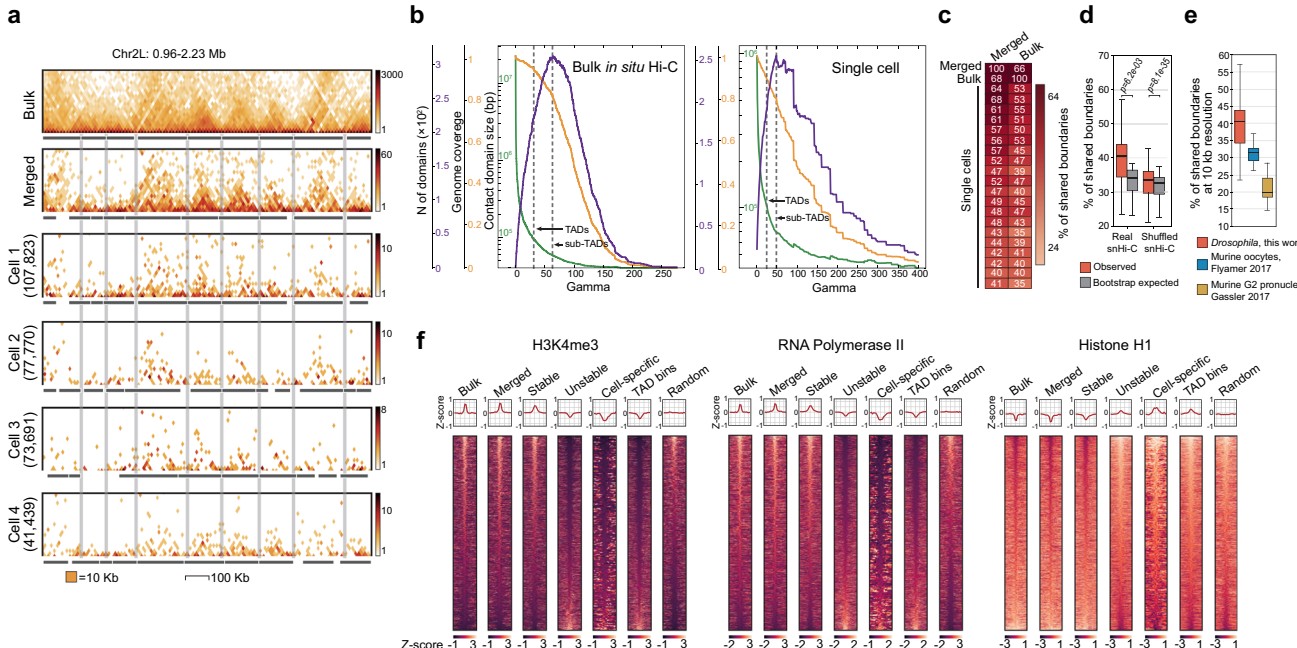

**Fig. 3 Stable TAD boundaries are defined by high level of active epigenetic marks. a** Example of a genomic region on Chromosome 2L with a high similarity of TAD profiles (black rectangles) in individual cells and bulk BG3 in situ Hi-C data. Number of unique captured contacts is shown in brackets. Positions of TAD boundaries identified in bulk BG3 in situ Hi-C data (top panel) are highlighted with gray lines. Here and below, TADs are identified using lavaburst software. **b** Dependence of the contact domain (CD) size (green), genome coverage by CDs (orange), and number of identified CDs (violet) on the $\gamma$ value in bulk (left) and single-cell (right) BG3 Hi-C data. $\gamma$ values selected for the calling of sub-TADs ($\gamma_{max}$) and TADs ($\gamma_{max}/2$) are marked with vertical gray lines. **c** Percentage of TAD boundaries shared between single cells, bulk BG3 in situ Hi-C, and merged snHi-C data. **d** Percentage of shared boundaries in real snHi-C, shuffled control maps, and bootstrap expected. Boxplots represent the median, interquartile range, maximum and minimum. \*\*$p$ < 0.01 using the Mann–Whitney two-sided test. $n = 380$ comparisons between individual cells. **e** Percentage of shared boundaries in real snHi-C for *Drosophila*, murine oocytes from Flyamer et al.[32] and G2 zygote pronuclei from Gassler et al.[34]. Boxplots represent the median, interquartile range, maximum and minimum. $n = 380$ comparisons between individual cells. **f** Heatmaps of active (H3K4me3, RNA Polymerase II) and inactive (H1 histone) chromatin marks centered at single-cell TAD boundaries from different groups (±100 kb). Bulk—conventional BG3 in situ Hi-C; merged—aggregated snHi-C data from all individual cells; stable and unstable—boundaries found in more and in less than 50% of cells, respectively; cell-specific—boundaries identified in any one individual cell; TAD bins—genomic bins from TAD interior; random—randomly selected genomic bins.

directly compare the *Drosophila* and mouse snHi-C maps (see below). Next, to verify that these sparse snHi-C matrices were not generated by random fluctuations of captured contacts, we calculated the distributions of the contact numbers in sliding non-intersecting windows of different fixed sizes. In contrast to the shuffled maps, these distributions in the original data are distinct from the Poisson shape typical for random matrices (Fig. 2e, see "Methods" and Supplementary Fig. 4). We conclude that the snHi-C maps obtained here are of acceptable quality and indeed reflect specific patterns of spatial contacts in chromatin.

Visual inspection of snHi-C maps revealed distinct 50–200 kb contact domains that closely recapitulated the TAD profile in the bulk BG3 in situ Hi-C data (Fig. 3a). To call TADs in snHi-C data systematically, we used the lavaburst Python package with the modularity scoring function[32]. For each nucleus, we performed TAD segmentation in snHi-C maps of 10-kb resolution at a broad range of the gamma ($\gamma$) master parameter values (Fig. 3b, see "Methods" and Supplementary Fig. 5). Of note, the majority of the identified boundaries were resistant to the data downsampling, indicating that these boundaries did not result from fluctuations of captured contacts in sparse snHi-C matrices (Supplementary Fig. 6). In individual nuclei, we identified 554–1402 TADs with a median size of 60 kb covering 40–76% of the genome at the $\gamma$ value corresponding to the maximal number of TADs called ($\gamma_{max}$). At 10–20 kb resolution, the median size of *Drosophila* TADs was previously estimated as 100–150 kb[5,24,25]. To obtain a robust TAD profile, we used $\gamma_{max}/2$

corresponding to TADs with a median size equal to that for TADs identified in the *Drosophila* cell population according to the previously published data[24]. At $\gamma_{max}/2$, we identified 510–1175 TADs with a median size ~90 kb covering up to 89% of the genome in best snHi-C matrices (Supplementary Fig. 5).

To additionally validate the single-cell TAD segmentations, we utilized a modification of the recently published[42] spectral clustering method based on the non-backtracking random walks (NBT; see "Methods"). The non-backtracking operator is used to resolve communities in sufficiently sparse networks[42,43], thus providing a useful tool for TAD annotation in single-cell Hi-C matrices. The method performs dimensionality reduction of the network using the leading eigenvectors of the non-backtracking operator, which has a distinctive disc-shape complex spectrum with a number of isolated eigenvalues on the real axis (Supplementary Fig. 7d). The resulting average size of the detected TADs was 110 kb, closely matching the typical TAD size in the population-averaged data and in the single-cell modularity-derived segmentations. The mean number of detected TADs per cell (855 and 920 for the NBT and modularity, respectively) and single-cell TAD segmentations were remarkably similar between the two methods (Supplementary Fig. 7a) and demonstrated the same epigenetic properties (Supplementary Fig. 7c, see below). Moreover, the modularity-derived TAD boundaries were robust to the data resolution changes. On average, 84.8% of modularity-derived boundaries at the 20-kb resolution and 78.6% of boundaries at the 40-kb resolution have a matching boundary

at the 10-kb resolution. This is significantly higher than the 43 and 58% expected at random, respectively. Taken together, these results indicate that TAD profiles are robust and, thus, acceptable for the downstream analysis.

**TADs are largely conserved in individual *Drosophila* nuclei, and stable TAD boundaries are enriched with active chromatin**. We found that TADs tended to occupy similar positions in different cells regardless of the number of captured contacts (Fig. 3a, Supplementary Fig. 8). On average, 46.6% of population-identified TAD boundaries were present in each of the single cells analyzed (Fig. 3c), and 39.5% of boundaries were shared between different cells in pairwise comparisons (Supplementary Fig. 8). This is significantly higher than the percentage of shared boundaries for shuffled control maps (32.9%) and the percentage expected at random (33.1%, Fig. 3d). Notably, 44% of NBT-identified single-cell TAD boundaries were conserved in pairwise cell-to-cell comparisons (Supplementary Fig. 7b), supporting the results obtained in the analysis of modularity-derived TAD boundary profiles. In individual mammalian cells, TADs frequently overpassed the boundaries identified in the cell population, arguing for a substantial degree of stochasticity in genome folding[32,35,44]. We used the ORBITA algorithm to reanalyze previously published snHi-C data from murine oocytes[32] and G2 zygote pronuclei[34] and found that 31.2 and 21% of boundaries were shared on average between any two cells, respectively (Fig. 3e, Supplementary Fig. 9). This result is reproduced at 40-kb resolution and persists for a broad range of snHi-C datasets' quality (Supplementary Fig. 10). We conclude that, in *Drosophila*, TADs have more stable boundaries as compared to mammals. This corroborates recent observations of the Cavalli lab[37] and may reflect the differential impact of loop extrusion[18,19,34] and internucleosomal contacts[24] on TAD formation[16,23].

Population TADs in *Drosophila* identified at 10–20 kb resolution mostly correspond to inactive chromatin, whereas their boundaries and inter-TAD regions correlate with highly acetylated active chromatin[24,45]. These are further partitioned into much smaller domains with the size of about 9 kb[25] and, thus, unavailable for the analysis at the resolution of our Hi-C maps. To examine the properties of TAD boundaries at the single-cell level, we divided all TAD boundaries from snHi-C data into three groups according to the proportion of cells where these boundaries were present and analyzed them separately (number of boundaries of each type and distances between neighboring boundaries within each type are shown in Supplementary Fig. 13). The boundaries present in a large fraction of cells (more than 50% of cells) defined here as "stable" overlapped 73% of conserved boundaries between BG3 and Kc167 cell lines[46] and had high levels of active chromatin marks (RNA polymerase II, H3K4me3; Fig. 3f, Supplementary Figs. 11, 12). They were also slightly enriched in some architectural proteins associated with active promoters (BEAF-32, Chriz, CTCF, and GAF; Supplementary Fig. 11, 12). In contrast, boundaries identified in less than 50% of cells and defined here as "unstable" (as well as boundaries identified in just one cell termed cell-specific boundaries) were remarkably depleted of acetylated histones and features of transcriptionally active chromatin while being enriched in histone H1 and other proteins of repressed chromatin similarly to the internal TAD bins (Fig. 3f, Supplementary Fig. 11, 12). The epigenetic profiles of "unstable" boundaries may be due to the fact that actual profiles of active chromatin in individual cells differ from the bulk epigenetic profiles used in our analysis. However, it may also reflect a certain degree of stochasticity in chromatin fiber folding into contact domains[35]. Taking into consideration the fact that active chromatin regions mostly colocalize with

stable boundaries, one would expect the "unstable" boundaries tend to be located in the inactive parts of the chromosome.

**TADs in individual *Drosophila* cells are not hierarchical**. *Drosophila* TADs are hierarchical in cell population-based Hi-C maps[45,47]. It is, however, not clear whether the hierarchy exists in individual cells or emerges in the bulk BG3 in situ Hi-C maps as a result of averaging of alternative chromatin configurations over a number of individual cells. To test this proposal, we focused on two TAD segmentations: at $\gamma_{max}/2$ (TADs) and $\gamma_{max}$ (smaller domains referred to as sub-TADs located inside TADs, Fig. 4a). We analyzed only the haploid X chromosome to avoid combined folding patterns of diploid somatic chromosomes. We assumed that if TADs in individual nuclei are truly hierarchical, then sub-TADs belonging to the same TAD should be demarcated with well-defined boundaries arising from specific folding of the chromatin. To determine whether this is the case, we tested the resistance of sub-TAD boundaries to the data downsampling (two-fold depletion of total number of contacts in the snHi-C maps). In contrast to relatively stable TAD boundaries, sub-TAD boundaries showed a two-fold reduction in the probability of detection in downsampled datasets (Fig. 4b). Moreover, we found that profiles of sub-TADs were highly different in individual nuclei: only approximately 20% of sub-TAD boundaries in individual cells were shared in pairwise comparisons, similar to the shuffled controls (Supplementary Fig. 14). Hence, a potential hierarchy of TAD structure in single cells appears to reflect local Hi-C signal fluctuations. The hierarchical structure of TADs observed in bulk *Drosophila* Hi-C data[45,48], thus, likely results from the superposition of multiple alternative chromatin folding patterns present in individual nuclei; this is also supported by the visual inspection of snHi-C maps (Fig. 4c).

**A-compartment in individual *Drosophila* nuclei**. In animal cells, TADs of the same epigenetic type interact with each other across large genomic distances, forming compartments that spatially segregate active and inactive genomic loci in the nuclear space[13]. Similarly to *Drosophila* embryo[5], S2[49], and Kc167 cells[50], we observed an increased long-range interaction frequency within the A-compartment in the bulk BG3 in situ Hi-C data (Fig. 4d–f; Supplementary Fig. 15). Supporting this observation, we also found increased long-range interactions between genomic regions of the X chromosome activated by male-specific-lethal (MSL) complex binding[51] (Fig. 4h) in both BG3 in situ Hi-C data and the merged cell. In contrast, we observed a weak enrichment of long-range interactions between Polycomb-repressed regions[52,53] bound by dRING (Fig. 4i)[54] and nearly no enrichment for B-compartment regions (Fig. 4d, e, g).

We could not directly detect compartments in individual nuclei due to the sparsity of the maps, but we observed a substantial enrichment of contacts in the A-compartment after averaging contacts in each individual nucleus across the population-based compartment mask (Fig. 4d, Supplementary Fig. 15). Compartmentalization might, thus, be a genuine feature of chromatin folding of *Drosophila* individual nuclei. The presence of extensive long-range contacts between the active genome regions in individual chromosomes is also supported by the contact probability $P_c(s)$ plotted for active and inactive genomic bins separately: $P_c(s)$ between active genome regions has a gentler slope outside TADs, indicating that active, but not inactive chromatin forms spatial contacts across large genomic distances (Fig. 4e). These results suggest that active and inactive genome loci are spatially segregated in individual *Drosophila* nuclei; active regions establish long-distance contacts, possibly at transcription factories and nuclear speckles[55–58].

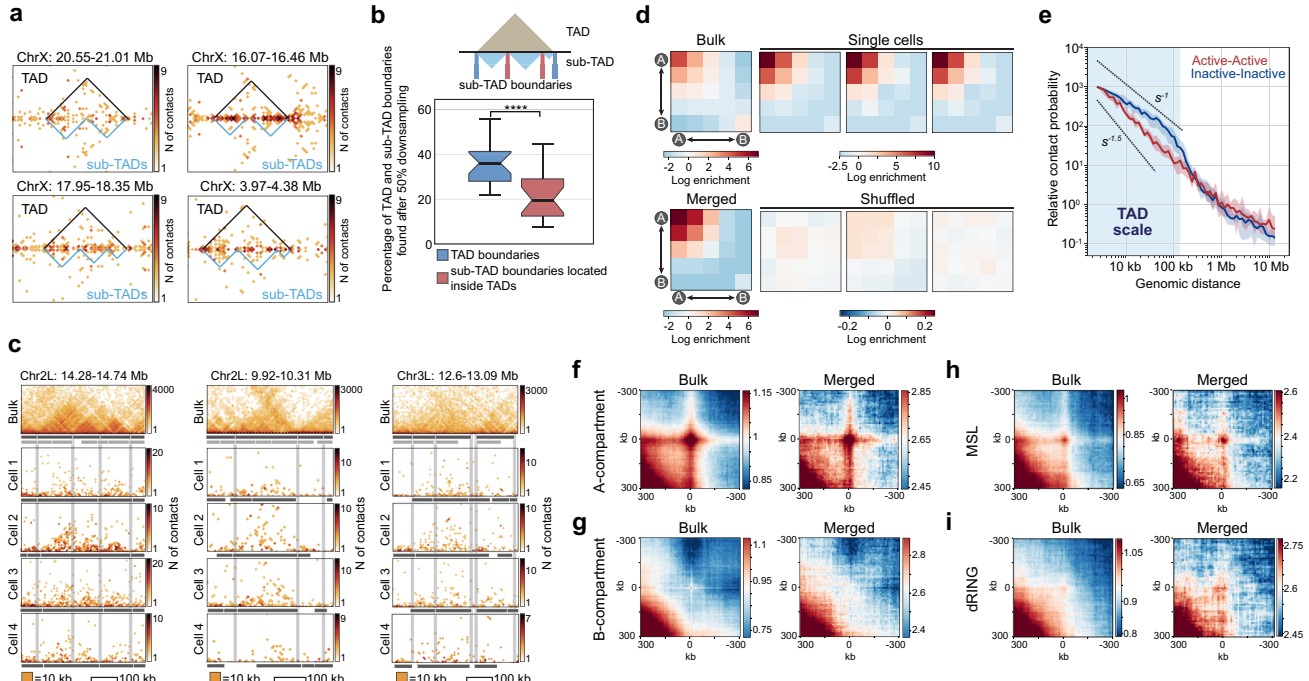

**Fig. 4 Chromatin in individual *Drosophila* cells is compartmentalized and lacks folding hierarchy at the level of TADs. a** Examples of TAD (black triangles) and sub-TAD (light blue triangles) positions in the haploid X chromosome in individual nuclei with 77,770 unique contacts. **b** Percentage of TAD and sub-TAD boundaries per cell (excluding TAD boundaries for the same cells) found as sub-TAD boundaries in the snHi-C maps after 50% downsampling. Downsampling was performed 10 times. At the top: TAD boundaries are highlighted with blue lines, sub-TAD boundaries located inside TADs are highlighted with red lines. Boxplots represent the median, interquartile range, maximum and minimum. ****$p$-value < 0.0001 using the Mann–Whitney two-sided test. $n = 20$ cells. **c** Genomic regions with alternative chromatin folding patterns in individual cells. Positions of sub-TADs and TADs identified in bulk BG3 in situ Hi-C data (top panels) are highlighted with light gray and dark gray rectangles, respectively. Positions of TAD boundaries in bulk BG3 in situ Hi-C data are shown with vertical light gray lines. **d** Heatmaps showing log₂ values of contact enrichment between genomic regions belonging to putative A- (negative PC1 values) and B- (positive PC1 values) compartments (saddle plot). PC1 profile is constructed using the bulk BG3 in situ Hi-C data. **e** Contact probability $P_c(s)$ between transcriptionally active (red) and inactive (blue) genomic bins in the snHi-C data. The light blue shading shows the genomic distances corresponding to the average TAD size in single nuclei. **f** Average plot of long-range interactions between top 1000 regions of A compartment annotated by bulk Hi-C data (in bulk Hi-C and merged snHi-C). **g** Average plot of long-range interactions between top 1000 regions of B compartment annotated by bulk Hi-C data (in bulk Hi-C and merged snHi-C). **h** Average plot of interactions between top 500 regions enriched in MSL (in bulk Hi-C and merged snHi-C) on chromosome X. **i** Average plot of interactions between top 500 regions enriched in dRING (in bulk Hi-C and merged snHi-C).

**Modeling of DNA fiber folding within X-chromosome by constrained polymer collapse.** We next applied dissipative particle dynamics (DPD) polymer simulations[59] to reconstruct the 3D structures of haploid X chromosomes (Supplementary Fig. 16a) in individual cells using the snHi-C data (Fig. 5a, Supplementary Fig. 16b). The chromatin fiber path in these structures is strictly determined by the pattern of contacts derived from the snHi-C experiments and, thus, reflects the actual folding of the X chromosome in living cells[60]. As revealed by TAD annotation, the DPD simulations successfully reproduced chromatin fiber folding even at short and middle genomic distances because TAD positions along the X chromosome were largely preserved between the models and the original snHi-C data (Fig. 5a, Supplementary Figs. 17, 19a, b; also see "Methods"). Moreover, the simulations correctly reproduced chromatin folding at the scale of the whole chromosome with a well-defined A-compartment (Fig. 5a, Supplementary Fig. 18). Additionally, to validate the simulation results using an alternative approach, we performed multicolor in situ fluorescence hybridization (FISH) with two intra-TAD probes and one probe located outside the selected TAD. The distributions of inter-probe spatial distances extracted from the X chromosome model closely resemble those of the FISH analysis (Supplementary Fig. 19c). Taken together, these observations confirm the validity of our approach.

The snHi-C maps show remarkable cell-to-cell variability in the distribution of captured contacts (Figs. 3a, 4c); therefore, we performed a pairwise comparison of 3D models of the X chromosome in individual cells using the coefficient of the difference at a broad range of genomic distances (Fig. 5b; see "Methods"). The higher the value of the coefficient, the higher the difference between the distance matrices obtained from the models. We have found that chromatin fiber conformation was strikingly different between individual models (red curve, Fig. 5b) in comparison to different configurations (at different time points) of each particular model (blue curve, Fig. 5b), showing the prominent cell specificity in the organization of the X chromosome territory (CT). Notably, shuffling of contacts (see "Methods") in the snHi-C data prior to simulations significantly decreased the variability in the chromatin fiber conformation at long distances (gray curve, Fig. 5b). Despite cell-to-cell differences in the overall 3D shape of a particular TAD (Fig. 5c, Supplementary Fig. 19d), the variability of the chromatin fiber conformation was substantially lower at short ranges (within TADs) as compared to long-range distances (Fig. 5b). This difference could be due to an increased flexibility in chromatin folding arising from larger genomic distances. In addition, the curve of the coefficient of difference between individual models reached the plateau outside TADs (Fig. 5b), suggesting that the

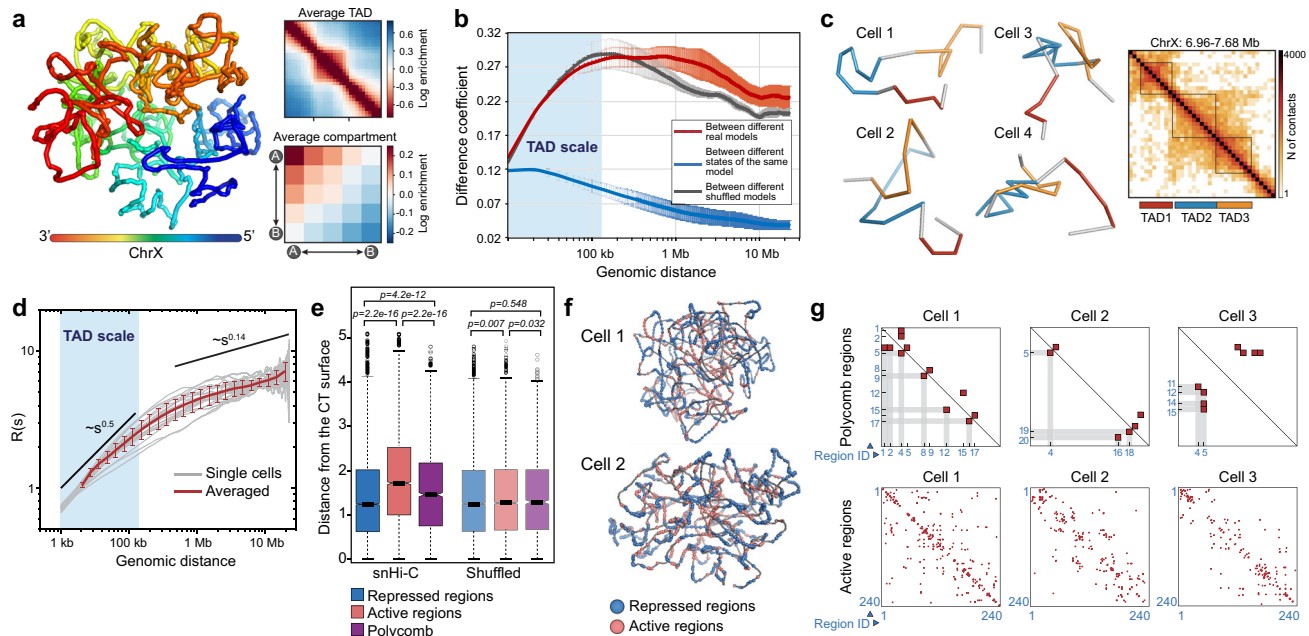

**Fig. 5 3D folding of the haploid X chromosome. a** Left panel: 3D structure of the haploid X chromosome from an individual nucleus derived from snHi-C data by the DPD polymer simulations. Right panel: averaging of contacts in the 3D model over TAD positions in the corresponding snHi-C data (top) and compartment (bottom) positions annotated in the bulk BG3 in situ Hi-C data. Source data are provided as a Source data file. **b** Coefficient of difference over a broad range of genomic distances. The central curves represent average values. Error bars show standard deviation (SD) for 20 independent model realizations, n = 2242 distinct ranges of genomic distances used for the curve construction. **c** Single-nucleus 3D structures of a genomic region covered by three TADs (left). Right, bulk BG3 in situ Hi-C map of this region. TAD positions are shown by colored rectangles below the map and by black squares on the map. **d** Dependence of the Euclidean spatial inter-particle distance R (see "Methods") on the genomic distance s between these particles along the chromatin fiber. Black lines show the slopes characteristic for the random walk behavior ($s^{0.5}$) and the crumpled globule build-up from Gaussian blobs ($s^{0.14}$). Error bars show standard deviation (SD) for 20 independent model realizations, n = 20. **e** Spatial distance from the surface of a chromosome territory (CT) to transcriptionally active (n = 8966) and inactive (n = 17,103; according to RNA-seq from ref. [24]) regions, and Polycomb-bound domains (n = 2160; according to the 9-state chromatin type annotation[54]). Boxplots show data aggregated from all individual models analyzed. Boxplots represent the median, interquartile range, maximum and minimum. P-values are calculated using the Mann–Whitney two-sided test. **f** Examples of simulated ChrX 3D structures demonstrating the preferential location of transcriptionally inactive regions (blue particles) at the surface of the CT. **g** Heatmap showing cell-to-cell variability in interactions detected between Polycomb domains (upper panels) or between transcriptionally active regions (bottom panels) in individual cells. Red rectangles denote detected interactions. The total number of Polycomb and active regions identified in the X chromosome are 20 and 240, respectively (see "Methods"). Only interacting domains are numbered for Polycomb domains; interacting active regions are not numbered due to their multiplicity.

variability of chromatin folding inside and outside TADs was governed by different rules. Due to the fact that TADs in *Drosophila* (at the 10–20 kb resolution of the Hi-C maps) are largely composed of inactive chromatin, we propose that the chromatin fiber conformation within TADs is mostly determined by interactions between adjacent non-acetylated nucleosomes. In contrast, at large genomic distances, TADs interact with each other in a stochastic manner, imposing the spherical form of the CT that is observed in all model structures (Fig. 5a, Supplementary Figs. 16, 20). In line with this hypothesis, the dependence of spatial distance R between any two particles on the genomic distance s revealed two distinct modes of polymer folding (Fig. 5d). At the scale of ~100 kb (e.g., inside TADs), the chromatin fiber demonstrated a random walk behavior ($s^{0.5}$) similar to the chromatin of budding yeast. At larger distances, R(s) had a scaling similar to a crumpled globule build of Gaussian blobs ($s^{0.14}$)[61]. Thus, chromatin folding within TADs and at the scale of the whole CT could be driven by different molecular mechanisms.

Analysis of the radial distributions of transcriptionally active, inactive, and Polycomb-bound genome regions in our models demonstrated that active chromatin tended to be located in the CT interior, whereas inactive regions were located near the CT surface (Fig. 5e, f); this can be driven by interactions with the nuclear lamina[62]. Formation of nuclear microcompartments such

as Polycomb bodies[63] represents another factor determining the large-scale spatial structure of the X chromosome territory. We analyzed patterns of interactions between individual Polycomb-occupied regions in the 3D models. To this aim, each of such regions was assigned a consecutive number according to their positions along the chromosome. The examples of 2D maps demonstrating regions residing in a spatial proximity in each cell are presented in Fig. 5g (upper panels). We found that Polycomb-occupied regions interacted with each other in a cell-specific manner and, moreover, such contacts occurred between loci regardless of the genomic distances between them (Fig. 5g, upper panels). Using a similar approach, we constructed 2D interaction maps of active genomic regions (Fig. 5g, bottom panels). Active genome regions also interacted with each other across large genomic distances in a cell-specific manner (Fig. 5g, bottom panels). We propose that these two types of long-range interactions: stochastic assembly of Polycomb bodies and transcription-related microcompartments (factories[64]), underlie the cell-specific conformation of the chromatin fiber within CTs in *Drosophila*.

## Discussion
Folding of interphase chromatin in eukaryotes is driven by multiple mechanisms operating at different genome scales and generating distinct types of the 3D genome features[16,20]. In

mammalian cells, cohesin-mediated chromatin fiber extrusion mainly impacts the genome topology at the scale of ~100–1000 kb by producing loops, resulting in the formation of TADs[18,19] and establishing enhancer-promoter communication[65]. Chromatin loop formation by the loop extrusion complex (LEC) in mammalian cells is a substantially deterministic process due to the preferential positioning of loop anchors encoded in DNA by CTCF binding sites (CBS). The cohesin-CTCF molecular tandem modulates folding of intrinsically disordered chromatin fiber[16,23]. On the other hand, association of active and repressed gene loci in chromatin compartments[13,14], and formation of Polycomb and transcription-related nuclear bodies[66,67] in both mammalian and *Drosophila* cells shape the 3D genome at the scale of the whole chromosome. These associations appear to be stochastic: a particular Polycomb-bound or transcriptionally active region in individual cells interacts with different partners located across a wide range of genomic distances[68].

Here, we applied the single-nucleus Hi-C to probe the 3D genome in individual *Drosophila* cells at a relatively high resolution that was not achieved previously in single-cell Hi-C studies. Based on our observations, we suggest that, in *Drosophila*, both deterministic and stochastic forces govern the chromatin spatial organization (Fig. 6a).

We found that the entire individual *Drosophila* genomes were partitioned into TADs; this observation supports the results of recent super-resolution microscopy studies[37]. TAD profiles are highly similar between individual *Drosophila* cells and demonstrate lower cell-to-cell variability as compared to mammalian TADs. According to our model[24], large inactive TADs in *Drosophila* are assembled by multiple transient electrostatic interactions between non-acetylated nucleosomes in transcriptionally silent genome regions. Conversely, TAD boundaries and inter-TAD regions at the 10-kb resolution of Hi-C maps in *Drosophila* were found to be formed by transcriptionally active chromatin. This result may explain why TADs in individual cells occupy virtually the same genomic positions (Fig. 6b). Gene expression profile is a characteristic feature of a particular cell type, and, thus, should be relatively stable in individual cells within the population. In agreement with this, we demonstrated that invariant TAD boundaries present in a major portion of individual cells were highly enriched in active chromatin marks. Moreover, stable boundaries were also largely conserved in other cell types (see "Results" and ref. [46]), possibly due to the fact that TAD boundaries were frequently formed at the position of housekeeping genes.

In contrast to stable TAD boundaries, the boundaries that demonstrate cell-to-cell variability bear silent chromatin. Some cell-specific TAD boundaries may originate at various positions due to a putative size limit of large inactive TADs or other restrictions in chromatin fiber folding. Indeed, it appears that the assembly of randomly distributed TAD-sized self-interacting domains is an intrinsic property of chromatin fiber folding[35]. In mammals, the positioning of these domains is modulated by cohesin-mediated DNA loop extrusion[35], whereas in *Drosophila*, it may be modulated by segregation of chromatin domains bearing distinct epigenetic marks[16,23]. Even if cell-specific and unstable TAD boundaries are distributed in a random fashion, they should be depleted in active chromatin marks because active chromatin regions are mainly occupied by stable TAD boundaries. We also cannot exclude that variable boundaries and the TAD boundary shifts are caused by local variations in gene expression and active chromatin profiles in individual cells that we cannot assess simultaneously with constructing snHi-C maps.

Our results are also compatible with an alternative mechanism of TAD formation. Given that the above-mentioned cohesin-driven loop extrusion is evolutionarily conserved from bacteria to

mammals[69], it is compelling to assume that extrusion works in *Drosophila* as well. Despite the presence of all potential components of LEC (cohesin, its loading and releasing factors), TAD boundaries in *Drosophila* are not significantly enriched with CTCF[24,25] and do not form CTCF-enriched interactions or TAD corner peaks. These observations suggest that the binding sites of CTCF or other distinct proteins do not constitute barrier elements for the *Drosophila* LEC even if these proteins are enriched in TAD boundaries; this may be due to some other properties of a genomic region. For example, stably bound cohesins were proposed to act as the barriers for cohesin extrusion in yeast[70].

Active transcription interferes with DNA loop extrusion[71,72]. Because TAD boundaries in *Drosophila* are highly transcribed, we propose that open chromatin with actively transcribing polymerase and/or a high density of chromatin remodeling complexes could serve as a barrier for the *Drosophila* LEC. Contrary to the strictly positioned and short CBSs in mammals, active loci flanking *Drosophila* TADs represent relatively extended regions up to several dozens of kb in length. Probabilistic termination of LEC at varying points within such regions in different cells of the population could explain the absence of canonical loop signals and the presence of strong compartment-like interactions between active regions flanking a TAD (Fig. 6c). This model also provides a potential explanation for the relatively high stability of TAD positioning in individual *Drosophila* cells in comparison to mammals. A relative permeability of CBSs in mammalian cells allows LEC to proceed through thousands of kilobases and to produce large contact domains[17]. Extended active regions acting as "blurry" barrier elements where LEC termination occurs at multiple points, should stop the LEC more efficiently, making the TAD pattern more stable and pronounced.

Taken together, the order in the *Drosophila* chromatin 3D organization is manifested in a TAD profile that is relatively stable between individual cells and likely dictated by the distribution of active genes along the genome. On the other hand, our molecular simulations of individual haploid X chromosomes indicate a prominent stochasticity in both the form of individual TADs and the overall folding of the entire chromosome territory. According to our data, the active A-compartment is easily detectable in individual cells, and the profiles of interaction between individual active regions are highly variable between individual cells. Notably, this also holds true for Polycomb-occupied loci that are known to shape chromatin fiber in living cells[48].

Although these highly variable long-range interactions of active regions and Polycomb-occupied loci are closely related to the shape of chromosome territory (CT), the cause-and-effect relationships between them and the stochastic nature of the cell-specific chromatin chain path are currently unclear. The main question to be answered by future studies is whether these interactions are fully stochastic or at least partially specific. The possible molecular mechanisms that may provide specific communication between remote genomic loci separated by up to megabases of DNA are not known. In a scenario of the absence of any specificity, the pattern of contacts inside A-compartment and within Polycomb bodies in a particular cell is established by stochastic fluctuations of the large-scale chromatin fiber folding. In this case, the large-scale chromatin fiber folding dictates the cell-specific location of Polycomb-enriched and active chromatin regions in the 3D nuclear space. The formation of Polycomb bodies and transcription-related chromatin hubs is achieved by confined diffusion of these regions and might be further stabilized by specific protein-protein interactions and liquid-liquid phase separation[73]. This mechanism allows to sort through alternative configurations of the 3D genome and to transiently stabilize those that are functionally relevant under specific conditions. A balance

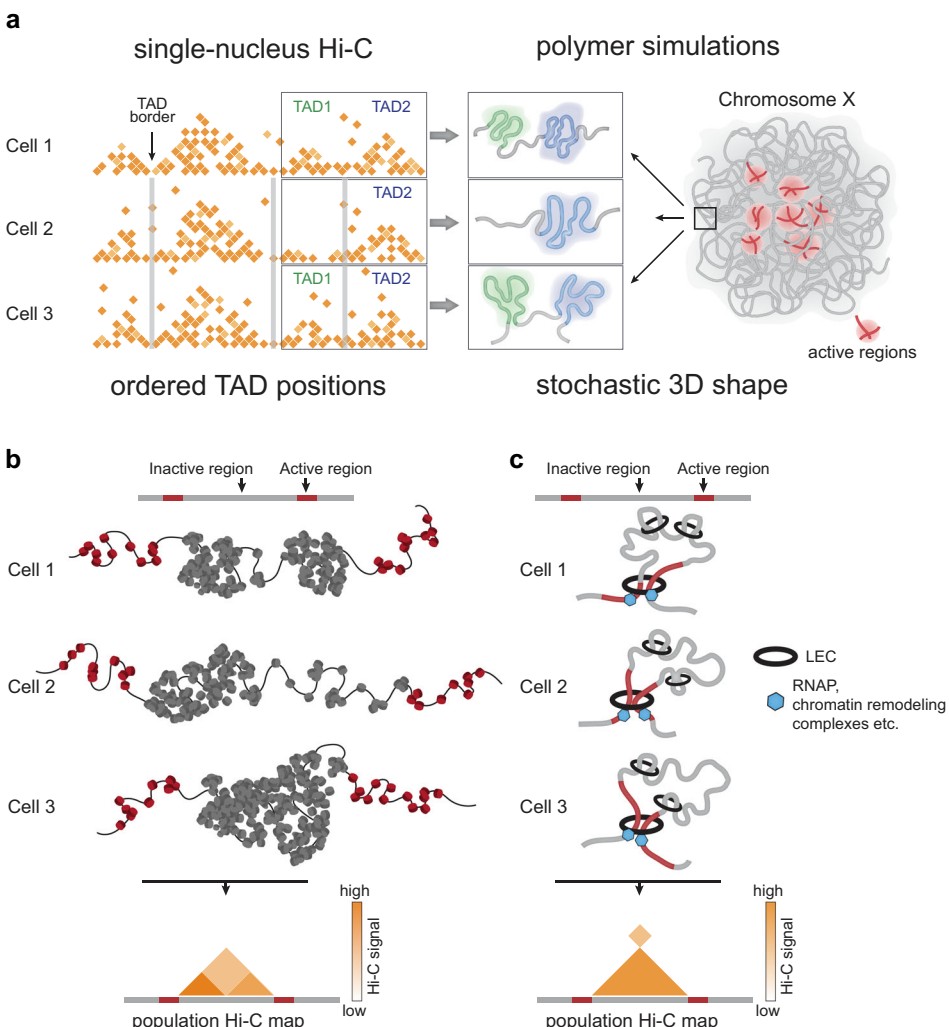

**Fig. 6 Order and stochasticity in the _Drosophila_ 3D genome. a** Schematic representation of the ordered and stochastic components in the _Drosophila_ genome folding. Positions of TAD boundaries are largely conservative between individual cells and determined by active chromatin. Chromatin fiber path within a particular TAD and within the whole chromosome territory is largely stochastic and demonstrate prominent cell-to-cell variability. **b** Determined positions of active regions along the _Drosophila_ genome define TAD boundaries persistent in individual cells. Inactive region is folded into chromatin globule due to interactions between non-acetylated "sticky" nucleosomes. This region adopts different configurations in individual cells (and at different time points in a particular cell). In a cell 1, it is folded into two globules separated with stochastically formed fuzzy boundary. In a cell 2, one part of the region is compact (left) and the other part (right) is transiently decondensed. In a cell 3, the entire region forms one densely packed globule. Averaging of these configuration results in a TAD containing two sub-TADs in a population-based Hi-C map. Note, that the hierarchical structure of the TAD emerging in a population Hi-C map reflects different configuration of the region in individual cells. We note that the absence of any structure at inactive TAD borders denotes ambiguity of folding of these regions with snHi-C, but not the absence of this structure. **c** Extended active regions serving as barrier elements for potential loop extrusion complex (LEC) in _Drosophila_ cells. It has been previously shown that transcription might interfere with loop extrusion[71,72]. Since stable TAD boundaries in _Drosophila_ are enriched with transcribed genes, we propose that extended regions of active chromatin but not binding sites of architectural proteins represent barrier elements for LEC in _Drosophila_ cells. In this scenario, LEC is looping out a TAD and terminates within flanking active regions colliding with RNA-polymerases, large chromatin-remodeling complexes and other components of active chromatin. In different individual cells, termination occurs accidentally at different points within these regions. In a population-based Hi-C map that results in a compartment-like signal but not in a conventional pointed loop observed in mammalian cells where CTCF binding sites serve as barrier elements for LEC.

between the order and the stochasticity appears to be an intrinsic property of nuclear organization that enables rapid adaptation to changing environmental conditions.

## Methods

**Cell culture**. _Drosophila melanogaster_ ML-DmBG3-c2 cell line (Drosophila Genomics Resource Center) was grown at 25 °C in a mixture (1:1 v/v) of Shields and Sang M3 insect medium (Sigma) and Schneider's Drosophila Medium (Gibco) supplemented with 10% heat-inactivated fetal bovine serum (FBS, Gibco), 50 units/ml penicillin, and 50 μg/ml streptomycin.

**Single-nucleus Hi-C library preparation**. We modified the previously published single-nucleus Hi-C protocol[32] as follows: 5–10 million cells were fixed in 1× phosphate-buffered solution (PBS) with 2% formaldehyde for 10 min with occasional mixing. The reaction was stopped by the addition of 2 M glycine to give a final concentration of 125 mM. Cells were centrifuged (1000 × _g_, 10 min., 4 °C), resuspended in 50 μl of 1× PBS, snap-frozen in liquid nitrogen, and stored at −80 °C. Defrozen cells were lysed in 1.5 ml isotonic buffer (50 mM Tris-HCl pH 8.0, 150 mM NaCl, 0.5% (v/v) NP-40 substitute (Fluka), 1% (v/v) Triton-X100 (Sigma), 1× Halt™ Protease Inhibitor Cocktail (Thermo Scientific)) on ice for 15 min. Cells were centrifuged at 2500 × _g_ for 5 min, resuspended in 100 μl of 1× DpnII buffer (NEB), and pelleted again. The pellet was resuspended in 200 μl of 0.3% SDS in 1.1× DpnII buffer and incubated at 37 °C for 1 h. Then, 330 μl of 1.1× DpnII buffer and 53 μl of 20%

Triton X-100 (Sigma) were added, and the suspension was incubated at 37 °C for 1 h. Next, 600 U of DpnII enzyme (NEB) were added, and the chromatin was digested overnight (14–16 h) at 37 °C with shaking (1400 rpm). On the following day, 200 U of DpnII enzyme were added, and the cells were incubated for an additional 2 h. DpnII was then inactivated by incubation at 65 °C for 20 min. Nuclei were centrifuged at 3000 × g for 5 min, resuspended in 100 µl of 1× T4 DNA ligase buffer (Fermentas), and pelleted again. The pellet was resuspended in 400 µl of 1× T4 DNA ligase buffer, and 75 U of T4 DNA ligase (Fermentas) were added. Chromatin fragments were ligated at 16 °C for 6 h. Next, the nuclei were centrifuged at 5000 × g for 5 min, resuspended in 100 µl of sterile 1× PBS, stained with Hoechst, and single nuclei were isolated into wells of a standard 96-well PCR plate (Thermo Fisher) using FACS (BD FACSAriaTMIII). Each well contained 3 µl of sample buffer from the Illustra GenomiPhi v2 DNA amplification kit (GE Healthcare). Sample buffer drops containing isolated nuclei were covered by 5 µl of mineral oil (Thermo Fisher) and incubated at 65 °C for 3 h to reverse formaldehyde cross-links. Total DNA was amplified according to a previously published protocol[74]. The amplification was considered successful if the sample contained ≥1 µg DNA. The DNA was then dissolved in 500 µl of sonication buffer (50 mM Tris-HCl (pH 8.0), 10 mM EDTA, 0.1% SDS) and sheared to a size of ~100–1,000 bp using a VirSonic 100 (VerTis). The samples were concentrated (and simultaneously purified) using AMICON Ultra Centrifugal Filter Units to a total volume of about 50 µl. The DNA ends were repaired by adding 62.5 µl MQ water, 14 µl of 10× T4 DNA ligase reaction buffer (Fermentas), 3.5 µl of 10 mM dNTP mix (Fermentas), 5 µl of 3 U/µl T4 DNA polymerase (NEB), 5 µl of 10 U/µl T4 polynucleotide kinase (NEB), 1 µl of 5 U/µl Klenow DNA polymerase (NEB), and then incubating at 20 °C for 30 min. The DNA was purified with Agencourt AMPure XP beads and eluted with 50 µl of 10 mM Tris-HCl (pH 8.0). To perform an A-tailing reaction, the DNA samples were supplemented with 6 µl 10× NEBuffer 2, 1.2 µl of 10 mM dATP, 1 µl of MQ water, and 3.6 µl of 5 U/µl Klenow (exo-) (NEB). The reactions were carried out for 30 min at 37 °C in a PCR machine, and the enzyme was then heat-inactivated by incubation at 65 °C for 20 min. The DNA was purified using Agencourt AMPure XP beads and eluted with 100 µl of 10 mM Tris-HCl (pH 8.0). Adapter ligation was performed at 22 °C for 2.5 h in the following mixture: 41.5 µl MQ water, 5 µl 10× T4 DNA ligase reaction buffer (Fermentas), 2.5 µl of Illumina TruSeq adapters, and 1 µl of 5 U/µl T4 DNA ligase (Fermentas). Test PCR reactions containing 4 µl of the ligation mixture were performed to determine the optimal number of PCR cycles required to generate sufficient PCR products for sequencing. The PCR reactions were performed using KAPA High Fidelity DNA Polymerase (KAPA) and Illumina PE1.0 and PE2.0 PCR primers (10 pmol each). The temperature profile was 5 min at 98 °C, followed by 6, 9, 12, 15, and 18 cycles of 20 s at 98 °C, 15 s at 65 °C, and 20 s at 72 °C. The PCR reactions were separated on a 2% agarose gel containing ethidium bromide, and the number of PCR cycles necessary to obtain a sufficient amount of DNA was determined based on the visual inspection of gels (typically 12–15 cycles). Four preparative PCR reactions were performed for each sample. The PCR mixtures were combined, and the products were separated on a 1.8% agarose gel. 200–600 bp DNA fragments were excised from the gel and purified with a QIAGEN Gel Extraction Kit.

**Bulk BG3 in situ Hi-C library preparation.** Bulk BG3 in situ Hi-C libraries were prepared as described previously[24] with minor modifications. The first steps of the protocol (from fixation to DpnII enzyme inactivation) were completely identical to the corresponding steps in the single-cell Hi-C library preparation procedure described above. After DpnII inactivation, the nuclei were harvested for 10 min at 5000 × g, washed with 100 µl of 1× NEBuffer 2, and resuspended in 50 µl of 1× NEBuffer 2. Cohesive DNA ends were biotinylated by the addition of 7.6 µl of the biotin fill-in mixture prepared in 1× NEBuffer 2 (0.025 mM dATP (Thermo Scientific), 0.025 mM dGTP (Thermo Scientific), 0.025 mM dTTP (Thermo Scientific), 0.025 mM biotin-14-dCTP (Invitrogen), and 0.8 U/µl Klenow enzyme (NEB)). The samples were incubated at 37 °C for 75 min with shaking (1400 rpm). Nuclei were centrifuged at 3000 × g for 5 min, resuspended in 100 µl of 1× T4 DNA ligase buffer (Fermentas), and pelleted again. The pellet was resuspended in 400 µl of 1× T4 DNA ligase buffer, and 75 U of T4 DNA ligase (Fermentas) were added. Chromatin fragments were ligated at 20 °C for 6 h. The cross-links were reversed by overnight incubation at 65 °C in the presence of proteinase K (100 µg/ml). After cross-link reversal, the DNA was purified by single phenol-chloroform extraction followed by ethanol precipitation with 20 µg/ml glycogen (Thermo Scientific) as the co-precipitator. After precipitation, the pellets were dissolved in 100 µl 10 mM Tris-HCl pH 8.0. To remove residual RNA, samples were treated with 50 µg of RNase A (Thermo Scientific) for 45 min at 37 °C. To remove residual salts and DTT, the DNA was additionally purified using Agencourt AMPure XP beads (Beckman Coulter). Biotinylated nucleotides from the non-ligated DNA ends were removed by incubating the Hi-C libraries (2 µg) in the presence of 6 U of T4 DNA polymerase (NEB) in NEBuffer 2 supplied with 0.025 mM dATP and 0.025 mM dGTP at 20 °C for 4 h. Next, the DNA was purified using Agencourt AMPure XP beads. The DNA was then dissolved in 500 µl of sonication buffer (50 mM Tris-HCl (pH 8.0), 10 mM EDTA, 0.1% SDS) and sheared to a size of approximately 100–1000 bp using a VirSonic 100 (VerTis). The samples were concentrated (and simultaneously purified) using AMICON Ultra Centrifugal Filter Units to a total volume of approximately 50 µl. The DNA ends were repaired by adding 62.5 µl MQ water, 14 µl of 10× T4 DNA ligase reaction buffer (Fermentas), 3.5 µl of 10 mM dNTP mix (Fermentas), 5 µl of 3 U/µl T4 DNA polymerase (NEB), 5 µl of 10 U/µl T4 polynucleotide kinase (NEB), 1 µl of 5 U/µl Klenow DNA polymerase (NEB), and then incubating at 20 °C for 30 min. The DNA was purified with Agencourt AMPure XP beads and eluted with 50 µl of 10 mM Tris-HCl (pH 8.0). To perform an A-tailing reaction, the DNA samples were supplemented with 6 µl 10× NEBuffer 2, 1.2 µl of 10 mM dATP, 1 µl of MQ water, and 3.6 µl of 5 U/µl Klenow (exo−) (NEB). The reactions were carried out for 30 min at 37 °C in a PCR machine, and the enzyme was then heat-inactivated by incubation at 65 °C for 20 min. The DNA was purified using Agencourt AMPure XP beads and eluted with 100 µl of 10 mM Tris-HCl (pH 8.0). Biotin pulldown of the ligation junctions was performed as described previously, with minor modifications. Briefly, 4 µl of MyOne Dynabeads Streptavidin C1 (Invitrogen) beads were used to capture the biotinylated DNA, and the volumes of all buffers were decreased by 4-fold. The washed beads with captured ligation junctions were resuspended in 50 µl of adapter ligation mixture comprising 41.5 µl MQ water, 5 µl 10× T4 DNA ligase reaction buffer (Fermentas), 2.5 µl of Illumina TruSeq adapters, and 1 µl of 5 U/µl T4 DNA ligase (Fermentas). Adapter ligation was performed at 22 °C for 2.5 h, and the beads were sequentially washed twice with 100 µl of TWB (5 mM Tris-HCl (pH 8.0), 0.5 mM EDTA, 1 M NaCl, 0.05% Tween-20), once with 100 µl of 1× binding buffer (10 mM Tris-HCl (pH 8.0), 1 mM EDTA, 2 M NaCl), once with 100 µl of CWB (10 mM Tris-HCl (pH 8.0) and 50 mM NaCl), and then resuspended in 20 µl of MQ water. Test PCR reactions containing 4 µl of the streptavidin-bound Hi-C library were performed to determine the optimal number of PCR cycles required to generate sufficient PCR products for sequencing. The PCR reactions were performed using KAPA High Fidelity DNA Polymerase (KAPA) and Illumina PE1.0 and PE2.0 PCR primers (10 pmol each). The temperature profile was 5 min at 98 °C, followed by 6, 9, 12, 15, and 18 cycles of 20 s at 98 °C, 15 s at 65 °C, and 20 s at 72 °C. The PCR reactions were separated on a 2% agarose gel containing ethidium bromide, and the number of PCR cycles necessary to obtain a sufficient amount of DNA was determined based on the visual inspection of gels (typically 12–15 cycles). Four preparative PCR reactions were performed for each sample. The PCR mixtures were combined, and the products were separated on a 1.8% agarose gel. 200–600 bp DNA fragments were excised from the gel and purified with a QIAGEN Gel Extraction Kit. Two biological replicates were performed.

**snHi-C raw data processing and contact annotation.** The whole-genome amplification step of snHi-C uses the Phi29 DNA polymerase, which is known to produce chimeric DNA molecules by randomly switching the DNA template[40]. DNA molecules created by the template switch were further amplified during the snHi-C protocol and resulted in chimeric reads. Notably, in theory, template switches can be detected by the presence of two consecutive parts of the same read that map to different genomic locations and do not align immediately next to the restriction sites at the DNA breakpoint. This situation is different from the standard Hi-C, where each read pair is considered to be a true contact pair regardless of the DNA breakpoint presence and annotation. Standard Hi-C processing tools, such as hiclib[32,41], Juicer[75], and HiCExplorer[26], typically rely on mapping of both reads in a Hi-C pair and do not account for the presence of chimeric parts in a single side of paired-end sequencing. We devised a more accurate approach for processing of snHi-C data that annotates each DNA breakpoint observed in each single-end read, and selects the contacts that do not represent possible template switches of Phi29 polymerase. Thus, we developed a custom approach for snHi-C data processing termed ORBITA (One Read-Based Interaction Annotation), as described below.

**Reads mapping.** As the first step of the approach, FASTQ files with paired-end sequencing data are mapped to Drosophila reference genome dm3 using Burrows-Wheeler Aligner (BWA-MEM, console version 0.7.17-r1188)[76] with default parameters. Notably, this mapping procedure allows independent alignment of chimeric parts of both forward and reverse reads. This step results in BAM files with paired-end mapping information.

**Annotated pairs retrieval.** In the next step, the BAM files are parsed with an adapted version of pairtools (https://github.com/mirnylab/pairtools) with our newly implemented option ORBITA. Among many other utilities for Hi-C data processing, we selected pairtools from the Mirny lab as the basis of our approach, due to the convenience and modular structure of its code. This version of the tool can be accessed at the GitHub repository https://github.com/agalitsyna/pairtools.

ORBITA treats each read in the BAM file independently, regardless of whether it is forward or reverse. Reads that are uniquely mapped to a single location of the genome are marked as type P, meaning that they are part of a standard Hi-C Pair with no DNA breakpoint evidence. Reads that contain precisely two successive regions uniquely mapped to different genomic locations (MAPQ > 1) are selected for further DNA breakpoint annotation. ORBITA takes the genome restriction annotation (provided as a BED file with DpnII restriction fragments positions, produced by cooler digest[77]) and compares each breakpoint against the list of restriction sites. For each 3′-end of the right chimeric part and 5′-end of the left chimeric part (in other words, ligated ends), both upstream and downstream restriction sites are annotated, and the distance to the closest one is calculated. If both ends are located sufficiently close (<10 bp) to any restriction site in the genome, ORBITA considers them as a true ligation junction of restricted fragments

in the snHi-C proximity ligation step. These cases are marked as J type (ligation Junction), with the evidence of traversing the ligation junction of DpnII restriction fragments. If at least one ligated end of the chimeric read was not mapped to the restriction site, ORBITA marks it as H (template switch, or Hopping of Phi29 DNA polymerase). To simplify the ORBITA approach, we omit the cases with more complicated scenarios of read mapping, when three or more uniquely mapped chimeric parts of a single-end read were present. If the read contains multiple mapped chimeric parts, it is discarded. ORBITA produces the resulting PAIRS file with annotation of JJ pairs (with the evidence of the ligation) that are accepted for further processing. If not explicitly mentioned, the generic names "pair" or "contact" are used for snHi-C contacts with the evidence of the ligation junction.

**Amplification duplicates removal.** In the next step, we performed a correction for amplified duplicates of snHi-C contacts. Standard Hi-C uses amplification by the Illumina PCR protocol with primers that are ligated to the ends of sheared DNA[17]. Thus, two independent Hi-C pairs can be PCR duplicates if their mapping positions coincide (e.g., see hiclib). However, the amplification in snHi-C[32] is followed by sonication, resulting in random breaks of ligated DNA fragments. Hence, coinciding mapping positions cannot be used as a criterion of PCR duplication. Notably, we cannot distinguish the amplified pair contacting restriction fragments from the contacts of the same regions in the homologous chromosomes. Thus, we removed all multiple copies of restriction fragment pairs and retained unique contacts for each combinatorial pair of restriction fragments.

**Fragment filtration.** In the next step, we used restriction fragment filtration to reduce the possible contribution of copy number variation, read misalignment, and Phi29 DNA polymerase template switch that had not been removed by the ORBITA filter.

In theory, each restriction fragment of DNA has two ends and is present twice in the diploid nucleus of ML-DmBG3-c2 *Drosophila* cells; thus, we expect the upper limit of four unique contacts per restriction fragment if no unannotated genomic rearrangements, mismappings, or template switches occurred. For each restriction fragment, we calculated the observed number of contacts and removed fragments that had more than four contacts.

Before contact filtration by this rule, we compared the number of restriction fragments with more than four unique contacts according to ORBITA and one previous approach, hiclib for Flyamer et al. 2017. We obtained datasets for mouse nuclei from Flyamer et al. 2017 and Nagano et al. 2017 and mapped with the hiclib and ORBITA pipelines. We found a significant reduction in the number of unique contacts per fragment for snHi-C from Phi29 DNA polymerase datasets (Flyamer et al. 2017, present work], but not for scHi-C without Phi29 DNA polymerase (Nagano et al. 2017) (Supplementary Figs. 2, 3). Thus, we conclude that ORBITA is an effective approach to reduce the number of snHi-C artefactual contacts arising from random template switches of Phi29 DNA polymerase.

**Cell selection by raw data subsampling.** We obtained filtered contacts for 88 individual nuclei after the initial round of sequencing. Before the second round of sequencing, we assessed the robustness of the number of unique contacts by subsampling of raw datasets (Supplementary Fig. 2a). For each library, we created a uniform grid of sequencing depth (from 0 to the resulting number of reads with the step of 100,000 reads). We then randomly selected X reads from the full library and calculated the number of unique contacts (as described above) for each number from the grid X. We repeated this procedure ten times and plotted the mean number of unique contacts for each sequencing depth from the grid.

We proposed that there are a significant number of cells containing PCR duplicates and that the number of contacts increases slowly depending on the sequencing depth due to the poor efficiency of the snHi-C protocol. Further sequencing of these cells would result in a relatively small improvement of the detectable number of unique contacts. The number of contacts for other cells increases more rapidly with the number of reads but reaches a plateau once the maximum number of unique contacts is achieved. Thus, additional sequencing of these cells might result in reading duplicated contacts.

For other cells, the number of contacts grew slowly with sequencing depth (Supplementary Fig. 2a). However, for all these cells, the number of unique contacts gradually increased with no plateau signature. We selected the cells displaying the best growth of the number of contacts, indicative of the good quality of the dataset. The top 20 cells by the number of unique contacts were subjected to an additional round of sequencing. The same mapping and parsing pipeline was used for these datasets. Technical replicates (initial and additional rounds of snHi-C libraries sequencing) were merged at the annotated PAIRS file stage.

**snHi-C interaction map construction.** The resulting pair data were binned at 1 kb, 10 kb. 20-kb, 40-kb, and 100-kb resolutions with *cooler* version 0.8.5[77] and stored in the COOL format. We constructed the merged dataset by summing all snHi-C maps. To exclude self-interacting genomic bins and possible contribution of dangling ends, self-circles[41], and mirror reads[78], we removed the first diagonal in both single cells and the merged maps. The *HiGlass* server was used for data visualization[79]. 10-kb resolution was used throughout the paper if another resolution is not specified.

**Bulk BG3 in situ Hi-C raw data processing.** For bulk BG3 in situ Hi-C (two biological replicates), reads were mapped to *Drosophila* reference genome dm3 with Burrows-Wheeler Aligner (BWA-MEM, console version 0.7.17-r1188)[76] with default parameters. For consistency with the snHi-C analysis, the resulting BAM files were parsed with *pairtools* v0.3.0, (https://github.com/mirnylab/pairtools) using default parameters. The resulting files were sorted by the *pairtools* module "sort"; replicates were merged by the *pairtools* module "merge" and duplicates were removed, allowing one mismatch between possible duplicates (*pairtools* dedup with --max-mismatch 1 and —mark-dups options). The resulting PAIRS file was binned with *cooler*[77] at the same resolutions as the single-cell datasets. To remove the contribution of possible Hi-C technical artifacts, such as backward ligation, dangling ends, self-circles[41], and mirror reads[78], the first two diagonals of Hi-C maps were removed. As the last step of bulk Hi-C processing, the maps were iteratively corrected for the removal of coverage bias[41] by the *cooler* balance tool with default parameters[77].

For the reproducibility control, both replicates were converted to interaction maps independently by the above pipeline. The resulting maps demonstrated a correlation of 0.9–0.95 as estimated by the *HiCRep* stratum-adjusted correlation coefficient for intrachromosomal maps smoothed with one-bin offset and genomic distance up to 300 kb at 20 kb resolution[80].

**snHi-C background model construction.** We sought to create a background model for snHi-C that can be used as a control for the subsequent analysis of intrachromosomal snHi-C interaction maps. For that, we considered two major factors contributing to the intrachromosomal contact frequency in the genomic region: the contact probability for a particular genomic distance $P_c(s)$[13], and region visibility[81].

For bulk BG3 in situ Hi-C, the $P_c(s)$ is assessed by the mean number of contacts for a certain genomic distance[13]. However, the same procedure cannot be readily used for snHi-C due to data sparsity and missing data. Thus, to calculate $P_c(s)$ for a snHi-C dataset, we counted the number of contacts for a certain genomic distance and normalized by the number of genomic bins that had contact in at least one snHi-C experiment at any distance. Notably, we use the same procedure for the visualization of snHi-C $P_c(s)$ dependence on the genomic distance $s$ (Fig. 1f and Fig. 4e); the genomic distance step size was set to 1 kb. For snHi-C background models, we used $P_c(s)$ genomic distance step size 10 kb.

We assessed the region visibility in snHi-C by the marginal distribution of the number of contacts for the region $marg_i$ (in other words, the total number of observed intrachromosomal contacts for a genomic region) using maps at a 10-kb resolution.

For each snHi-C map, we calculated $P_c(s)$ and the marginal distribution of contacts and shuffled the positions of the contacts for each chromosome, so that the marginal distribution was preserved, and $P_c(s)$ was at least approximated (Supplementary Fig. 4a–d). Note that for 3D modeling, we used more crude shuffling without saving the marginal distribution of contacts.

**Assessment of percentage of recovered contacts.** To compare snHi-C datasets across species (Fig. 2a–c), we assessed the percentage of recovered contacts out of all possible contacts per nuclei.

First, we determined the theoretical size of the pool of restriction fragments for the nucleus of each species and cell type. For *Drosophila*, we used a diploid male cell line. Thus, the total number of restriction fragments was ~600,000, composed of the double amount of fragments in autosomes ($2 \times 265,167$, as assessed by the dm3 in silico digestion) in addition to the number of fragments on chromosome X (64,108). For mice, Flyamer et al. (2017) analyzed oocytes with four copies of the genome, resulting in a total of $4 \times 6,407,802 \sim 25,600,000$ fragments. Gassler et al. (2017) analyzed G2 zygotes pronuclei with two copies of the genome, resulting in a total of $2 \times 6,407,802 \sim 12,800,000$ fragments (we did not distinguish between the maternal and paternal pronuclei because the contribution of chromosome X is not as significant for the mouse genome).

We next assessed the upper limit of the total number of possible contacts per single nucleus, which is achieved when each restriction fragment formed two contacts with the ends of any other restriction fragments from the pool. Because the valency of each fragment is two, the theoretical upper limit is equal to the number of restriction fragments.

We then divided the total number of observed contacts (recovered by ORBITA) by the upper bound of the possible number of contacts, and we recovered up to ~16% of the total number of possible contacts for *Drosophila* (see Fig. 2b); this number is approximately 2.6% for the best mouse dataset. The mean percentage of recovered contacts is 4.9% for our dataset and <1% for Flyamer et al. (2017) and Gassler et al. (2017).

However, this assessment of the percentage of recovered contacts is not exact for several reasons: (1) we did not perform sorting prior to snHi-C to isolate G1 cells; hence, some regions of the genome might have an increased copy number in S or G2 cells; (2) some regions of the genome might be affected by deletions and copy number variations that were not accounted for in our analysis. However, even in the worst-case scenario, if we imagine that all *Drosophila* cells are in the G2 phase of the cell cycle, we recovered at least 8% of all possible contacts for the best cells in our analysis, which is still a substantial improvement compared to recovery for the best cells from mammalian studies.

**TAD calling in snHi-C and bulk BG3 in situ Hi-C data**. We used Hi-C map segmentation with *lavaburst* (v0.2.0) (https://github.com/nvictus/lavaburst) with the modularity scoring function for TAD calling in Hi-C maps at 10-kb resolution[32]. All TAD segments smaller or equal to 3 bins (30 kb) were considered to be inter-TADs[24]. *lavaburst* has a gamma ($\gamma$) parameter controlling the size and the number of resulting TADs. We varied g from 0 to 375 with a step of 0.1 for *Drosophila* datasets. The range and the step were selected to guarantee the comprehensive coverage of both extremes (a small amount of unusually large TADs and a large amount of smallest possible TADs). We observed a sharp decrease in median TAD size and an increase in the number of TADs with the $\gamma$ increase (Fig. 3b, Supplementary Fig. 5). After reaching the peak, the number of TADs starts to drop because many segments fall beyond the minimal allowed TAD size. For large $\gamma$, both the number of TADs and mean TAD size reach a plateau at low levels. We considered the point of the maximum number of TADs ($\gamma_{max}$) as the most informative segmentation reachable by the algorithm for a particular dataset. The mean TAD size is ~70 kb on average between cells compared to the expected 120 kb size of *Drosophila* TADs[24]. Thus, we considered this level to be the sub-TADs. To guarantee a uniform $\gamma$ selection procedure for all the cells, we arbitrarily selected $\gamma_{max}/2$ to obtain a resulting TAD segmentation (mean TAD size ~90 kb).

For the other resolutions of snHi-C maps, the same protocol of TAD calling was applied, except the inter-TAD size threshold was set to 60 kb (3 bins at 20 kb) for 20 kb and 120 kb (3 bins of 40 kb) for 40 kb.

**Robustness of TAD calling**. To assess TAD calling robustness and filter out potentially artifactual TAD boundaries, we performed TAD calling on snHi-C maps with random subsampling of the contacts as a control. For each cell, we performed ten iterations of independent subsampling of contacts leaving 95%, 90%, … 5% of the initial number of unique contacts per dataset. For each subsampling, we performed the TAD calling in the same manner as for the full dataset. We then assumed the bins found as TAD boundaries in the full snHi-C maps with no subsampling to be positives and inner TAD bins to be negatives. Based on this definition, we calculated both false positive rates (FPR) and false negative rates (FNR) for each cell and all subsampling levels. As expected, FNR gradually decreased with the percentage of remaining contacts. FPR reached a maxima at 10–30% subsampling level and then gradually decreased (Supplementary Fig. 6a, b).

We then defined a TAD boundary support for a given subsampling level (X%). TAD boundary support is calculated for each genomic bin as the number of subsampling iterations with the number of contacts equal to or larger than X%, where the bin was annotated as the TAD boundary (allowing a one-bin offset). We used TAD boundary support as a predictor of observed TAD boundaries in each cell (with no subsampling of the snHi-C dataset). We plotted receiver operating characteristic (ROC) curves for each X = (95%, 90%, … 5%) and calculated the ROC area under the curve (AUC) for each case (Supplementary Fig. 6c). Based on the largest ROC AUC, we selected the best subsampling level predictive of boundaries, X = 90% ROC AUC 0.9969 (Supplementary Fig. 6c). We then chose the TAD boundary support threshold by optimizing the accuracy. We obtained an accuracy of 0.9765 for the final criteria that the TAD boundary support is larger than 45% for (90%..95%) subsampling levels.

We refined the boundaries based on these final criteria and observed only a mild decrease in the number of boundaries per cell (Supplementary Fig. 6d). Thus, we conclude that the TAD calling procedure is robust to subsampling. We used the non-refined boundaries set in the paper if not stated otherwise.

For the refined boundaries set, we allowed a 10-kb offset for each boundary and assessed the number of cells in which each genomic bin was annotated as a boundary. We then defined the stable boundaries as bins that were annotated as boundaries in more than or equal to 50% of cells ($>= 7$), and unstable boundaries as the bins annotated as boundaries in less than 50% of cells (<7).

We compared stable boundaries with boundaries conserved between Kc167 and BG3 cells[46]. For that, we obtained TAD positions from[46], mapped them to the dm3 genome with liftover, and coarse-grained the coordinates to 10-kb bins. We then allowed the 10-kb offset and counted the boundaries that overlapped with stable boundaries obtained in the single-cell analysis.

**Segmentation comparison**. We introduced two types of similarity scores for TAD/sub-TAD segmentation comparison:

(1) the percentage of shared boundaries, where we fixed the first segmentation and compared it with the second segmentation. Each TAD boundary bin of the second segmentation was allowed to include two of its closest neighbors at a 10 kb distance (one bin offset). The number of shared boundaries between two segmentations was calculated as a simple intersection of sets. The percentage was calculated by division by the total number of bins annotated as TAD boundaries in the first segmentation.

(2) Jaccard index for TAD bins, where the bins inside a TAD (excluding the boundaries) were considered. The shared TAD bins between two segmentations were calculated and divided by the total number of bins annotated as TADs in both segmentations.

To assess the significance of obtained similarity score of TADs, we randomized the locations of TAD boundaries preserving the distributions of TAD and inter-TAD sizes and the number of TADs/inter-TADs per chromosome. Each

randomization was performed 1000 times; the distribution of scores was approximated by Gaussian distribution; p-values were inferred from these backgrounds. The same procedure was used for sub-TADs.

**Non-backtracking approach for annotation of TADs in single cells contact maps**. The chromatin network, constructed on the basis of the single-cell Hi-C data, can be classified as sparse (i.e., the number of actual contacts per bin in a single-cell contact matrix (adjacency matrix of the network) is much less than the matrix size N). The sparsity of the data significantly complicates the community detection problem in single cells. It is known that upon dilution of the network, there is a fundamental resolution threshold for all community detection methods[82]. Furthermore, traditional operators (adjacency, Laplacian, modularity) fail far above this resolution limit (i.e., their leading eigenvectors become uncorrelated with the true community structure above the threshold)[43]. That is explained by the emergence of tree-like subgraphs (hubs) overlapping with true clusters in the isolated part of the spectrum for these operators. Localization on the hubs, but not on true communities in the network, is a drawback of all conventional spectral methods in the sparse regime.

To overcome the sparsity issue and to make spectral methods useful in the sparse regime, Krzakala et al.[43] proposed to construct the transfer-matrix of non-backtracking random walks (NBT) on a directed network. The NBT operator B is defined on the edges $i \to j$, $k \to l$ as follows:

$$B_{i \to j, k \to l} = \delta_{il}(1 - \delta_{jk}) \tag{1}$$

By construction, NBT walks cannot revisit the same node on the subsequent step and, thus, they do not concentrate on hubs. It has been shown that the non-backtracking operator is able to resolve the community structure in a sparse stochastic block model up to the theoretical resolution limit. In recently published paper[42], we have proposed the neutralized towards the expected contact probability NBT operator for the sake of a large-scale splitting of a sparse polymer network into two compartments.

Here, we are interested in the small-scale clustering into TADs, for which the conventional NBT operator is appropriate. To eliminate the compartmental signal from the data, we first cleansed all chromosome contact matrices starting from the diagonal, corresponding to 1 Mb separation distance (100th diagonal in the 10-kb resolution). To respect the polymeric nature of the contact matrices, we have filled all empty cells on the leading sub-diagonals with 1. Then, the NBT spectra of all single-cell contact matrices were computed. The majority of eigenvalues of the non-Hermitian NBT operator are located inside the disc in a complex plane, and some number of isolated eigenvalues with large amplitudes lie on the real axis. The edge of the isolated part of the spectrum was defined as the real part of the largest in absolute value eigenvalue with a non-zero imaginary part. All eigenvalues $\lambda_i$ such that $R_e(\lambda_i) > r_c$ are isolated, and the corresponding eigenvectors correlate with annotation into the TADs. The position of the spectral edge, determined by the procedure above, has been found to be very close to the edge of the disk for the stochastic block model $r_c = \sqrt{d^{-1} \langle \frac{d}{d-1} \rangle}$, where $d$ is the vector of degrees[83]. The typical number of the isolated eigenvalues was around 100 for dense contact matrices and somewhat less for sparser ones. The leading eigenvectors define the coordinates $u_j^{(i)}$, $j = 1, 2, \dots, N$ of the nodes (bins) of the network in the space of reduced dimension $k << N$. At the second step, the clustering of the data was performed using the spherical k-means method, realized in the Python library *spherecluster*[84]. The number of isolated eigenvalues establishes a lower bound on the new space dimension $k$ to be used for the clustering algorithm, since the respective leading eigenvectors are linearly independent. The dimension of the space $k$ establishes a lower bound on the number of clusters because the leading eigenvectors are linearly independent. To take into account the hierarchical organization of TADs, we have communicated to the spherical k-means the number of clusters somewhat larger than the lower bound. Although the final splitting was found to be not particularly sensitive to this number, we have chosen to split the network into $2.5*k$ clusters in order to obtain the same mean amount of TADs per chromosome as with the modularity method (171 TADs).

The annotations produced by the spherical k-means on the single-cell Hi-C matrices were contiguous (i.e., the clusters were sequence respective, thus resembling TADs). The clusters (i) of size less than 30 kb and (ii) with amount of contacts equal to $2(l - 1)$ (i.e., with no contacts other than on the sub-diagonals) were excluded from the set as the inter-TADs regions. The ultimate median size of the TADs across all single cells obtained by this algorithm was 110 kb (from 60 kb to 260 kb), and the mean chromosome coverage was 82% (from 57 to 93%). The same analyses of shuffled contact maps have revealed a similar number, size, and coverage of the domains, formed purely due to fluctuations. The boundaries of the NBT TADs in single cells were significantly conserved from cell to cell: the mean pairwise fraction of matched boundaries was 44% for all the cells and 59% for the five densest ones (for the shuffled cells with preservation of stickiness and scaling, see the MSS model; the mean pairwise fraction was 38 and 50% for the five densest cells).

Regarding the comparison of TAD boundaries with the modularity approach, the mean fraction of conserved modularity boundaries is somewhat less – 42% for all pairs of cells in the analyses and 52% for the five densest cells, whereas the number of TADs per chromosome is the same in the two methods (171). Between

the two methods, the mean number of matched boundaries for the corresponding cells is 61%.

**Compartment annotation in snHi-C and bulk BG3 in situ Hi-C.** For compartment annotation in bulk BG3 in situ Hi-C, we used eigenvector decomposition of cis-interactions maps for each chromosome, as implemented in *cooltools* call-compartments tool version 0.2.0 (https://github.com/mirnylab/cooltools). We then reversed the sign of eigenvalues based on GC content (positive values corresponding to an A compartment with larger GC content)[26]. We next carried out a saddle plot analysis for each snHi-C dataset based on bulk BG3 in situ Hi-C compartment annotation[32]. For this procedure, the bins in raw scHi-C maps were reordered by ascending first eigenvector values and averaged to 5 × 5 saddle plots[32].

**Epigenetic analysis of TAD boundaries.** For the functional annotation of TAD boundaries, we downloaded modENCODE normalized array files[85]: total RNA of ML-DmBG3-c2 cell line assessed by RNA tiling array (modENCODE id 713) and the ChIP-chip for MOF (id 3041), BEAF-32 (id 921), Chriz (275), CP190 (924), CTCF (3280), dmTopo-II (5058), GAF (2651), H1 (3299), HP1a (2666), HP1b (3016), HP1c (942), HP2 (3026), HP4 (4185), ISWI (3030), JIL-1 (3035), mod (mdg4) (324), MRG15 (3045), NURF301 (5063), Pc (325), RNA-polymerase-II (950), Su(Hw) (951), Su(var)3-7 (2671), Su(var)3-9 (952), WDS (5148), H3 (3302), H3K27ac (295), H3K27me3 (297), H3K36me1 (299), H3K36me3 (301), H3K4me1 (2653), H3K4me3 (967), H3K9me2 (310), H3K9me3 (312), H4K16ac (316). For RNA-Seq coverage, we used the data from ref. [24]. The files were binned at 10-kb resolution by summation.

We plotted the ChIP-chip signal around different types of boundaries with *pybbi* utility (https://github.com/nvictus/pybbi.git) based on UCSC tools[86] and constructed six sets of boundaries: boundaries found in the bulk in situ Hi-C, boundaries found in the merged snHi-C dataset, boundaries present in > = 50% of cells (> = 7 cells, stable boundaries), boundaries present in <50% of cells (<7 cells, unstable boundaries), boundaries present in just one single cell, and random boundaries. To obtain randomized boundaries, we shuffled bulk in situ Hi-C boundaries across the *Drosophila* genome, preserving the number of boundaries per chromosome. We also used the bins from the inner parts of TADs as a control for the epigenetic analysis.

**Functional annotation of distant contacts.** The 10-kb genomic bins were separated into four groups based on chromatin states for BG3 from Kharchenko et al.[54]: active chromatin (>0.5 of RED and MAGENTA color), inactive chromatin (>0.5 LIGHT GRAY), Polycomb chromatin (>0.5 DARK GRAY), and unannotated (all the rest) for functional annotation of distant contacts. The thresholds for functional enrichment of particular types of chromatin were selected in order to guarantee the selection of the regions with the most prominent properties of active/inactive/Polycomb chromatin.

The 10-kb genomic bins were split into five groups based on the average expression from two RNA-seq replicates in BG3 cells[24] (0 expression, 38.1–40%, 40–60%, 60–80%, top 20% expression) for expression activity annotation. We were not able to split the data using an even grid of percentiles (e.g., 0–20%, 20–40%) because ~38% of all genomic bins had zero expression in both replicates. The same functional annotation was used later for polymer model coloring.

**Average loop.** For the construction of an average loop of A-compartment regions (Fig. 4f) and B compartment regions (Fig. 4g), MSL complex (Fig. 4h) and Polycomb (Fig. 4i), we selected the top 1000 genomic regions with the highest abundance of the corresponding genomic annotations as potential looping positions. A and B compartments were assessed by a *cis*-derived eigenvector of the bulk BG3 Hi-C data. MSL ChIP-Seq was obtained from Ramirez et al.[51], GEO ID GSE58821). dRING binding data were obtained from modENCODE as a ChIP-chip normalized array file (ID 927[54]). We considered the pairs of potential looping positions corresponding to intrachromosomal interactions, at the genomic distances of more than 600 kb, separated by up to 50 other looping positions. The snipping of Hi-C square 600-kb windows, centered on the corresponding looping positions, was done with cooltools (https://github.com/mirnylab/cooltools/tree/master/cooltools). The aggregation was performed by summation. $\log_{10}$ values were plotted as heatmaps.

**Assessment of folding hierarchy of TADs.** To assess the folding hierarchy at the level of TADs, we used the assumption that the successive sub-TADs that form the same TAD will have more interactions in the observed real snHi-C maps than in the control maps described in the section "snHi-C background model" of these Methods. We calculated the number of contacts directly from snHi-C maps and the control maps. Only sequential sub-TADs falling into the same TAD were considered. The distribution of the number of contacts in the windows between sequential sub-TADs was calculated. We compared the distributions of the number of contacts between sub-TADs falling into the same TAD for real snHi-C maps and the control maps. For each cell, we used either TAD/sub-TAD annotations from the corresponding snHi-C map or TAD/sub-TAD annotation from bulk in situ Hi-C.

**Marginal scaling (MS) and marginal scaling and stickiness (MSS) models.** We carried out the statistical analysis of the single-cell Hi-C maps to provide statistical arguments supporting the premise that the clustering observed in snHi-C contact matrices "is not random". For this, we used two different models of a polymer network based on Erdos-Renyi graphs, where bins of the contact map resemble graph vertices, and contacts between bins are graph edges[87] (Supplementary Fig. 4a):

(a) In the MS model, we require the probability of contact between nodes to respect the contact probability of the experimental contact map, i.e. $P(s) = P_c(|i-j|)$. Decay of the contact probability originates from the intrinsic linear connectivity of the chromatin nodes; therefore, it is an important ingredient for studying fluctuations in a polymer network. The probability of the link between $i$ and $j$ in the random graph I, $j = 1, 2..., N$ is, thus, defined as follows:

$$p_{ij} = \frac{P_c(|i-j|)}{\sum_{s=1}^{N-1}(N-s)P_c(s)}N_c \tag{2}$$

where the normalization factor in the denominator guarantees that the mean number of links in the graph equals $N_c$ (i.e., the number of experimentally observed links in each single cell). To obtain the average scaling, we merge all contacts from the available single cells and compute the average $P_c(s)$. Given the probability $p_{ij}$ by Eq. 2, we randomly generate adjacency matrices that have a homogenous distribution of contacts along the diagonals and do not respect local peculiarities of the bins, such as insulation score, acetylation, and protein affinity. Nevertheless, some non-homogeneity (clustering) of contacts still emerges as a result of stochasticity in each realization of this graph (Supplementary Fig. 4e).

(b) the MSS model introduces probabilistic non-homogeneity along the diagonals of the adjacency matrices through definition of the "stickiness" of bins, or. Specifically, under "stickiness", we understand a non-selective affinity $k_i$ of a bin $i$ to other bins; the probability that the bin $i$ forms a link with any other bin in the polymer graph is proportional to its stickiness. Thus, the clusters of contacts close to the main diagonal of contact matrices form as a result of different "stickiness" of bins in the MSS model. Stickiness might effectively emerge as a result of a particular distribution of "sticky" proteins, such as PcG proteins known to mediate bridging interactions between nucleosomes and to participate in stabilization of the repressed chromatin state.

Assuming that the stickiness is distributed independently of the polymer scaling $P_c(|i-j|)$, we use the following expression for the probability of the link, $p_{ij}$, in the MSS model:

$$p_{ij} = \frac{k_i k_j P_c(|i-j|)}{\sum_{i<j} k_i k_j P_c(|i-j|)}N_c \tag{3}$$

To derive the values of stickiness, we calculated the coverage at each bin in the merged contact map $\tilde{k}_i$, which stands for the average number of contacts at a particular bin. Due to the polymer scaling, the rates of contacts along each row (column) vary. Thus, $\tilde{k}_i$ is not equal to stickiness, $\tilde{k}_i \neq k_i$. To determine the stickiness values $k_i$, one should correlate the experimental coverage $\tilde{k}_i$ with the theoretical mean number of contacts per bin, according to Eq. 3:

$$\tilde{k}_i = \sum_j p_{ij} = k_i \alpha_i \tag{4}$$

where is "activity" of surrounding bins, measured for the $i$-th bin:

$$\alpha_i = \frac{1}{Z}\sum_j k_j P_c(|i-j|), \quad Z = \frac{1}{N_c}\sum_{i<j} k_i k_j P_c(|i-j|) \tag{5}$$

Equation 3 sets a system of $N$ non-linear equations that cannot be solved analytically. To determine the stickiness values, we implement the numerical method of iterative approximations. Namely, we start with:

$$k_i^{(0)} = \tilde{k}_i, \quad \alpha_i^{(0)} = \alpha_i(\tilde{k}_i) \tag{6}$$

and recalculate $k_i^{(1)}$ using Eqs. (4, 5) at the second step. After several recursive steps, we find good convergence of the stickiness and activity to their limiting values $k_i^{\infty}$ and $\alpha_i^{\infty}$. In particular, the derived values of the stickiness provide a good estimate for the averaged theoretical coverage $\tilde{k}_i$ as compared to the experimental coverage; see Supplementary Fig. 4f, g. Therefore, the derived null-model of single-cell maps reproduces, on average, the observed coverage of contacts of each bin by means of the individual stickiness assignment. We would like to point out the difference between the limiting values of the stickiness and $\tilde{k}_i$, used as a starting approximation in the iterative procedure; Supplementary Fig. 4h. This difference is a result of the non-homogeneous redistribution of contacts at each particular row in accordance with the marginal polymeric scaling $P_c(|i-j|)$.

**Number of contacts in windows.** The MS and MSS models introduced above demonstrate apparent clustering of generated contacts close to the main diagonal in realizations of adjacency matrices. In the MS model, this is purely due to fluctuations: the mean weight of the link $w_{ij} = p_s$ depends only on the genomic

distance between the bins $s = |i - j|$ in the respective Poisson version of the weighted network. In contrast, in the MSS model, the non-homogeneity of bin sicknesses allows for a deterministic non-homogeneous distribution of contacts along the main diagonal.

To statistically compare the clustering of contacts generated by the two models with the clustering in experimental single cell Hi-C maps, we studied distributions of the number of contacts in certain "windows" of different sizes. The inspected windows are isoscele triangles with the base located on the main diagonal and having the angle with the congruent sides. These windows look like TADs but, in contrast to the latter, have a fixed size throughout the genome.

At a given window size $W$, we sampled the number of contacts falling in the defined windows in each snHi-C map. We compared the samples originating from 100 random MS-generated maps and 100 random MSS-generated maps with derived limiting values of stickiness (see the previous section for discussion of the models).

Note that in the theoretical models (MS and MSS), all contacts are statistically independent: in both models, the number of contacts falling in a window of size can be interpreted as a number of "successes" occurring independently in a certain fixed interval. In the MS model, the "success" rate is constant along each diagonal; thus, for rather sparse MS maps (i.e. sufficiently small rates), one would expect the observed contacts in the windows to follow the Poisson distribution. In the MSS maps, the stickiness distributions introduce non-homogeneity to "success" rates along the diagonals; however, as our analyses suggest, the random MSS maps exhibit much more satisfactory Poisson statistics than their original experimental counterparts; Supplementary Fig. 4j, k.

Deviations from the Poisson statistics of the snHi-C contact maps are evaluated by the $p$-value of the $\chi^2$ goodness of fit test (Supplementary Fig. 4k). The heatmaps of the common logarithm of $p$-values for the top-10 single cells and the corresponding MS and MSS maps are presented in Supplementary Fig. 4j. The random maps (the second and third rows) demonstrate reasonably even distributions of the $p$-values across distinct single cells that rarely enter below the significance level $\alpha = 10^{-5}$. Several atypically low $p$-values correspond either to the most dense single cells and small window sizes (upper-left corner), for which the sparse Poisson limit is violated, or to a quite uneven distribution of stickiness for a given chromosome. Notably, the snHi-C maps demonstrate remarkable deviations from the Poisson statistics for small window size $W < 40$ bins (<400 kb). As can be seen from the heatmaps (Supplementary Fig. 4j) the $\chi^2$ test rejects the null hypothesis at the significance level $\alpha = 10^{-5}$ for most of the single cells at small scales. Therefore, the probability that the experimental contact maps are described by the Poisson statistics is significantly low ($\alpha$).

To understand the source of inconsistency between the experimental and Poisson distributions, we plotted the histograms of the number of contacts along with their best Poisson-fit for $W = 10$ (Supplementary Fig. 4k, left) and $W = 40$ (Supplementary Fig. 4k, right). The presence of large-scale heavy tails and low-scale shoulders in the experimental histograms results in the rejection of the null hypothesis.

Finally, the samples corresponding to larger windows are notably better described by the Poisson distribution, exhibiting a level of $p$-values similar to the random maps. The crossover $W_0 \approx 40$ (400 kb) corresponds to the scale of 3–4 typical TADs; this implies that the positioning of the contacts inside a single TAD is sufficiently correlated. Correlations between the contacts of different pairs of loci can originate from a specific non-ideal folding of chromatin (e.g., fractal globule) or be a signature of active processes (e.g., loop extrusion) operating at the scale of one TAD. Larger window sizes accumulate contacts from different TADs, whereas most of the inter-TADs contacts are much less correlated. As a result, we see reasonable Poisson statistics of the number of contacts from larger windows with $W > W_0$. Taken together, we conclude that correlations in contacts is a structural feature of experimental single cell maps and that clusters (TADs) identified in the maps cannot be reduced to random fluctuations imposed by the white noise or imperfections of the experimental setup.

**Fluorescence in situ hybridization**. The cells were harvested overnight on poly-l-lysine coated coverslips placed in culture flasks. The cells were fixed in 4% paraformaldehyde for 10 min, permeabilized in 0.5% Triton X-100, washed in PBS, dehydrated in ethanol series, air-dried, stored at room temperature for 2 days, and then frozen at −80 °C. Probes were prepared from fosmids by labeling with fluorophore-conjugated dUTPs using nick-translation. Approximately 150 ng of each probe was used in hybridization. Denaturation was performed at 80 °C for 30 min in 70% formamide (pH 7.5), 2× SSC. Hybridization of probes was done for 24 h in 50% formamide, 2× SSC, 10% dextran sulfate, 1% Tween 20. Washing steps were performed in 2× SSC at 45 °C followed by 0.1× SSC at 60 °C and 4× SSC, 0.1% Triton X-100. For imaging, cells were counterstained with DAPI, and epifluorescent images were acquired using a microscope setup comprising a Zeiss Axiovert 200 fluorescence microscope (Carl Zeiss UK, Cambridge, UK), X-Cite ExFo 120 Mercury Halide (Exfo X-cite 120, Excelitas Technologies) fluorescent source with liquid light guide and 10-position excitation, neutral density, and emission filter wheels (Sutter Instrument, Novato, CA), ASI PZ2000 3-axis XYZ stage with integrated piezo Z-drive (Applied Scientific Instrumentation, Eugene, OR), Retiga R1 CCD camera (Qimaging, Surrey, BC, Canada). The filter wheels were populated with a #89903 ET BV421/BV480/AF488/AF568/AF647 quinta set (Chroma Technology

Corp., Rockingham, VT). Image capture was performed using Micromanager 1.4 (https://open-imaging.com/). Hardware control and image capture were carried out using µManager[88]. Images were deconvolved using Nikon NIS-Elements. Measurements were taken using Imaris.

**Polymer simulations**. Simulation of 3D chromatin fiber enabled substantiation of assumptions about factors that play key roles in chromatin organization and to obtain important information about its packaging. We focused on the static properties of the system and did not consider its dynamic properties.

**Modeling pipeline, general description of the procedure**. Many methods are currently used to perform computer modeling of polymers. Due to the actual size and complexity of the chromatin, the all- or united-atom model cannot be used to simulate spatial scales of interest. The dissipative particle dynamics (DPD) technique was used because it enables modeling of the physical properties of polymer systems[59]. DPD is a coarse-grain method of molecular dynamics. Newton's equations are solved numerically for each particle in the system for every time step. The total force consists of conservative, dissipative, random, and elastic forces.

Conservative force is described by a soft potential within the sphere with cutting radius $R_c = 1.0$. The soft potential has no singularity at the zero point (Supplementary Fig. 21a). It is possible to use a large time step in the Velocity Verlet integration scheme, in contrast to classical molecular dynamics (CMD) with the Lennard-Jones potential. The typical time step in CMD is 20 times smaller than in DPD. The solvent is taken into account explicitly; it is necessary for the DPD thermostat to work[89,90]. The temperature control of the system is ensured by a balance of dissipative and random forces that conserve the momentum. The elastic force simulates the presence of a bond between beads. An ensemble of NVT (number of particles, volume, temperature) is used. A detailed description of the simulation method can be found elsewhere[91]. We used our own implementation of DPD that is 2D parallelized and lightweight[92].

In all simulations, the following parameters were used: $a_{pp} = a_{ss} = 25.0$, $a_{ps} = 26.63$ (soft potential repulsion coefficient), in terms of Flory-Huggins' theory $\chi = 0.5 = 0.306*(a_{ps} - a_{pp})$, where $a_{pp}$—repulsion coefficient between polymer and polymer beads, $a_{ss}$—between solvent and solvent beads, $a_{ps}$—between polymer and solvent beads; $l_0 = 0.5$ (undeformed bond length), $k = 40$ (bond stiffness), $dt = 0.04$ (integration timestep), $\sigma = 3$ (number density), simulation box size $22 \times 22 \times 22$ DPD a.u.

With these parameters, the polymer chain (or chromatin fiber) is able to self-intersect but still has an effective excluded volume. At $\chi = 0.5$, the single polymer chain in a dilute solution has a Gaussian conformation (i.e. it corresponds to a simple random walk).

Each simulation was organized as follows:

The polymer chain is generated as a random walk within the cubic cell with the size of 10 DPD units. Adjacent solvent particles are included into the simulation cell with the size of 22 DPD units until the number density $\sigma = 3$. Additional bonds between beads are added according to the snHi-C contact matrix. If $i$-th and $j$-th beads have a contact, an additional harmonic bond between $i$-th and $j$-th beads is added to the system if $|i - j| > 1$. We define contact as an event when the distance between two beads $(i, j)$ meets criterion $D_{ij} < R_{cut} = 0.7$ Such $R_{cut}$ value corresponds to the average bond length. We count all the contacts in the system. So, in a system any bead can have more than 1 contact. Additional bonds could be overstretched; therefore, the system is equilibrated over 106 steps. The simulation time is two orders of magnitude higher than the necessary equilibration time (Supplementary Fig. 21b); hence, there are no doubts regarding the system equilibrium. According to our calculations, the equilibration time is ~20k steps. The equilibrated system contained overstretched bonds, which were removed one by one until the maximum length became less than the threshold $l_{max} < 1.5$ DPD a.u. (Supplementary Fig. 21c, Supplementary Table 2). Backbone bonds were not removed, because they represented reliable information. The system was equilibrated for 20k steps after each bond removal.

Values of the single-cell Hi-C matrix elements could vary because the restriction fragment is smaller than the selected resolution (10 kb). Data regarding the exact number of contacts between two fragments were not used. Therefore, the contact matrix could be simulated because it is binary. Only the X chromosome was simulated because it is haploid. The X chromosome corresponds to the polymer chain consisting of 2242 beads at 10 kb resolution. Every single chain bead represents 50 nucleosomes. Our model does not consider the shape of a 10-kb region or any other internal properties.

Control simulations were organized in the same manner, but the contacts were shuffled. Shuffling was performed while maintaining the number of contacts at each genomic distance. We also performed simulations with shuffling on the long genomic distances only and sampling the contacts from two cells (Supplementary Table 3). The second case shows that reconstruction of the 3D conformation from diploid chromosomes is meaningless in comparison with haploid chromosomes.

**Coefficient of the difference**. To compare two 3D structures, corresponding distance matrices were calculated. Orientation of the chain in 3D space did not affect the elements of distance matrices. The Coefficient of the difference is introduced as $K = M_{asym}/M_{sym}$, where $M_{asym} = ||D–D'||/2$ and $M_{sym} = ||D–D'||/2$,

where $D$ and $D'$—distance matrices. $||Matrix||$—is the Euclidean distance ($d = \sqrt{a_{11}^2 + a_{12}^2 + .. + a_{21}^2 + \dots}$, $a_{\#\#}$—matrix element). To avoid the contribution of thermal fluctuations, each distance matrix was averaged over 100 conformations with an output rate of 10k steps.

To demonstrate the independence of the final result on the initial conformation, we repeated the calculation of the system ten times with the maximal number of contacts. For each repeat, we created a new independent initial conformation, but we kept the same set of additional bonds. The initial conformation does not affect the final result in the simulation protocol.

**Visualization of epigenetic states**. The visualization was performed using the pymol software v. 2.3.2 (https://pymol.org/2/). 1D epigenetic data were added to the structure as a bead type and represented with a corresponding color. Analysis of different epigenetic states was performed via Python scripts (https://github.com/polly-code/DPD_withRemovingBonds). Before the visualization, some of the conformations were smoothed by averaging coordinates within the window of 15 beads along the chain. This approach ensured that thermal fluctuations were avoided (Supplementary Figs. 16, 21).

**Radial distances and center of mass**. We calculated the surface of the chromosome territory as a convex hull. The distance to the surface was evaluated as the minimal distance from the particle to the surface, and then the distance arrays were averaged.

**Reporting summary**. Further information on research design is available in the Nature Research Reporting Summary linked to this article.

## Data availability

Raw and processed snHi-C and bulk BG3 in situ Hi-C data are available in the GEO NCBI under accession number "GSE131811". List of publicly available GEO sources used in this study: "GSE122603" (Hi-C for Kc167 and BG3 cell lines for comparison of stable TAD boundaries), "GSE58821" (MSL; ChIP-seq), "GSE69013" (RNA-Seq). List of publicly available modENCODE data sources used in this study: total RNA of ML-DmBG3-c2 cell line assessed by RNA tiling array (modENCODE id 713) and the ChIP-chip for MOF (id 3041), BEAF-32 (id 921), Chriz (id 275), CP190 (id 924), CTCF (id 3280), dmTopo-II (id 5058), GAF (id 2651), H1 (id 3299), HP1a (id 2666), HP1b (id 3016), HP1c (id 942), HP2 (id 3026), HP4 (id 4185), ISWI (id 3030), JIL-1 (id 3035), mod(mdg4) (id 324), MRG15 (id 3045), NURF301 (id 5063), Pc (id 325), RNA-polymerase-II (id 950), Su(Hw) (id 951), Su(var)3-7 (id 2671), Su(var)3-9 (id 952), WDS (id 5148), H3 (id 3302), H3K27ac (id 295), H3K27me3 (id 297), H3K36me1 (id 299), H3K36me3 (id 301), H3K4me1 (id 2653), H3K4me3 (id 967), H3K9me2 (id 310), H3K9me3 (id 312), H4K16ac (id 316). dRING binding data were obtained from modENCODE as a ChIP-chip normalized array file (id 927). All other relevant data supporting the key findings of this study are available within the article and its Supplementary Information files or from the corresponding author upon reasonable request. A reporting summary for this Article is available as a Supplementary Information file. Source data are provided with this paper.

## Code availability

The data processing pipeline is available at https://github.com/agalitsyna/sc_dros. The modeling pipeline is available at https://github.com/polly-code/DPD_withRemovingBonds.

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

## Acknowledgements

This work was supported by Russian Science Foundation (RSF) grant #19-14-00016 to S.V.R. Bioinformatics analysis of the data was supported by RSF grant #19-74-00112 to E.E.K and Russian Foundation for Support of Fundamental Science (RFBR) grant #18-29-13013 to S.K.N. A.A.Gal. was supported by RFBR grant #19-34-90136. The research is carried out using the equipment of the shared research facilities of HPC computing resources at Lomonosov Moscow State University and the Makarich HPC cluster provided by the Faculty of Bioengineering and Bioinformatics. The research of P.I.K. is supported partly by RFBR grant #18-29-13041 and by Skoltech Systems Biology Fellowship. The research of A.V.C. is supported by RFBR grant #18-29-13041. S.V.U. and S.V.R. were supported by the Interdisciplinary Scientific and Educational School of Moscow University «Molecular Technologies of the Living Systems and Synthetic Biology». We thank the Center for Precision Genome Editing and Genetic Technologies for Biomedicine, IGB RAS, and IGB RAS facilities supported by the Ministry of Science and Higher Education of the Russian Federation for providing research equipment.

## Author contributions

S.V.R., S.V.U., and I.M.F. conceived the project; D.G. performed cell sorting; V.V.Z. and Y.S.V. prepared snHi-C and bulk BG3 in situ Hi-C libraries; A.A.Gal., K.E.P., E.E.K., S.V.U., A.A.Gav., A.S.G., S.K.N., and M.S.G. analyzed snHi-C, bulk BG3 in situ Hi-C, and publicly available data; P.I.K. and A.V.C. performed polymer simulations; I.M.F. performed FISH; E.A.M. and Y.Y.S. maintained cell cultures; M.D.L. performed sequencing of snHi-C and bulk BG3 in situ Hi-C libraries; S.V.U., V.V.Z., Y.S.V., A.A.Gal., E.E.K., and S.V.R. wrote the manuscript with input from all authors.

## Competing interests

The authors declare no competing interests.
