## [Peer Review File · Nature Communications]

REVIEWER COMMENTS

Reviewer #1 (Remarks to the Author):

Ulianov et al. investigate 3D chromatin organisation in *Drosophila* BG3 cells in single nuclei. They identified that approximately half of the population Hi-C TAD borders are also TAD borders in each single cell, which is more prominent than in mammalian systems (where less than 30% are conserved). The authors label TAD borders that are conserved in more than 50% of the nuclei as stable and borders that are conserved in less than 50% as unstable. One of the main results is that stable borders are enriched in active marks. Furthermore, the authors propose that the hierarchical structure of TADs is likely to be the result of superposition of multiple alternative folding patterns in individual nuclei. I think this is very important and one of the key results of this manuscript. The paper is well written and presents very important results. There are some points that authors would need to address before I could recommend this paper for publication:

Specific points.

1. The separation of all borders present in less than 50% of nuclei as unstable and in more than 50% as stable could impair the clarity of the results. It is difficult to say that something present in 49% of the nuclei is unstable and something present in 51% is stable. I would suggest the authors to split in less than 40% and more than 60% (or less than 30% and more than 70%) to observe stronger differences.
2. The authors find that stable regions are enriched in active marks. Recently, we showed that conserved TAD borders between BG3 and Kc167 (from high resolution Hi-C data) are also enriched in active marks (Dnase-I, Pol II, H3K27ac, H3K4me3) (<https://genome.cshlp.org/content/29/4/613.short>). It would be interesting to look at the overlap between the stable TAD borders the authors identify and the constitutive TAD borders we identified between BG3 and Kc167 cells.
3. We also identified for the first time enrichment of CTCF at TAD borders, but only in BG3 cells (<https://genome.cshlp.org/content/29/4/613.short>). I checked figure S9 in this manuscript and this seems to be case as well for stable, but not unstable borders. This was not discussed at all in the text. I think this is important and should be mentioned by the authors.
4. The DNA polymer simulations are very interesting, but, while I could understand their results, that section is not well explained in the text. I would advice the authors to spend some time improving the clarity of the text for that section.
5. In Figure 2F and S9, the window of heat maps is +/-100Kb while the average TAD size is 60Kb, how can we be sure they do not catch the signal of neighbouring borders.
6. Line 172, how are the cell-specific borders defined? Are these borders that appear only in individual nuclei? This needs to be clarified.
7. One of the claims of the paper is that many of the single cell TADs might be sub TADs in the population HiC. In Figure 3C, the authors should also add sub-TADs borders from bulk Hi-C to compare with single cells TAD borders.

Reviewer #2 (Remarks to the Author):

In this manuscript Ulianov et al., perform single cell HiC and in situ bulk HiC on the *Drosophila* BG3 cell line and characterize genomic topological features such as TAD boundaries, TADs, sub TADs and compartments in both the population and single cells. They also assess the stability and stochasticity of the different structures in the single cells and at variable genomic distances. They also produce polymer simulations to reconstruct the 3D structures of X chromosomes in individual cells and suggest that chromatin folding is best described by the random walk model within TADs and is best approximated by a crumpled globule build of Gaussian blobs at longer distances. Some of the findings presented in this work are interesting in the context of characterizing and understanding the differences between mammalian and *Drosophila* genome architecture and the

molecular mechanisms dictating chromosome folding at the single cell level. However, there are some important points that authors would need to address before I could recommend this paper for publication:

1. The authors state

"we performed an improved single-^[1]_{SEP} 63 nucleus Hi-C (snHi-C)⁷ (Fig. 1a)" but there is no systematic comparison of their protocol with the one cited. The authors should provide a thorough comparison and data supporting this statement.

2. The authors should describe a brief rationale of the downsampling procedure in the main text.

3. The authors state

"86 Similar to the previously published low-resolution (0.1-1 Mbp) snHi-C maps of single 87 mammalian cells^{7,11-14}, the obtained snHi-C maps of Drosophila cells are sparse despite 88 the high number of captured contacts (on average, 33,221 per nucleus; Supplementary 89 Table 1)." However depending on the restriction enzyme used thus the number of fragments produced for a given genome size, and the implemented method, the yield of captured contacts can vary greatly. It would be useful if the authors could justify more solidly why this number of contacts is considered high for a single cell experiment on the Drosophila genome.

4. Supplementary Figure 5 is hard to visualize.

5. In supplementary Figure 7 and in reading the main text one can see that the snHiC data shares 46.6% boundaries with the population HiC and 34.95% of boundaries are shared between the controls (with shuffled contacts) and the population HiC. Between single cells, the percentages of shared boundaries detected by scHiC, is 39.45% while 32.47% are shared between shuffled maps. One would expect that the shuffled maps retain far less boundaries at the same positions than the population HiC and between single cell pairs from scHiC and controls, but the difference is only ~12% and ~7% respectively. How do the authors explain and justify these small differences with a random map?

6. The author's state: "We used the ORBITA^[1]_{SEP}

157 algorithm to reanalyze previously published snHi-C data from murine oocytes⁷ and 158 found that less than 30% of boundaries were shared between any two cells 159 (Supplementary Fig. 8). We conclude that, in Drosophila, TADs have more stable 160 borders as compared to mammals. This corroborates the recent observations of Cavalli 161 lab²⁰ and may reflect differential impact of loop extrusion process²¹⁻²³ and 162 internucleosomal contacts¹⁶ on TAD formation^{24,25}.

How meaningful is a 10% difference regarding shared TADs between Drosophila single cells and mammalian single cells? The authors would benefit from contrasting their data with additional single cell HiC data available from mammals and confirm that indeed this 10% is reproducible and argue why 10% more stability is considered important.

7. The authors describe

"163 Population TADs in Drosophila mostly correspond to inactive chromatin, whereas 164 their boundaries and inter-TAD regions correlate with highly acetylated active chromatin^[1]_{SEP} 165 at 20 kb resolution^{16,26}." However this statement is controversial and the authors only cite two previous works from their own group. If most TADs are formed by repressed chromatin is arguable and there are evidences suggesting otherwise both in cell lines and embryos (Ramirez et al., 2015, Rowley et al., 2017, Hug et al., 2017, Wang et al., 2018, Keerthi T et al., 2019, Arzate-Mejia et al., 2020). Thus the authors need to discuss their observations and their findings giving plausible explanations on the differences arising from their data and what has been reported

before.

8. The random set used in figure 2F seems more enriched in open chromatin marks than the unstable boundaries, which is unexpected. The authors should clarify how the random set was selected and discuss this result.

9. In line 174 the authors refer to Figure 2e, which shows a different result.

10. Supplementary figure 9 is hard to visualize as the plots are small.

11. The authors state "179 Drosophila TADs are remarkably hierarchical in a cell population^{26,27}"
What does remarkably means in this context and compared to what?

12. For the TAD segmentation analysis the authors should specify at which resolution they are doing their analysis

13. The authors state "184 TADs, Fig. 3a). We analyzed only the haploid X chromosome to avoid combined folding

185 patterns of diploid somatic chromosomes." BG3 cells are reported to be from diploid to tetraploid (<http://flybase.org/reports/FBtc0000068.html>). The authors should provide cytological evidence confirming their cell line batch is diploid.

14. "187 with well-defined boundaries aroused from specific folding of the chromatin. To probe this, we tested the resistance of sub-TAD boundaries to the data downsampling (2-fold." Line 187 should say prove not probe

15. Figure 3b is confusing as the y-axis writes percentage of subTAD boundaries but both TAD and subTAD boundaries are presented in the graph.

16. The authors state "Similarly to Drosophila S2 cells³⁰ and 220 contrary to early embryo nuclei², we observed an increased interaction frequency only 221 between active regions in the bulk BG3 in situ Hi-C data (i.e. we confirmed the presence 222 of the A-compartment in the BG3 cell population) (Fig. 3d, Supplementary Fig. 11)"
However this observation is contradictory with their findings that most inner TAD bins and unstable boundaries are enriched in heterochromatin features. Wouldn't they expect a high frequency of contacts supporting their B to B interactions if most TADs fall into the B compartment? It is also contradictory with the long range interactions mediated by Polycomb (described below). Also there are other reports in the literature in which S2, and BG3 HiC has been performed and this observation is not fully supported (Wang et al., 2018, Keerthi T et al., 2019, Arzate-Mejia et al., 2020). The authors need to discuss this in more detail.

17. The authors describe "We next applied dissipative particle dynamics (DPD) polymer simulations³⁵ to 237 reconstruct the 3D structures of haploid X chromosomes in individual cells from the 238 snHi-C data (Fig. 4a, Supplementary Fig. 12)".
As stated before the authors need to check if their BG3 cell line batch has a diploid karyotype, otherwise the model would arise from two molecules and this would have to be taken into consideration. The authors have performed FISH (Supplementary figure 15C). They should present several examples of the images analysed to confirm the quality of their signals and that they are present in just one molecule confirming the haploidy of the modelled X chromosome.

19. The authors describe "Due to the fact that TADs in 269 Drosophila are largely composed of inactive chromatin, we propose that the chromatin 270 fiber conformation within TADs is mostly determined by interactions between adjacent 271 non-acetylated nucleosomes. In contrast, at large genomic distances, TADs interact

272 with each other in a stochastic manner, imposing the ellipsoidal form of the CT that is ^[11]_[SEP]
273 observed in all model structures (Fig. 4a, Supplementary Fig. 12).” How do the authors
reconcile the high variability they report at the subTAD level with the more stable behaviour shown
in their modelling? Also how do they reconcile the high variability at the subTAD structures with
their proposal of fiber conformation within TADs is mostly determined by interactions between
adjacent non-acetylated nucleosomes? ^[11]_[SEP]

20. The authors state: “We found that Polycomb-occupied regions interacted with each other in a
cell- 288 specific manner and, moreover, such contacts occurred even between loci regardless of
^[11]_[SEP]
289 the genomic distances between them (Fig. 4g, upper panels)” Do the authors mapped
Polycomb occupied regions to the B compartment? How does this correspond with the B
compartment not being supported by interactions? The authors need to explain this discrepancy

Reviewer #3 (Remarks to the Author):

The paper is interesting and provide novel insight into the folding mechanisms in eukaryotes. I've
found particularly original the experimental approach of considering single-cell HiC.

Unfortunately, my expertise is more on the theoretical side and I can give little advise if the paper
can be improved further on the experimental one. Yet, it is my impression that the work is sound
and well presented in all its main aspects, even the most technical ones.

The simulation work which accompanies the paper looks solid, so I have no particular suggestion
related to it either.

To summarize, my judgement about the work is full positive and I recommend its publication in
Nature Comm. without further hesitation.

Reviewer #4 (Remarks to the Author):

In this work, the authors characterize the chromatin structure in individual *Drosophila* nuclei using
Hi-C and polymer modeling. They applied a recently developed single-nucleus biochemical Hi-C
(snHi-C) assay (2017 Nature) together with a more effective strategy to filter out likely inaccurate
data (called ORBITA here) to ultimately describe the genomic structure of 20 individual cells
ostensibly at 10 kb resolution. They find that a very large percentage (> 40%) of TAD borders are
the same between different cells, which is strikingly different from the lack of shared borders in
single mammalian oocytes (in the 2017 Nature). This difference is indeed a significant and well
supported finding, though less so the magnitude, as explained below. The authors then build
models of the X chromosome of each cell based on the Hi-C data, from which they infer details of
the folding behavior at different length scales as well as heterogeneity in the structures that are
attributed to the stochasticity in biological processes in the different cells.

Overall, this is an impressive and interesting work that is very well analyzed. However, there are
some major concerns that should be addressed before consideration for publication.

1. It is not clear that the authors could accurately describe this work as a genome-wide
characterization of the chromatin structure with 10 kb resolution. A common definition for
“resolution” of population-level Hi-C maps is that 80% of the bins contain at least 1000 reads (Rao
et al, Cell 1665 (2014)). While this is undoubtedly beyond the capability right now for single cell

Hi-C, it is not clear what justification the authors used to settle on 10 kb. It would appear as though that many (about half) of the cells in this work have only 1 or 2 reads (on average) for each 10 kb bin, which is very low. In the previous application of this method to mammalian cells (2017 Nature), which I believe described data with roughly the same number of reads per genome length as in the present work, the maps were analyzed at 40 kb resolution (Fig 2c in that paper). For each cell, what percentage of the genome are bins with no reads? With at least some cells (such as Cell 4 in Fig 2a), it appears as though there could be ~40% of the bins with no reads. So, describing this characterization as "genome-wide" might be inaccurate, if, strictly speaking, there is a very large portion of the genome with no data. Perhaps there are extended regions of the genome for which there is sufficient data in many cells to make sound conclusions, but other regions of the genome for which there is insufficient data. Alternately, perhaps 20 kb or 25 kb resolution could be justified in some legitimate way (such as, for example, two reads per bin in 80% of the genome in the majority of cells) and still be small enough to identify the 80-100 kb TADs.

2. Much of the description of the heterogeneity in structures stems from an analysis of the models. However, from the analysis shown in Suppl 15a,b, most of these structures disagree significantly with the Hi-C data. Nine of the 20 have an FNR > 0.5, which is somewhat shocking as this indicates that more than 1/2 of the contacts that are present in the Hi-C data are not present in these models. And this is with sparse Hi-C data. An FNR cutoff of 0.2, reflecting the absence of 20% of the Hi-C contacts, is satisfied by only 4 of the 20. For the TPR, an equally surprising 16 of the 20 are below a value of 0.5, which indicates that less than 1/2 of the contacts that are present in these models are also present in the Hi-C data. The authors provide FISH data that agrees with the models, but it could be argued that most of the structures do not adequately agree with the Hi-C data. With so poor FNR and TPR in the majority of structures, by what criteria do the authors conclude that these structures are consistent with the Hi-C data? It is also unnerving that those cells with the highest number of contacts (Cells 1 to 3) are associated with the least consistent models and those with the fewest number of contacts (Cells 15 to 20) are associated with the most consistent. It should be that more data leads to more reliable models, not less reliable models. During the modeling, I believe that the authors effectively down-sampled their data by 1/3 to 1/2 to remove overly stretched bonds. Are the loci pairs whose bond was removed during this down-sampling in close proximity in the final structures to possibly contact? Judging by the FNR and TPR, I am not optimistic, but this would provide some additional evidence that the models are sufficiently consistent with the Hi-C data to warrant more detailed analysis.

3. On the same subject, how can the authors be sure that some, maybe much, of the heterogeneity observed in the Hi-C data is not owing to the low (random) sampling during the biochemical assays? Two similar structures whose contacts are sparsely (randomly) sampled might only appear to be different. It might also be noted that while Fig 2a indeed shows that the distributions of these two cells are non-random, this of course does not mean that all of the Hi-C data is non-random.

4. The conclusion that there is greater conservation of the TAD boundaries between individual *Drosophila* cells than between the individual mammalian oocytes is supported by the analysis presented in this work and it is a genuinely significant result. However, that it is "over 40% of TAD boundaries" that are conserved between the cells is not well supported here since from "randomly shuffled maps", the same analysis shows that over 32% of TAD boundaries are conserved. Shouldn't the "random" data give essentially no shared borders, almost by the definition of "random"? Perhaps the TAD caller would be expected to identify a few TADs in noisy, sparse data, but why should the borders be at the same place in so many cells? Does this point to an inaccuracy in the TAD caller or calling method used with data that is so sparse? There are other TAD callers (for example TopDom, Zufferey et al, Genome Biology 19, 217 (2018)) or different values of the gamma parameter that could be examined to yield essentially no shared borders in the random data that, when obtained, could then be used to more correctly estimate the % of shared borders in the snHi-C data.

More minor concerns:

5. At first glance, I expected that the data presented in Fig 2a was an analysis of the whole genome for these two cells. However, I believe that this is not correct, as I believe that there is only about 20 Mb of the genome covered in the analysis for Cell 4 and about 5 Mb for Cell 6 (with a reference genome total length of about 130 Mb). Why was only such a small portion of the genome analyzed, especially for Cell 6 (4%)? It would be better to see this analysis from a more substantial portion of the genome, if not the full genome. But if this is computationally too demanding, at least some discussion for the reasons for choosing the regions that were analyzed should be included.

6. It is difficult to judge whether the TADs called in the individual cells in Fig 2b are in fact obvious in the figure since the black lines overlap and somewhat obscure the data. Perhaps the authors could depict the TADs as bars underneath the maps as they did in Fig 1 of their 2016 Genome Research paper.

7. The models were found to have the active chromatin within the CT interior and the inactive regions on the CT surface. By contrast, in mammalian cells, active transcription occurs on the surface of CTs (Shah et al, Cell 174, 363 (2018)). In the population-level Hi-C data of these cells, are there more inter-chromosome contacts between the A compartments or between the B compartments? If there are more contacts between the A compartments, particularly with the X chromosome, this would conflict with the models. Also, some description of how "contact" is defined in the analysis of the models to generate the corresponding Hi-C maps should be given in the Methods.

8. Were the TADs called in the oocyte data at 40 kb resolution or 10 kb resolution? Was a similar strategy used to define the gamma as with the Drosophila cells? After ORBITA, how many contacts per cell are there? Perhaps a few words in the Methods could be included.

9. Finally, line 153 in the text indicates that 40.5% of boundaries were shared between cells but the number in the legend to Suppl Fig 7 is 39.45%. Also it is the "-log₁₀ values that are shown in Suppl Fig 7b. I believe that the legend to Suppl Fig 2b and 2c refers to the data that is shown in Suppl Fig 2c and 2b, respectively. In the legend to Suppl Fig 15a and b, the definition of "false negatives" is literally the same as that of "true positives". Which probe set is shown in Suppl 15d? And it might be easier to appreciate the "Coverage" in Suppl Fig 5 if it was converted to percentage as in Fig 2c.

Reviewer #1 (Remarks to the Author):

Ulianov et al. investigate 3D chromatin organisation in *Drosophila* BG3 cells in single nuclei. They identified that approximately half of the population Hi-C TAD borders are also TAD borders in each single cell, which is more prominent than in mammalian systems (where less than 30% are conserved). The authors label TAD borders that are conserved in more than 50% of the nuclei as stable and borders that are conserved in less than 50% as unstable. One of the main results is that stable borders are enriched in active marks. Furthermore, the authors propose that the hierarchical structure of TADs is likely to be the result of superposition of multiple alternative folding patterns in individual nuclei. I think this is very important and one of the key results of this manuscript. The paper is well written and presents very important results. There are some points that authors would need to address before I could recommend this paper for publication:

Specific points.

1. The separation of all borders present in less than 50% of nuclei as unstable and in more than 50% as stable could impair the clarity of the results. It is difficult to say that something present in 49% of the nuclei is unstable and something present in 51% is stable. I would suggest the authors to split in less than 40% and more than 60% (or less than 30% and more than 70%) to observe stronger differences.

Reply:

We followed the recommendation and repeated our analysis with more stringent thresholds for stable and unstable boundaries (<40% and >60%, respectively; see **Additional Figure 1**). We observed neither a substantially stronger quantitative nor any additional qualitative difference between the boundary types; thus, we preferred to retain our initial (<50% and >50%) thresholds in the revised version of the text.

Additional Figure 1

2. The authors find that stable regions are enriched in active marks. Recently, we showed that conserved TAD borders between BG3 and Kc167 (from high resolution Hi-C data) are also enriched in active marks (Dnase-I, Pol II, H3K27ac, H3K4me3) (<https://genome.cshlp.org/content/29/4/613.short>). It would be interesting to look at the overlap between the stable TAD borders the authors identify and the constitutive TAD borders we identified between BG3 and Kc167 cells.

Reply:

We thank the reviewer for this suggestion. We retrieved the conserved set of boundaries between BG3 and Kc167 from Chathoth and Zabet (2019), as suggested. However, this set of boundaries was available for the dm6 genome and had a resolution different from our analysis. To make the boundary set comparable, we mapped the coordinates of the Chathoth and Zabet boundaries from the dm6 genome to dm3 with *liftover* and coarse-grained them at 10-kb resolution. This approach resulted in 251 genomic bins assigned as conserved boundaries. We next compared these boundaries with the stable boundaries obtained in our study (5196 genomic bins in total) and observed an overlap of 183 (72.9% of conserved boundaries). For this comparison, we applied the approach used throughout our paper: we defined stable boundaries as being present in more than 50% of individual cells and compared the boundaries, allowing an offset of 10 kb.

In addition, we followed a more stringent strategy, as proposed by the reviewer, and selected the boundaries present in more than 70% of individual cells (2969 boundaries). This resulted in a total overlap of 177 boundaries with the conserved set (70.5% of the conserved boundaries). Hence, we conclude that a major fraction of the BG3/Kc167 shared boundaries overlap with the stable boundaries from our study (approximately 70% depending on the definition of stable boundaries and the offset). This overlap confirms our observation that stable boundaries in individual cells have the properties of the population boundaries and contain a large fraction of boundaries conserved between cell lines.

However, a significant fraction of conserved boundaries did not intersect with the boundaries found to be stable in our analysis; this result might be due to differences in exact positions of TAD boundaries in the BG3 and Kc167 cell lines. We are also aware that some boundaries detected in Chathoth and Zabet (2019) at fragment size resolution cannot be detected in our analysis at 10-kb resolution. Given these limitations, we find the observation of overlap between the two sets of boundaries significant and important. Therefore, we modified the text on page 7 accordingly: “*The boundaries present in a large fraction of cells (more than 50% of cells) defined here as “stable” overlapped 73% of conserved boundaries between BG3 and Kc167 cell lines and had high levels of active chromatin marks (RNA polymerase II, H3K4me3; Fig. 3f, Supplementary Fig. 11). They were also slightly enriched in some architectural proteins associated with active promoters (BEAF-32, Chriz, CTCF and GAF; Supplementary Fig. 11)*”.

We also added the description of this analysis in the Methods section: “*We compared stable boundaries with boundaries conserved between Kc167 and BG3 cells. For that, we obtained TAD positions from, mapped them to the dm3 genome with liftover, and coarse-grained the coordinates to 10-kb bins. We then allowed the 10-kb offset and counted the boundaries that overlapped with stable boundaries obtained in the single-cell analysis*”.

3. We also identified for the first time enrichment of CTCF at TAD borders, but only in BG3 cells (<https://genome.cshlp.org/content/29/4/613.short>). I checked figure S9 in this

manuscript and this seems to be case as well for stable, but not unstable borders. This was not discussed at all in the text. I think this is important and should be mentioned by the authors.

Reply:

We thank the reviewer for this observation and we now highlight this fact in the Results section on page 7: “*They were also slightly enriched in some architectural proteins associated with active promoters (BEAF-32, Chriz, CTCF and GAF; Supplementary Fig. 11)*”.

4. The DNA polymer simulations are very interesting, but, while I could understand their results, that section is not well explained in the text. I would advice the authors to spend some time improving the clarity of the text for that section.

Reply:

Following the reviewer’s recommendation, we have rewritten and extended this part of the Results section (pages 9-11)

5. In Figure 2F and S9, the window of heat maps is +/-100Kb while the average TAD size is 60Kb, how can we be sure they do not catch the signal of neighbouring borders.

Reply:

The Z-scored curve centered at the boundaries of a specific type indeed catches signals from other boundaries located in a 100-kb window. Nevertheless, (i) the overwhelming majority of signals are from the interior of TADs, and (ii) these neighboring boundaries are located at different distances from the target boundary; hence, their signals do not contribute significantly to the resulting curve. We would like to note that Fig. 2f is now Fig. 3f, and Supplementary Fig. 9 is now Supplementary Fig. 11.

6. Line 172, how are the cell-specific borders defined? Are these borders that appear only in individual nuclei? This needs to be clarified.

Reply:

Cell-specific boundaries are defined as boundaries identified in just one cell among all cells analyzed. We clarified this in the revised version of the text. The text on page 8 was modified as follows: “...as well as boundaries identified in just one cell termed cell-specific boundaries)...”.

7. One of the claims of the paper is that many of the single cell TADs might be sub TADs in the population HiC. In Figure 3C, the authors should also add sub-TADs borders from bulk Hi-C to compare with single cells TAD borders.

Reply:

We altered the figure according to the reviewer’s recommendations. We would like to note that Fig. 3c is now Fig. 4c.

Additional corrections:

We noticed a technical problem with Supplementary Figures 3a and 9 regarding our re-processing of Flyamer and co-workers' (2017) snHi-C data. This problem affected our reported results for several cells in these figures. In particular, the ordering and selection of top-40 cells in Supplementary Figure 9 was wrong, as well as descriptive

statistics in Supplementary Figure 3a. This error is now corrected. We also restricted our analysis to the top-20 cells of Flyamer et al. (2017) for a better comparison with our 20 cells and the top-30 cells from Gassler et al. (2017).

We also noticed an error in the annotation of Supplementary Fig. 6c, where we reported a larger number of iterations of subsampling than we actually used for the estimation of the boundary's robustness. This approach led to an underestimation of parameters for the optimal boundary refinement strategy and the number of recoverable boundaries from the subsampling procedure described in the "Robustness of TAD calling" section of Online Methods. We have now increased this number of iterations to ten. We improved Supplementary Fig. 6c, 6d, and their legends, and corrected the Online Methods. These modifications are minor and do not affect the conclusions based on a smaller number of iterations.

We also improved Supplementary Fig. 9 by removing redundant elements of the plot and increasing the image resolution.

Reviewer #2 (Remarks to the Author):

In this manuscript Ulianov et al., perform single cell HiC and in situ bulk HiC on the *Drosophila* BG3 cell line and characterize genomic topological features such as TAD boundaries, TADs, sub TADs and compartments in both the population and single cells. They also assess the stability and stochasticity of the different structures in the single cells and at variable genomic distances. They also produce polymer simulations to reconstruct the 3D structures of X chromosomes in individual cells and suggest that chromatin folding is best described by the random walk model within TADs and is best approximated by a crumpled globule build of Gaussian blobs at longer distances. Some of the findings presented in this work are interesting in the context of characterizing and understanding the differences between mammalian and *Drosophila* genome architecture and the molecular mechanisms dictating chromosome folding at the single cell level. However, there are some important points that authors would need to address before I could recommend this paper for publication:

1. The authors state: "We performed an improved single-nucleus Hi-C (snHi-C) (Fig. 1a)" but there is no systematic comparison of their protocol with the one cited. The authors should provide a thorough comparison and data supporting this statement.

Reply:

We wanted to say that we used the snHi-C protocol previously published by us, which was improved as compared to other protocols published to date in terms of contact recovery. In the revised version of the text, this sentence is rephrased as follows: "*To investigate the nature of TADs in single cells and to characterize individual cell variability in Drosophila 3D genome organization, we performed single-nucleus Hi-C (snHi-C) (Fig. 1a)...*".

2. The authors should describe a brief rationale of the downsampling procedure in the main text.

Reply:

We downsampled the snHi-C data at several steps of the analysis for different reasons: (i). We used subsampling to select the libraries suitable for deep sequencing.

(ii). We downsampled the snHi-C data to show that the identified boundaries were resistant to the depletion of the number of contacts in the map. In this way, the identified boundary positions do not represent fluctuations in sparse data (experimental noise), and the contact profile is robust.

(iii). Downsampling of the snHi-C data was used to compare the resistance of sub-TAD and TAD boundaries to contact depletion. In this case, we followed the same logic: if sub-TAD boundaries did not emerge from stochastic contact profile fluctuations, they should be stable after removing a significant fraction of contacts.

We added brief rationales for the data downsampling to all relevant parts of the revised MS.

3. The authors state: “*Similar to the previously published low-resolution (0.1-1 Mbp) snHi-C maps of single mammalian cells, the obtained snHi-C maps of Drosophila cells are sparse despite the high number of captured contacts (on average, 33,221 per nucleus; Supplementary Table 1).*”

However depending on the restriction enzyme used thus the number of fragments produced for a given genome size, and the implemented method, the yield of captured contacts can vary greatly. It would be useful if the authors could justify more solidly why this number of contacts is considered high for a single cell experiment on the *Drosophila* genome.

Reply:

We thank the reviewer for this important remark. Indeed, initially, we have not described the reasoning behind the claim of a high number of captured contacts. We have now reformulated and substantially revised this part. The following text was inserted on page 5: “*To estimate the overall quality of the snHi-C libraries, we first calculated the number of captured contacts per cell. On average, we extracted 33,291 unique contacts from individual nuclei that represented 5% of the theoretical maximum number of contacts and corresponded to four contacts per 10-kb genomic bin (see Methods); in the best cell, 17% of contacts were recovered (Fig. 2a, b, Supplementary Table 1). Relying on the number of captured contacts, we then estimated the proportion of the genome available for the downstream analysis. At 10-kb resolution, ~82% of the genome on average was covered with contacts in each individual cell, and 67% of genomic bins established more than 1 contact (Fig. 2c). Notably, in the previously published mouse snHi-C datasets, ~0.6% of theoretically possible contacts were detected on average (Fig. 2b). Because the top-20 mouse snHi-C libraries from Flyamer et al. demonstrated a comparable genome coverage with contacts and a number of contacts per 10-kb genomic bin (Fig. 2d), we could directly compare the Drosophila and mouse snHi-C maps (see below)*”

We also added Fig.2a-c, and the section “Assessment of the percentage of recovered contacts” in the Methods section to clarify this point. We briefly describe our reasoning and improvements to the text below.

The largest number of contacts per nucleus is currently reported for snHi-C in Flyamer et al. (2017), reviewed by Ulyanov et al. (2017). Here, we used the same protocol with a 4-cutter restriction endonuclease DpnII. However, the *Drosophila* genome is ~10 times smaller than the mouse genome, and this affects the number of restriction fragments potentially involved in the contact formation. To compare snHi-C datasets across species, we assessed the percentage of recovered contacts out of all possible contacts per nucleus. First, we determined the theoretical size of the pool of restriction fragments for the nucleus of each species and cell type. For *Drosophila*, we used a diploid male

cell line. Thus, the total number of restriction fragments is ~600,000, composed of the double amount of fragments in autosomes ($2 \times 265,167$, as assessed by the *dm3 in silico* digestion) plus the number of fragments in chromosome X (64,108). For mouse, Flyamer et al. (2017) analyzed oocytes with four copies of the genome, resulting in a total of $4 \times 6,407,802 \sim 25.6$ mln fragments.

We reanalyzed the single-cell Hi-C dataset on mouse G2 zygotes' pronuclei from Gassler et al. (2017). In G2 zygotes' pronuclei, the copy number of the genome is two, and the number of restriction fragments is ~12.8 mln fragments (we did not distinguish between the maternal and paternal pronuclei because the contribution of chromosome X is not as significant for the mouse genome).

We next assessed the upper limit of the total number of possible contacts per single nucleus, which is achieved when each restriction fragment forms two contacts with the ends of any other restriction fragments from the pool. Because the valency of each fragment is two, the theoretical upper limit is equal to the number of restriction fragments. We then divided the total number of observed contacts (recovered by ORBITA) by the upper bound of the possible number of contacts. As a result, for *Drosophila*, we recovered up to ~17% of the total number of possible contacts (see new Fig. 2b); this number is approximately 2.6% for the best mouse dataset. The mean percentage of recovered contacts is 4.9% for our dataset and <1% for Flyamer et al. (2017) and Gassler et al. (2017). Thus, it can be concluded that we indeed have recovered a significantly higher fraction of contacts per individual nucleus compared to the previous studies.

This assessment of the percentage of recovered contacts may be inexact due to several reasons: (1) we did not perform sorting prior to snHi-C to isolate G1 cells; therefore, some regions of the genome might have an increased copy number after replication; (2) some genome regions might be affected by deletions and copy number variations that were not accounted for in our analysis. However, even in the worst-case scenario, when the *Drosophila* genome is doubled in the S phase of the cell cycle, we recovered at least than 8% of all possible contacts for the best cells in our analysis, which is still much better than the best cells from mammalian sources.

4. Supplementary Figure 5 is hard to visualize.

Reply:

In this figure, we aimed to demonstrate that the dependency between the gamma value and parameters of the TAD profile is highly concordant both in individual cells and between individual chromosomes. To show this clearly, we plotted all chromosomes from all cells separately with an image quality suitable for visual examination of each panel of the figure. All details are visible upon zooming in any appropriate PDF viewer. Hence, we believe that Supplementary Fig. 5 does not require correction.

5. In supplementary Figure 7 and in reading the main text one can see that the snHiC data shares 46.6% boundaries with the population HiC and 34.95% of boundaries are shared between the controls (with shuffled contacts) and the population HiC. Between single cells, the percentages of shared boundaries detected by scHiC, is 39.45% while 32.47% are shared between shuffled maps. One would expect than the shuffled maps retain far less boundaries at the same positions than the population HiC and between single cell pairs from scHiC and controls, but the difference is only ~12% and ~7% respectively. How the authors explain and justify these small differences with a random map?

Reply:

We thank the reviewer for this remark. The mean number of TAD boundaries detectable in individual cells is 1,460 out of the total genome size of 11,901 bins (10-kb bin size, chromosome X and all autosomes except for chromosome 4). Thus, approximately every eighth bin of the genome is annotated as a TAD boundary. Two random sets with these properties are expected to have ~12.3% of shared boundaries. However, in our analysis, we allow for one bin offset of the one set of boundaries, which would reduce the effective size of the genome and increase the expected percentage. However, this scenario does not account for the restrictions on the TAD/inter-TAD sizes. Hence, we used two types of controls.

Firstly, we called TADs on randomized snHi-C maps. On average, the percentage of shared boundaries was 32.9% between these maps, ~7% lower than that for real maps (Fig. 3d). This result is significant and sufficiently substantial for demonstrating that the observed boundaries are not random.

The second control was shuffling of the positions of TADs/inter-TADs so that the size distributions were preserved. This approach resulted in the percentage of shared boundaries of 33.1% on average (~6% smaller than the observed mean). This procedure was used to calculate the significance of the observed percentages in Supplementary Figures 8b, 8e, 9b. Notably, most of the percentages of shared boundaries between the real cells were significantly larger than expected at the 0.01 confidence level, but this was not so for the shuffles (Supplementary Fig. 8b).

Therefore, it can be concluded that the observed percentages are indeed significant and are higher than expected. We, therefore, changed the text on page 7 as follows:

“This is significantly higher than the percentage of shared boundaries for shuffled control maps (32.9%) and the percentage expected at random (33.1%, Fig. 3d). Notably, 42% of NBT-identified single-cell TAD boundaries were conserved in pairwise cell-to-cell comparisons (Supplementary Fig. 7b), supporting the results obtained in the analysis of modularity-derived TAD boundary profiles.”

We also added the percentages of shared boundaries in real data and two types of controls to the main text, Fig. 3d.

6. The author's state: *“We used the ORBITA algorithm to reanalyze previously published snHi-C data from murine oocytes and found that less than 30% of boundaries were shared between any two cells (Supplementary Fig. 8). We conclude that, in Drosophila, TADs have more stable borders as compared to mammals. This corroborates the recent observations of Cavalli lab and may reflect differential impact of loop extrusion process and internucleosomal contacts on TAD formation”.*

How meaningful is a 10% difference regarding shared TADs between Drosophila single cells and mammalian single cells? The authors would benefit from contrasting their data with additional single cell HiC data available from mammals and confirm that indeed this 10% is reproducible and argument why 10% more stability is considered important.

Reply:

We reanalyzed the data from Gassler et al. (2017) (GSE100569) with ORBITA and selected the top-20 (by the number of contacts) wild-type G2 zygotes. We then called TADs and calculated the percentage of shared boundaries, as described for the *Drosophila* dataset, for single nuclei in Flyamer et al. (2017). We observed that the number of shared boundaries between single G2 zygotes in mice is even lower than that for oocytes. The mean percentage of shared boundaries between cells is 21% for

this dataset. This result is significantly lower than the percentage of shared boundaries between individual *Drosophila* cells.

We added this result to the main text: “We used the ORBITA algorithm to reanalyze previously published snHi-C data from murine oocytes and G2 zygote pronuclei and found that 31.2% and 21% of boundaries were shared on average between any two cells, respectively (Fig. 3e, Supplementary Fig. 9)”. This analysis of Gassler and colleagues' (2017) dataset is now reflected in Fig. 2a-d, Fig. 3e and Supplementary Fig. 9.

We also demonstrated that the percentage of shared boundaries was significantly larger for *Drosophila* at 10-kb and 40-kb resolution as compared to both mouse datasets (Fig. 3e and Supplementary Fig. 10b, respectively). However, we noticed that the percentage of shared boundaries might depend on the data quality. Hence, we calculated the mean number of contacts per genomic bin for each pair of snHi-C maps and compared percentages of shared boundaries at comparable levels of data quality (Supplementary Fig. 10a,c). We observed that the percentage of shared boundaries was 5–10% larger for *Drosophila* cells. It was, therefore, concluded that our result was reproducible for different datasets and was robust with respect to the data resolution and quality.

To reflect this, we added Supplementary Fig. 10 and the following text: “This result is reproduced at 40-kb resolution and persists for a broad range of snHi-C datasets' quality (Supplementary Fig. 10).”

7. The authors describe: “Population TADs in *Drosophila* mostly correspond to inactive chromatin, whereas their boundaries and inter-TAD regions correlate with highly acetylated active chromatin at 20 kb resolution.”

However this statement is controversial and the authors only cite two previous works from their own group. If most TADs are formed by repressed chromatin is arguable and there are evidences suggesting otherwise both in cell lines and embryos (Ramirez et al., 2015, Rowley et al., 2017, Hug et al., 2017, Wang et al., 2018, Keerthi T et al., 2019, Arzate-Mejia et al., 2020). Thus the authors need to discuss their observations and their findings giving plausible explanations on the differences arising from their data and what has been reported before.

Reply:

Several studies mentioned by the reviewer were performed at the ultra-high resolution of Hi-C maps (up to 200 bp). Their results suggest that active chromatin is organized in TADs as well. In contrast to large TADs bearing repressed regions, these “active” TADs are typically small (according to Wang et al. 2018 NatComm, 9 kb in length) and, thus, could not be analyzed at the 10–20 kb resolution. Due to the limited number of recovered contacts, we were unable to build the snHi-C maps at sub-kb resolution to consider active TADs in the analysis.

However, all analyzed epigenetic and other properties of TADs are relevant at a medium Hi-C map resolution (10-20 kb). To reflect this, we reformulated the sentence in question: “Population TADs in *Drosophila* identified at 10-20 kb resolution mostly correspond to inactive chromatin, whereas their boundaries and inter-TAD regions correlate with highly acetylated active chromatin. These are further partitioned into much smaller domains with the size of about 9 kb and, thus, unavailable for the analysis at the resolution of our Hi-C maps”.

8. The random set used in figure 2F seems more enriched in open chromatin marks than the unstable boundaries, which is unexpected. The authors should clarify how the random set was selected and discuss this result.

Reply:

According to Fig. 3f, a random set is neither enriched in nor depleted of any epigenetic marks (included active ones). This is characterized by flat Z-curves indicating that the dataset is actually composed of bins randomly selected across the entire genome without precedence of active or repressed regions. Thus, the random set has epigenetic properties of an “averaged” genomic bin which, clearly, is not enriched on or depleted of any chromatin marks. In contrast, unstable boundaries are depleted of active marks as compared to the random set. This observation is now highlighted in the Results section on page 8 as follows: *“The epigenetic profiles of “unstable” boundaries may be due to the fact that actual profiles of active chromatin in individual cells differ from the bulk epigenetic profiles used in our analysis. However, it may also reflect a certain degree of stochasticity in chromatin fiber folding into contact domains. Taking into consideration the fact that active chromatin regions mostly colocalize with stable boundaries, one would expect the “unstable” boundaries tend to be located in the inactive parts of the chromosome”*.

We also modified the Discussion section on page 12: *“In contrast to stable TAD boundaries, the boundaries that demonstrate cell-to-cell variability bear silent chromatin. Some cell-specific TAD boundaries may originate at various positions due to a putative size limit of large inactive TADs or other restrictions in chromatin fiber folding. Indeed, it appears that the assembly of randomly distributed TAD-sized self-interacting domains is an intrinsic property of chromatin fiber folding. In mammals, the positioning of these domains is modulated by cohesin-mediated DNA loop extrusion, whereas in Drosophila, it may be modulated by segregation of chromatin domains bearing distinct epigenetic marks. Even if cell-specific and unstable TAD boundaries are distributed in a random fashion, they should be depleted in active chromatin marks because active chromatin regions are mainly occupied by stable TAD boundaries. We also cannot exclude that variable boundaries and the TAD boundary shifts are caused by local variations in gene expression and active chromatin profiles in individual cells that we cannot assess simultaneously with constructing snHi-C maps.”*

We described how the random set was obtained in the Methods section of the revised MS on page 51: *“To obtain randomized boundaries, we shuffled bulk in situ Hi-C boundaries across the Drosophila genome, preserving the number of boundaries per chromosome”*.

We also would like to note that Fig. 2f is now Fig. 3f.

9. In line 174 the authors refer to Figure 2e, which shows a different result.

Reply:

We now refer to the correct version of the Figure.

We would like to note that we have removed the old Fig.2e as non-informative.

10. Supplementary figure 9 is hard to visualize as the plots are small.

Reply:

In this figure, we aimed to present a comprehensive view of the epigenetic properties of different types of TAD boundaries. In the revised version of the MS, we used a more readable font and increased the resolution of the images. All details are now visible upon zooming in any appropriate PDF viewer. Thus, we believe that Supplementary Fig. 11 (9 in the initial version) is now suitable for the visual examination of each panel.

11. The authors state *“Drosophila TADs are remarkably hierarchical in a cell population.”* What does remarkably means in this context and compared to what?

Reply:

We reformulated this sentence as follows: “*Drosophila TADs are hierarchical in cell population-based Hi-C maps*”.

12. For the TAD segmentation analysis the authors should specify at which resolution they are doing their analysis.

Reply:

TAD segmentation was performed at 10-kb resolution. In the revised text (page 6), we added the following information: “*For each nucleus, we performed TAD segmentation in snHi-C maps of 10-kb resolution at a broad range of the gamma (γ) master parameter values (Fig. 3b, see Methods and Supplementary Fig. 5)*”.

13. The authors state “We analyzed only the haploid X chromosome to avoid combined folding patterns of diploid somatic chromosomes.” BG3 cells are reported to be from diploid to tetraploid (<http://flybase.org/reports/FBtc0000068.html>). The authors should provide cytological evidence confirming their cell line batch is diploid.

Reply:

The ML-DmBG3-c2 was described as a diploid male cell line both by the karyotype analysis and by genome sequencing (Lee et al., Genome Biology 2014; <http://genomebiology.com/2014/15/8/R70>). This is indicated on page 4 of the above-cited paper: “Therefore, we also examined mitotic spreads (Figure 2; Additional files 1 and 2) to make ploidy determinations <...> BG3-c2 and 1182-4H cells were diploid.”, as well as on Figure 3A. Only a short segment of the 3L chromosome in this cell line is tetraploid, and the X chromosome is almost completely haploid. We also provide here the images of the BG3 nuclei, clearly demonstrating the presence of single FISH signals in ChrX from our FISH experiments and indicating that ChrX is, in fact, haploid in the BG3 cells used (Supplementary Fig. 14a).

14. “...with well-defined boundaries aroused from specific folding of the chromatin. To probe this, we tested the resistance of sub-TAD boundaries to the data downsampling (2-fold.” Line 187 should say prove not probe.

Reply:

We corrected this in the revised version of the text.

15. Figure 3b is confusing as the y-axis writes percentage of subTAD boundaries but both TAD and subTAD boundaries are presented in the graph.

Reply:

We corrected this in the revised version of the Figure.
We would like to note that Fig. 3b is now Fig. 4b.

16. The authors state: “*Similarly to Drosophila S2 cells³⁰ and contrary to early embryo nuclei, we observed an increased interaction frequency only between active regions in the bulk BG3 in situ Hi-C data (i.e. we confirmed the presence of the A-compartment in the BG3 cell population) (Fig. 3d, Supplementary Fig. 11)*”.

However this observation is contradictory with their findings that most inner TAD bins and unstable boundaries are enriched in heterochromatin features. Wouldn't they expect a high frequency of contacts supporting their B to B interactions if most TADs fall

into the B compartment? It is also contradictory with the long range interactions mediated by Polycomb (described below). Also there are other reports in the literature in which S2, and BG3 HiC has been performed and this observation is not fully supported (Wang et al., 2018, Keerthi T et al., 2019, Arzate-Mejia et al., 2020). The authors need to discuss this in more detail.

Reply:

We are grateful for this remark of the reviewer. Indeed, in our BG3 Hi-C and snHi-C data, we observe intra-compartment interactions of both types, A-to-A and B-to-B. However, B-to-B interactions occur on smaller genomic scales than A-to-A ones, as indicated by the contact probability plot (Fig. 4e). Thus, B-to-B interactions contribute predominantly to TAD formation, as correctly noticed by the reviewer. In contrast, the A-to-A interactions prevail on a global scale, supported by the saddle plots in Fig. 4d. These plots demonstrate the enrichment of A-to-A interactions over the expected background but do not exclude B-to-B interactions. Thus, we reformulated our statement in the main text: “*Similarly to Drosophila embryo, S2, and Kc167 cells, we observed an increased long-range interaction frequency within the A-compartment in the bulk BG3 in situ Hi-C data (Fig. 4d, e, f; Supplementary Fig. 13)*”.

To further support this result, we added average plots indicative of long-range looping (Fig. 4h-j). We observed a strong enrichment of interactions between regions from the A compartment (Fig. 4h), including the regions of chromosome X bound by dosage compensation MSL complex (Fig. 4i). However, we observed nearly no enrichment of long-range interactions between the regions of the B compartment (Fig. 4j).

We did not find this result contradictory to the published literature, and, in particular, to the studies mentioned by the reviewer. First, the suggested papers do not provide saddle plots nor average loops of the A/B compartment that might indicate enrichment of intra-compartment interactions on a whole-genome scale. Second, these papers do not mention any comparable analysis of compartments on a whole-genome scale and over a broad range of genomic distances. Wang et al. (2018) restrict their analysis of active and inactive TADs to the interactions of neighboring TADs. Chathoth et al. (2019), Keerthi T et al. (2019) mentioned by the reviewer compare the compartments/looping patterns between two cell lines but do not report the whole-genome interaction preferences for each cell line individually. Arzate-Mejía et al. (2020) provide this analysis only for a relatively small genomic fragment instead of genome- or chromosome-wide approach. Thus, our conclusion in the main text is indeed a novel and self-consistent result. Moreover, we found a study confirming the predominance of A-to-A contacts in the Kc167 cell line (Rowley et al. 2019) and added this citation to the main text.

As an answer to the second part of the reviewer’s remark, we assessed average loop plots for the top 1,000 genomic regions enriched in DARKGRAY (Polycomb chromatin state, Additional Figure 2c), H3K27me3 (Polycomb-associate chromatin mark, Additional Figure 2d), and Polycomb-associated factors dRING (Additional Figure 2e), Pc (Additional Figure 2f) and PCL coverage (Additional Figure 2g). We observed a weak looping for Pc and dRING and included the latter to the main text as Fig. 4i. The looping of Polycomb is in line with that observed in the Kc167 cell line (Eagen et al. 2017) and Drosophila embryos (Ogiyama et al. 2018).

These results are now added to the main text:

“*Supporting this observation, we also found increased long-range interactions between genomic regions of the X chromosome activated by male-specific-lethal (MSL) complex*

binding (Fig. 4h) in both BG3 in situ Hi-C data and the merged cell. In contrast, we observed a weak enrichment of long-range interactions between Polycomb-repressed regions bound by dRING (Fig. 4i) and nearly no enrichment for B-compartment regions (Fig. 4d, e, g) “.

Additional Figure 2

17. The authors describe: “We next applied dissipative particle dynamics (DPD) polymer simulations to reconstruct the 3D structures of haploid X chromosomes in individual cells from the snHi-C data (Fig. 4a, Supplementary Fig. 12)”.

As stated before the authors need to check if their BG3 cell line batch has a diploid karyotype, otherwise the model would arise from two molecules and this would have to be taken into consideration. The authors have performed FISH (Supplementary figure 15C). They should present several examples of the images analysed to confirm the quality of their signals and that they are present in just one molecule confirming the haploidy of the modelled X chromosome.

Reply:

See the response to the comment 13 of the reviewer.

Note that Fig.4a is now Fig.5a, Supplementary Fig. 12 is now Fig.14, and Supplementary Fig. 15c is now Fig.17c.

19. The authors describe: “*Due to the fact that TADs in Drosophila are largely composed of inactive chromatin, we propose that the chromatin fiber conformation within TADs is mostly determined by interactions between adjacent non-acetylated nucleosomes. In contrast, at large genomic distances, TADs interact with each other in a stochastic manner, imposing the ellipsoidal form of the CT that is observed in all model structures (Fig. 4a, Supplementary Fig. 12).*”

How do the authors reconcile the high variability they report at the subTAD level with the more stable behaviour shown in their modelling? Also how do they reconcile the high variability at the subTAD structures with their proposal of fiber conformation within TADs is mostly determined by interactions between adjacent non-acetylated nucleosomes?

Reply:

As described in the initial version of the text, according to the analysis of sub-TAD boundary stability in the downsampled datasets, sub-TADs identified in snHi-C maps resulted from local random fluctuations of a Hi-C signal: “*Hence, a potential hierarchy of TAD structure in single cells appears to reflect local Hi-C signal fluctuations*”. Thus, the “sub-TAD structure” of TADs identified should not be considered in the interpretation of the results obtained in polymer simulations. “Sub-TAD boundaries” called appear to be genome regions stochastically depleted of contacts and randomly distributed along the genome, contributing to the noise in the snHi-C data. The coefficient of the difference used to probe cell-to-cell variability in chromatin fiber folding is resistant to noisy fluctuations in the data and, thus, serves as a robust metric for the estimation of differences in chromatin folding at a broad range of genome scales.

20. The authors state: “*We found that Polycomb-occupied regions interacted with each other in a cell-specific manner and, moreover, such contacts occurred even between loci regardless of the genomic distances between them (Fig. 4g, upper panels)*”. Do the authors mapped Polycomb occupied regions to the B compartment? How does this correspond with the B compartment not being supported by interactions? The authors need to explain this discrepancy.

Reply:

We thank the reviewer for this important question. Based on modENCODE Polycomb marks (H3K27me3, dRING, Pc and PCL), we determined the 10 Kb-regions that have the top 5% coverage by each mark (439 regions in total). We then checked their compartment composition. 92.3% of these Polycomb regions were located in the A compartment called on bulk BG3 *in situ* Hi-C.

It should be noted that the A compartment is defined as the set of regions with a positive sign of the projection to the first eigenvector called on Hi-C maps. The

information about the region's activity is used when the direction of the eigenvector is selected (positive sign for the regions with more genes) [Lieberman-Aiden 2009, Imakaev 2012]. Hence, the A compartment consists of active chromatin regions predominantly, but not exclusively. On the other hand, Polycomb regions are involved in long-range interactions, similarly to active regions; thus, the algorithmic positioning of these regions in the A compartment might be explained.

Of note, Polycomb regions occupy only 4% of the total A compartment. We, thus, conclude that strong interactions of A (observed in saddle plots in Fig. 4d) occur independently of Polycomb. To clarify this in the main text, we added the average loop plot of dRING factor of Polycomb (Fig. 4i) and provided the following explanation in the main text: "*In contrast, we observed a weak enrichment of long-range interactions between Polycomb-repressed regions bound by dRING (Fig. 4i) and nearly no enrichment for B-compartment regions (Fig. 4d, e, g)*".

Additional corrections:

We noticed a technical problem with Supplementary Figures 3a and 9 regarding our re-processing of Flyamer and co-workers' (2017) snHi-C data. This problem affected our reported results for several cells in these figures. In particular, the ordering and selection of top-40 cells in Supplementary Figure 9 was wrong, as well as descriptive statistics in Supplementary Figure 3a. This error is now corrected. We also restricted our analysis to the top-20 cells of Flyamer et al. (2017) for a better comparison with our 20 cells and the top-30 cells from Gassler et al. (2017).

We also noticed an error in the annotation of Supplementary Fig. 6c, where we reported a larger number of iterations of subsampling than we actually used for the estimation of the boundary's robustness. This approach led to an underestimation of parameters for the optimal boundary refinement strategy and the number of recoverable boundaries from the subsampling procedure described in the "Robustness of TAD calling" section of Online Methods. We have now increased this number of iterations to ten. We improved Supplementary Fig. 6c, 6d, and their legends, and corrected the Online Methods. These modifications are minor and do not affect the conclusions based on a smaller number of iterations.

We also improved Supplementary Fig. 9 by removing redundant elements of the plot and increasing the image resolution.

Reviewer #3 (Remarks to the Author):

The paper is interesting and provide novel insight into the folding mechanisms in eukaryotes. I've found particularly original the experimental approach of considering single-cell HiC.

Unfortunately, my expertise is more on the theoretical side and I can give little advise if the paper can be improved further on the experimental one. Yet, it is my impression that the work is sound and well presented in all its main aspects, even the most technical ones.

The simulation work which accompanies the paper looks solid, so I have no particular suggestion related to it either.

To summarize, my judgement about the work is full positive and I recommend its publication in Nature Comm. without further hesitation.

Reply:

We thank the reviewer for positive feedback.

Additional corrections:

We noticed a technical problem with Supplementary Figures 3a and 9 regarding our re-processing of Flyamer and co-workers' (2017) snHi-C data. This problem affected our reported results for several cells in these figures. In particular, the ordering and selection of top-40 cells in Supplementary Figure 9 was wrong, as well as descriptive statistics in Supplementary Figure 3a. This error is now corrected. We also restricted our analysis to the top-20 cells of Flyamer et al. (2017) for a better comparison with our 20 cells and the top-30 cells from Gassler et al. (2017).

We also noticed an error in the annotation of Supplementary Fig. 6c, where we reported a larger number of iterations of subsampling than we actually used for the estimation of the boundary's robustness. This approach led to an underestimation of parameters for the optimal boundary refinement strategy and the number of recoverable boundaries from the subsampling procedure described in the "Robustness of TAD calling" section of Online Methods. We have now increased this number of iterations to ten. We improved Supplementary Fig. 6c, 6d, and their legends, and corrected the Online Methods. These modifications are minor and do not affect the conclusions based on a smaller number of iterations.

We also improved Supplementary Fig. 9 by removing redundant elements of the plot and increasing the image resolution.

Reviewer #4 (Remarks to the Author):

In this work, the authors characterize the chromatin structure in individual *Drosophila* nuclei using Hi-C and polymer modeling. They applied a recently developed single-nucleus biochemical Hi-C (snHi-C) assay (2017 Nature) together with a more effective strategy to filter out likely inaccurate data (called ORBITA here) to ultimately describe the genomic structure of 20 individual cells ostensibly at 10 kb resolution. They find that a very large percentage (> 40%) of TAD borders are the same between different cells, which is strikingly different from the lack of shared borders in single mammalian oocytes (in the 2017 Nature). This difference is indeed a significant and well supported finding, though less so the magnitude, as explained below. The authors then build models of the X chromosome of each cell based on the Hi-C data, from which they infer details of the folding behavior at different length scales as well as heterogeneity in the structures that are attributed to the stochasticity in biological processes in the different cells.

Overall, this is an impressive and interesting work that is very well analyzed. However, there are some major concerns that should be addressed before consideration for publication.

1. It is not clear that the authors could accurately describe this work as a genome-wide characterization of the chromatin structure with 10 kb resolution. A common definition for "resolution" of population-level Hi-C maps is that 80% of the bins contain at least 1000 reads (Rao et al, Cell 1665 (2014)). While this is undoubtedly beyond the capability right now for single cell Hi-C, it is not clear what justification the authors used to settle on 10 kb. It would appear as though that many (about half) of the cells in this

work have only 1 or 2 reads (on average) for each 10 kb bin, which is very low. In the previous application of this method to mammalian cells (2017 Nature), which I believe described data with roughly the same number of reads per genome length as in the present work, the maps were analyzed at 40 kb resolution (Fig 2c in that paper). For each cell, what percentage of the genome are bins with no reads? With at least some cells (such as Cell 4 in Fig 2a), it appears as though there could be ~40% of the bins with no reads. So, describing this characterization as “genome-wide” might be inaccurate, if, strictly speaking, there is a very large portion of the genome with no data. Perhaps there are extended regions of the genome for which there is sufficient data in many cells to make sound conclusions, but other regions of the genome for which there is insufficient data. Alternately, perhaps 20 kb or 25 kb resolution could be justified in some legitimate way (such as, for example, two reads per bin in 80% of the genome in the majority of cells) and still be small enough to identify the 80-100 kb TADs.

Reply:

As correctly pointed out by the reviewer, the data quality of snHi-C is currently limited and cannot reach that of bulk Hi-C. We agree that the common definition of resolution is not applicable to snHi-C. However, in the revised version of the paper, we demonstrated that the quality of our *Drosophila* snHi-C datasets at 10 kb was comparable, if not better, than that of the previously published datasets at 40 kb. We added Fig. 2a-d and improved the main text correspondingly: *“To estimate the overall quality of the snHi-C libraries, we first calculated the number of captured contacts per cell. On average, we extracted 33,291 unique contacts from individual nuclei that represented 5% of the theoretical maximum number of contacts and corresponded to four contacts per 10-kb genomic bin (see Methods); in the best cell, 17% of contacts were recovered (Fig. 2a, b, Supplementary Table 1). Relying on the number of captured contacts, we then estimated the proportion of the genome available for the downstream analysis. At 10-kb resolution, ~82% of the genome on average was covered with contacts in each individual cell, and 67% of genomic bins established more than 1 contact (Fig. 2c). Notably, in the previously published mouse snHi-C datasets, ~0.6% of theoretically possible contacts were detected on average (Fig. 2b). Because the top-20 mouse snHi-C libraries from Flyamer et al. demonstrated a comparable genome coverage with contacts and a number of contacts per 10-kb genomic bin (Fig. 2d), we could directly compare the Drosophila and mouse snHi-C maps (see below)”*.

Below, we summarize the modifications and describe our reasoning.

The effective resolution in Flyamer et al. (2017) and Gassler et al. (2017) is 40 kb for cell-specific analyses, such as TAD calling and insulation score calculation. Thus, we consider the data quality for these snHi-C experiments at 40 kb as a gold standard.

We now justified that 10-kb resolution for our snHi-C is comparable to that gold standard or better. For that, we reported quality controls at four different resolutions (10, 20, 40, and 100 kb) for snHi-C on 120 oocytes from Flyamer et al. (2017), 32 G2 zygotes from Gassler et al. (2017), and 20 *Drosophila* cells from this study. We also assessed these numbers for the merged *Drosophila* dataset. The results of the comparison are presented in a new Fig. 2 of the revised paper.

Following the reviewer’s suggestions, we calculated the following quality controls:

1. the percentage of bins with zero contacts out of the total (Percentage of Zero Bins, PZB),

2. the percentage of bins with at least two contacts out of the total (Percentage of Bins with at least 2 Contacts, PBC2),
3. the mean number of contacts per bin per dataset (Mean Contacts per Bin, MCB).

On average, PZB for *Drosophila* single nuclei at 10-kb resolution is $18.1 \pm 13.9\%$. For mouse oocytes (Flyamer et al. 2017) and G2 zygotes (Gassler et al. 2017), PZB is much higher, being $43.8 \pm 34.5\%$ and $52.5 \pm 20.3\%$, respectively; see Additional Figure 3a,g).

The mean PBC2 for our dataset is $67.4 \pm 20.9\%$ at 10 kb, significantly higher than for oocytes and zygotes at 40 kb ($43.7 \pm 35.7\%$ and $29.5 \pm 16.7\%$, respectively; see Additional Figure 3c).

Thus, the coverage of snHi-C at 10 kb in our experiment is significantly higher than the coverage of snHi-C in the Flyamer and Gassler mouse datasets at 40 kb resolution.

The reviewer suggested the formal criterion “at least two reads per bin in 80% of the genome”. Notably, this criterion is satisfied for 7 out of 20 cells at 10 kb for *Drosophila*, and for 25 out of 120 cells at 40 kb for Flyamer et al. (2017) oocytes. If we decrease this threshold and allow at least two reads per bin in 60% of the genome, 14 out of 20 cells satisfy the criterion at 10 kb for *Drosophila* and only 51 out of 120 cells at 40 kb for oocytes.

Notably, PZB at 10 kb for Cell 4 is 6.6% (see Additional Figure 3a), which resolves the reviewer’s comment that “for Cell 4 there could be ~40% of the bins with no reads for this cell”. Only two cells in our analysis do not satisfy the proposed criterion.

PZB of the merged dataset is as small as 0.86% (see Additional Figure 3a). Thus, with 20 cells, we see contacts in almost the entire genome. We, therefore, believe that our characterization of the contacts in single nuclei in *Drosophila* cells is indeed genome wide, even though a smaller fraction of the genome is captured in each individual cell.

In this study, we called TADs for top-coverage oocytes from Flyamer et al. (2017) at the 10-kb resolution, with the mean PZB being $64.9 \pm 28.4\%$ for all oocytes. This number is significantly higher than that for *Drosophila*. However, in the revised version of the paper, we consider only the top-20 oocytes by coverage (see Supplementary Fig. 9). The mean PZB for them is comparable to that in our dataset, $21.0 \pm 5.2\%$ (see Additional Figure 3b,d,f,g). In the main text, we calculated the complementary measure, the percentage of non-zero bins (1-PZB), and included it as Fig. 2c.

The third quality control, MCB, is 4.2 on average for our cells (sd 3, median 3.3, see Additional Figure 3e,f,g). This resolves the reviewer’s comment that “many (about half) of the cells in this work have only 1 or 2 reads (on average) for each 10 kb bin”. In fact, 7 out of 20 *Drosophila* cells at the 10-kb resolution have $MCB < 3$, and all 20 have $MCB > 1$. For comparison, all G2 zygotes from Gassler et al. (2017) and 78 out of 120 oocytes from Flyamer et al. (2017) have $MCB < 3$ at the 40-kb resolution.

Finally, as the reviewer has suggested, we tested TAD calling at different resolutions for both *Drosophila* and mouse datasets and analyzed the reproducibility of TAD boundaries. We applied the same procedure for TAD calling, as described in Methods for 10 kb, and we calculated the percentage of shared boundaries allowing a 1-bin offset.

On average, 84.8% of boundaries found in *Drosophila* at the 20-kb resolution and 78.6% of boundaries at the 40-kb resolution have a matching boundary at the 10-kb resolution. This result is significantly higher than the 43% and 58% expected at random, respectively. Thus, we conclude that the choice of the 10-kb resolution does not affect our conclusions.

For oocytes from Flyamer et al. (2017), 69.5% of boundaries found at the 40-kb resolution match a boundary at the 10-kb resolution, which is significantly higher than the 51.7% expected at random. Nevertheless, the resolution could affect one of the key conclusions of our work, that TAD boundaries in *Drosophila* are more stable compared to mice. However, on average, the percentage of shared boundaries at 40 kb is 26.0% for the top-20 oocytes of Flyamer et al. (2017) and 35.3% for *Drosophila* at the same resolution (allowing one bin offset). Thus, this conclusion holds for different bin sizes, and we would, therefore, like to retain the resolution of 10 kb for all analyses in this study.

Note that Fig. 2a is now Fig. 2e, and Fig. 2c is now Fig. 3b.

Additional Figure 3

g Quality controls for snHi-C datasets at different bin size

Bin size (Kb)	Data source	Percentage of Zero Bins, PZB %					Percentage of Bins with at least 2 Contacts, PCB2 %					Mean Contacts per Bin, MCB				
		mean	std	min	50%	max	mean	std	min	50%	max	mean	std	min	50%	max
10	Flyamer	64.9	28.4	8.7	70.3	99.8	20.0	21.4	0.02	12.3	83.0	0.8	0.9	0.003	0.5	4.7
	Flyamer, top 20 cells	21.0	5.2	8.7	20.6	28.4	59.0	9.0	46.1	58.0	83.0	2.5	0.7	1.7	2.3	4.7
	Gassler	75.1	12.7	50.9	75.6	99.1	11.5	7.2	0.1	10.6	26.6	0.5	0.3	0.01	0.4	1.0
	Gassler, top 20 cells	70.8	11.3	50.9	73.3	89.9	13.9	7.1	3.0	12.0	26.6	0.6	0.3	0.1	0.5	1.0
	This work	18.1	13.9	1.6	14.6	48.3	67.4	20.9	29.3	69.2	97.4	4.2	3.0	1.1	3.3	12.1
20	Flyamer	53.9	33.1	4.9	53.2	99.6	31.7	29.9	0.04	23.5	93.2	1.5	1.7	0.005	0.9	8.6
	Flyamer, top 20 cells	9.5	2.5	4.9	9.1	13.5	80.9	6.1	71.2	81.6	93.2	4.5	1.3	3.0	4.2	8.6
	Gassler	64.6	16.8	34.5	64.3	98.6	19.0	11.6	0.2	17.8	43.0	0.8	0.5	0.02	0.7	1.8
	Gassler, top 20 cells	58.9	14.2	34.5	61.7	84.3	22.9	11.1	5.2	19.8	43.0	0.9	0.5	0.2	0.8	1.8
	This work	8.2	8.2	0.8	5.0	29.1	83.3	15.2	49.8	87.7	99.0	7.4	5.1	2.1	5.9	20.7
40	Flyamer	43.8	34.5	4.1	32.8	99.3	43.7	35.7	0.1	40.8	95.5	2.7	3.1	0.01	1.7	15.9
	Flyamer, top 20 cells	5.2	0.7	4.1	5.0	6.4	92.2	2.1	88.5	92.8	95.5	8.3	2.4	5.3	7.7	15.9
	Gassler	52.5	20.3	20.1	50.3	97.8	29.5	16.7	0.3	28.7	62.4	1.3	0.8	0.03	1.2	3.0
	Gassler, top 20 cells	45.5	15.9	20.1	47.8	76.0	35.2	15.2	9.1	31.2	62.4	1.5	0.8	0.4	1.3	3.0
	This work	2.8	3.5	0.4	1.2	12.8	93.3	8.1	72.8	96.5	99.5	12.7	8.6	3.7	10.4	34.6
100	Flyamer	32.3	31.3	3.5	12.1	98.6	57.0	37.4	0.1	74.7	96.2	6.0	6.9	0.02	3.8	35.5
	Flyamer, top 20 cells	3.9	0.1	3.5	3.9	4.1	95.7	0.3	95.1	95.8	96.2	18.5	5.5	11.4	17.4	35.5
	Gassler	35.2	20.6	9.1	31.1	90.1	47.6	21.2	3.1	48.8	82.8	2.4	1.4	0.1	2.2	5.6
	Gassler, top 20 cells	29.4	15.3	9.1	29.6	61.7	53.9	18.5	17.4	50.7	82.8	2.8	1.4	0.7	2.4	5.6
	This work	0.7	0.6	0.2	0.5	2.7	98.4	1.8	93.6	99.0	99.7	25.4	16.5	7.7	21.0	66.4

2. Much of the description of the heterogeneity in structures stems from an analysis of the models. However, from the analysis shown in Suppl 15a,b, most of these structures disagree significantly with the Hi-C data. Nine of the 20 have an FNR > 0.5, which is somewhat shocking as this indicates that more than ½ of the contacts that are present in the Hi-C data are not present in these models. And this is with sparse Hi-C data. An FNR cutoff of 0.2, reflecting the absence of 20% of the Hi-C contacts, is satisfied by only 4 of the 20. For the TPR, an equally surprising 16 of the 20 are below a value of 0.5, which indicates that less than ½ of the contacts that are present in these models are also present in the Hi-C data. The authors provide FISH data that agrees with the models, but it could be argued that most of the structures do not adequately agree with the Hi-C data. With so poor FNR and TPR in the majority of structures, by what criteria do the authors conclude that these structures are consistent with the Hi-C data? It is also unnerving that those cells with the highest number of contacts (Cells 1 to 3) are associated with the least consistent models and those with the fewest number of contacts (Cells 15 to 20) are associated with the most consistent. It should be that more data leads to more reliable models, not less reliable models. During the modeling, I believe that the authors effectively down-sampled their data by 1/3 to 1/2 to remove overly stretched bonds. Are the loci pairs whose bond was removed during this down-sampling in close proximity in the final structures to possibly contact? Judging by the FNR and TPR, I am not optimistic, but this would provide some additional evidence that the models are sufficiently consistent with the Hi-C data to warrant more detailed analysis.

Reply:

As proposed by the reviewer, we directly compared the distances between snHi-C-derived pairs of loci for which the bonds had been removed and between loci for which the bonds had been included in the final models. The comparison shows that the average distance between loci with removed bonds was dramatically larger than between bonded loci, indicating that the obtained models correctly reproduced the contact patterns used for polymer simulations (Additional Figure 4).

However, we observed that the distributions overlap at the smaller distances indicating that the loci for which the bonds had been removed could be located in close proximity to each other in the final models. We explain these rare cases by the contacts established by the neighboring loci of these regions. Even though our approach slightly overestimates the number of bonds to be removed, we note that these bonds do not change the overall 3D structure and do not influence the downstream analysis.

To support our downsampling procedure, we performed modeling of a mammalian single-cell Hi-C dataset that was previously demonstrated to result in biologically significant models (Stevens et al. 2017 PMID 28289288). We selected chromosome 13 with the comparable number of monomers at 50-kb resolution (2396) and obtained TPR and FNR similar to that observed in our study of *Drosophila* chromosome X. TPR for mammalian Cell 1 was 0.43, Cell 2: 0.49, Cell 3: 0.47 and FNR for Cell 1: 0.57, Cell 2: 0.51, Cell 3: 0.53. Because the modeling based on this mammalian dataset with a similar algorithm is now considered to be a gold standard of single-nucleus modeling, we conclude that our FNR and TPR point to the high quality of modeling.

As additional support for our models, we note that despite a profound loss of observed snHi-C contacts, the modeling procedure preserves the important biological properties of real snHi-C data, particularly compartments and TADs, as described in the Results section: “As revealed by TAD annotation, the DPD simulations successfully reproduced chromatin fiber folding even at short and middle genomic distances because TAD positions along the X chromosome were largely preserved between the models and the original snHi-C data (Fig. 5a, Supplementary Fig. 15, 17a, b; also see Methods)”. The

fact that TAD profiles are highly concordant between the original snHi-C data and the models indicates that the contact patterns captured in the experiment and 3D structures based on these patterns are sufficiently consistent for the reliable downstream analyses of the spatial chromatin structure within the X chromosome territory.

3. On the same subject, how can the authors be sure that some, maybe much, of the heterogeneity observed in the Hi-C data is not owing to the low (random) sampling during the biochemical assays? Two similar structures whose contacts are sparsely (randomly) sampled might only appear to be different. It might also be noted that while Fig 2a indeed shows that the distributions of these two cells are non-random, this of course does not mean that all of the Hi-C data is non-random.

Reply:

We agree with the reviewer that some of the observed heterogeneity may arise from random loss of the DNA at different steps of the snHi-C protocol. Indeed, we cannot directly separate the heterogeneity arising from technical issues and actual differences in chromosome structures between cells. However, we can assess the heterogeneity of reconstructed models relative to the purely random background. The models based on the shuffled data cannot represent the same fiber folding and might be viewed as the expected background of heterogeneity. We provided the comparison with this background in Fig. 5b, where we plot the coefficient of difference between real models and shuffled models on different genomic scales. We observed that the models based on the real snHi-C data differed from each other more than models based on the shuffled data. Hence, the original structures of chromosomes are indeed dissimilar conformations, and this is adequately reflected in our models. Thus, the heterogeneity should be considered as biologically meaningful and not as technical noise. We provide additional reasoning supporting our results in the answer to comment #5.

4. The conclusion that there is greater conservation of the TAD boundaries between individual *Drosophila* cells than between the individual mammalian oocytes is supported by the analysis presented in this work and it is a genuinely significant result. However, that it is “over 40% of TAD boundaries” that are conserved between the cells is not well supported here since from “randomly shuffled maps”, the same analysis shows that over 32% of TAD boundaries are conserved. Shouldn't the “random” data give essentially no shared borders, almost by the definition of “random”? Perhaps the TAD caller would be expected to identify a few TADs in noisy, sparse data, but why should the borders be at the same place in so many cells? Does this point to an inaccuracy in the TAD caller or calling method used with data that is so sparse? There are other TAD callers (for example TopDom, Zufferey et al, *Genome Biology* 19, 217 (2018)) or different values of the gamma parameter that could be examined to yield essentially no shared borders in the random data that, when obtained, could then be used to more correctly estimate the % of shared borders in the snHi-C data.

Reply:

We thank the reviewer for this remark. The mean number of TAD boundaries detectable in individual cells is 1,460 out of the total genome size of 11,901 bins (10-kb bin size, chromosome X and all autosomes except for chromosome 4). Thus, approximately every eighth bin of the genome is annotated as a TAD boundary. Two random sets with these properties are expected to have ~12.3% of shared boundaries. In our analysis, we allow for one bin offset of the one set of boundaries, which will reduce the effective size of the genome and increase the expected percentage. However, this does not account for the restrictions on the TAD/inter-TAD sizes. Hence, we used two types of controls.

Firstly, we called TADs on randomized snHi-C maps. On average, the percentage of shared boundaries was 32.9% between these maps, ~7% lower than that for real maps (Fig. 3d). This result is significant and sufficiently substantial to demonstrate that the observed boundaries are not random.

The second control was shuffling of the positions of TADs/inter-TADs so that the size distributions were preserved. This approach resulted in the percentage of shared boundaries of 33.1% on average (~6% smaller than the observed mean). This procedure was used to calculate the significance of the observed percentages in Supplementary Figures 8b, 8e, 9b. Notably, most of the percentages of shared boundaries between the real cells are significantly larger than expected at the 0.01 confidence level, but it is not so for the shuffles (Supplementary Fig. 8b). Therefore, it was concluded that the observed percentages are indeed significant and are higher than expected.

We consequently changed the text on page 7 as follows: *“This is significantly higher than the percentage of shared boundaries for shuffled control maps (32.9%) and the percentage expected at random (33.1%, Fig. 3d). Notably, 42% of NBT-identified single-cell TAD boundaries were conserved in pairwise cell-to-cell comparisons (Supplementary Fig. 7b), supporting the results obtained in the analysis of modularity-derived TAD boundary profiles”*. We also added the percentages of shared boundaries in real data and two types of controls to the main text, Fig. 3d.

In the revised version of the MS, we used a modification of the recently published spectral clustering method based on non-backtracking (NBT) random walks to additionally validate the single-cell TAD segmentations. The obtained single-cell TAD profiles (i) are remarkably similar to the modularity-derived segmentations, (ii) demonstrate the same conservation between individual cells in pairwise comparisons, and (iii) NBT TAD boundaries are also highly enriched with active chromatin marks. We conclude that the TAD profiles extracted from the single-cell Hi-C data are robust to the method of their identification and, thus, biologically relevant (see the Methods section of the revised paper for details). We also added these new results to the Results section: *“To additionally validate the single-cell TAD segmentations, we utilized a modification of the recently published [ref] spectral clustering method based on the non-backtracking random walks (NBT; see Methods). The non-backtracking operator is used to resolve communities in sufficiently sparse networks, thus providing a useful tool for TAD annotation in single-cell Hi-C matrices. The resulting average size of the detected TADs was 110 kb, closely matching the typical TAD size in the population-averaged data and in the single-cell modularity-derived segmentations. The mean number of detected TADs per cell (855 and 920 for the NBT and modularity, respectively) and single-cell TAD segmentations were remarkably similar between the two methods (Supplementary Fig. 7a) and demonstrated the same epigenetic properties (Supplementary Fig. 7c, see below)”*.

5. At first glance, I expected that the data presented in Fig 2a was an analysis of the whole genome for these two cells. However, I believe that this is not correct, as I believe that there is only about 20 Mb of the genome covered in the analysis for Cell 4 and about 5 Mb for Cell 6 (with a reference genome total length of about 130 Mb). Why was only such a small portion of the genome analyzed, especially for Cell 6 (4%)? It would be better to see this analysis from a more substantial portion of the genome, if not the full genome. But if this is computationally too demanding, at least some discussion for the reasons for choosing the regions that were analyzed should be included.

Reply:

Analysis of the whole chromosome 2R (25.3 Mb, 19% of the entire genome) is shown for both cells. In Supplementary Fig. 4k, we also provide the results for chromosomes 3R and X as additional examples. Moreover, heatmaps of log₁₀ of p-values for the top-

10 cells (chromosomes 2R, 3R, X) are shown in Supplementary Fig. 4j. We believe that this information is sufficient to demonstrate the robustness of the analysis.

6. It is difficult to judge whether the TADs called in the individual cells in Fig 2b are in fact obvious in the figure since the black lines overlap and somewhat obscure the data. Perhaps the authors could depict the TADs as bars underneath the maps as they did in Fig 1 of their 2016 Genome Research paper.

Reply:

We modified the figure according to the reviewer's suggestions.

7. The models were found to have the active chromatin within the CT interior and the inactive regions on the CT surface. By contrast, in mammalian cells, active transcription occurs on the surface of CTs (Shah et al, Cell 174, 363 (2018)). In the population-level Hi-C data of these cells, are there more inter-chromosome contacts between the A compartments or between the B compartments? If there are more contacts between the A compartments, particularly with the X chromosome, this would conflict with the models. Also, some description of how "contact" is defined in the analysis of the models to generate the corresponding Hi-C maps should be given in the Methods.

Reply:

Although chromatin compartments originate due to interactions that occur both in *cis* (within the same chromosome territory) and in *trans* (between chromosomal territories), the contribution of *cis* interactions to the compartment strength is more prominent. For this reason, only *cis* interactions are taken into consideration when the compartment strength is estimated in most studies. We also considered interactions within the same chromosome territory. Multiple associations of active regions distributed across the entire chromosome expectedly result in the formation of an active chromatin core. If these interactions occur without much specificity, the active chromatin regions should be displaced toward the center of the chromosome territory where the probability of distinct active chromatin regions meeting each other would be higher. In contrast, noninteracting B-compartment segments should be displaced toward the periphery of the chromosome territory, as observed in this study. We note that in this scenario, the interaction of active chromatin regions would constitute an important force supporting the round shape of a chromosome territory.

To check the reviewer's suggestion, we normalized intra-compartment interactions (termed AA and BB for A and B compartment, respectively) *in cis* and *in trans* by the total number of interactions within A and B compartment for each snHi-C map (Additional Fig. 5). The ratio of *trans* AA over total AA is larger than *trans* BB over total BB. Thus, the A compartment establishes more inter-chromosome contacts as compared to the B compartment.

At first glance, this observation conflicts with the preferential location of active chromatin inside the chromosome territory. However, this contradiction can be explained:

(i) the observed localization of active regions inside the CT is a tendency but not a strict rule. Active regions tend to associate with each other in transcription-related nuclear bodies/microcompartments, such as transcription factories, speckles, and activatory hubs. The observed tendency for active regions to localize inside the chromosome territory does not exclude the possibility for their occasional looping towards the chromosome surface, where they can establish stable contacts with active regions from other chromosomes. This looping can be a source for an increased number of trans-contacts for regions from the A compartment.

(ii) repressed regions are largely represented by nuclear lamina-associated domains. Being exposed to the chromosome surface, they interact with the nuclear lamina, which constrains their movement in the nuclear space and potentially interferes with their interactions

(iii) in the modeling, we consider the 10-kb chromatin region as a spherical bead. However, such a chromatin region can occupy volumes of different shapes. Elongated conformations may have contacts far from their center of mass and outside the modeling bead. In the modeling, we do not consider any internal degrees of freedom of a bead; therefore, we cannot take into account the possible non-spherical shape of the region and possible far contacts. Hi-C is done at the restriction-fragment resolution and can detect any contacts regardless of the fragment conformation. In this sense, our models capture the general shape and path of chromatin fiber but may not capture a minor fraction of true contacts that occur far from the center of mass of the bead, particularly at the surface of CT.

Additional Figure 5

The reviewer:

“Also, some description of how “contact” is defined in the analysis of the models to generate the corresponding Hi-C maps should be given in the Methods”.

Our reply:

The Methods section on page 50 was modified as follows. “We define contact as an event when the distance between two beads (i, j) meets criterion $D_{ij} < R_{cut} = 0.7$. Such R_{cut} value corresponds to the average bond length. We count all the contacts in the system. So, in a system any bead can have more than 1 contact.”

8. Were the TADs called in the oocyte data at 40 kb resolution or 10 kb resolution? Was a similar strategy used to define the gamma as with the Drosophila cells? After ORBITA, how many contacts per cell are there? Perhaps a few words in the Methods could be included.

Reply:

TADs in oocytes were called at 10-kb resolution similarly to the Drosophila datasets and using exactly the same strategy to define the gamma value. The number of ORBITA-captured contacts in oocytes is now shown in Fig. 2. We added additional information on this issue in the Methods section.

9. Finally, line 153 in the text indicates that 40.5% of boundaries were shared between cells but the number in the legend to Suppl Fig 7 is 39.45%. Also it is the “–log10

values that are shown in Suppl Fig 7b. I believe that the legend to Suppl Fig 2b and 2c refers to the data that is shown in Suppl Fig 2c and 2b, respectively. In the legend to Suppl Fig 15a and b, the definition of “false negatives” is literally the same as that of “true positives”. Which probe set is shown in Suppl 15d? And it might be easier to appreciate the “Coverage” in Suppl Fig 5 if it was converted to percentage as in Fig 2c.

Reply:

We corrected these issues and (i) replaced 40.5% with 39.5% of shared boundaries in the Results section; (ii) replaced “log10” with “-log10” in the legend to Suppl. Fig. 7b (moved to 8b in the new version); (iii) interchanged panels “b” and “c” in Suppl. Fig. 2; (iv) Set2 is now visualized in Suppl. Fig. 15d (17d in the new version, we added this information to the legend). We note that in Suppl. Fig. 5 and in Fig. 2c (3b in the new version), “coverage” is shown at the middle vertical axis.

Additional corrections:

We noticed a technical problem with Supplementary Figures 3a and 9 regarding our re-processing of Flyamer and co-workers' (2017) snHi-C data. This problem affected our reported results for several cells in these figures. In particular, the ordering and selection of top-40 cells in Supplementary Figure 9 was wrong, as well as descriptive statistics in Supplementary Figure 3a. This error is now corrected. We also restricted our analysis to the top-20 cells of Flyamer et al. (2017) for a better comparison with our 20 cells and the top-30 cells from Gassler et al. (2017).

We also noticed an error in the annotation of Supplementary Fig. 6c, where we reported a larger number of iterations of subsampling than we actually used for the estimation of the boundary's robustness. This approach led to an underestimation of parameters for the optimal boundary refinement strategy and the number of recoverable boundaries from the subsampling procedure described in the “Robustness of TAD calling” section of Online Methods. We have now increased this number of iterations to ten. We improved Supplementary Fig. 6c, 6d, and their legends, and corrected the Online Methods. These modifications are minor and do not affect the conclusions based on a smaller number of iterations.

We also improved Supplementary Fig. 9 by removing redundant elements of the plot and increasing the image resolution.

REVIEWER COMMENTS

Reviewer #1 (Remarks to the Author):

The authors have addressed the main points that I raised, but there are two comments that still need to be considered.

Point 1. I appreciate that the authors did the requested analysis. "we preferred to retain our initial (<50% and >50%) thresholds in the revised version". I would disagree and think it is important to include the updated figure in the revised version of the manuscript.

Point 5. "the overwhelming majority of signals are from the interior of TADs, and (ii) these neighboring boundaries are located at different distances from the target boundary; hence, their signals do not contribute significantly". I am not sure if this is addressed in the manuscript/discussed. The authors need to add the numbers/percentages in the manuscript so the reader can judge these assumptions by themselves.

Reviewer #2 (Remarks to the Author):

The authors have addressed all concerns in full and the current manuscript is robust with more detailed arguments and rationales throughout.

The manuscript is suitable for publication in Nat Comm without further modifications.

Reviewer #4 (Remarks to the Author):

The authors have largely addressed my concerns. In particular, their efforts on the justification of the stated resolution is noteworthy – it lays bare in useful quantitative terms just how many reads per bin in their (and previous) data. It is likely that this will be the standard means of determining "map resolution" in single-nuclei Hi-C studies in the near future. The explanation of the significance of the difference in magnitude of the shared borders compared to the "random" case is also clear.

There are just a few issues that the authors may wish to address.

1. I am still a little concerned with the modeling. By the authors' own criteria mentioned in the reply, only 5 of the 20 *Drosophila* models exhibit both TPR and FNR values superior to those calculated from chromosome 13 from Stevens et al. And these 5 are among the 6 whose cells have the lowest number of reads. Also, there is no indication that the model generated of chromosome 13 matches that obtained by Stevens et al., who modeled all chromosomes in the nuclei at once (unlike what was done in this work). Inspection of the A/B compartments within the chromosome territories in Stevens et al seems to indicate A compartments that are not completely surrounded by B compartments, which would better explain AA interactions between chromosomes. Still, admittedly, the models described here reproduce the TAD and A-compartment features in the snHi-C data, even with the discarding of 1/3 to 1/2 of the contacts. If no other changes, perhaps the authors could comment on what they are concluding of these discarded contacts in terms of the generated structure. That is, by ignoring this data, are they indicating that this is just noise (and not a genuine feature of the physical genome structure from which the data were generated)? Strictly speaking, the models cannot account for 1/3 to 1/2 of the experimentally determined contacts.

2. I believe that there is still something incorrect about the data presented in Fig 2e. Adding up

the total number of windows, I get about 200 for Cell 4 and 50 for Cell 6. With a window width of 100 kb (as stated in the Fig 2e legend), this means about 20 Mb for Cell 4 and 5 Mb for Cell 6. In the reply, the authors stated that there is 25 Mb covered for both cells. This is close to my estimate for Cell 4, but significantly off for Cell 6. So I think that there is something wrong with the numbers for Cell 6.

3. The last sentence of the Introduction reads "At least 50% of TAD boundaries identified in each individual cell bear active chromatin marks and are highly reproducible between individual cells." (line 123, p.4). I believe that this was not stated in the Results section: the authors mentioned that 39.5% of all borders were shared between different cells in pairwise comparisons and the stable borders (defined as present in more than 50% of the cells) had high levels of active chromatin marks. But nowhere is it mentioned that "at least 50% of TAD boundaries identified in each cell" bear the active chromatin marks. That is, it is not stated anywhere that more than 50% of the borders in cell 1 (or any cell) have the active marks. And neither is it mentioned that this 50% of borders that bear the active marks are "highly reproducible" between individual cells. Perhaps this apparent discrepancy can be clarified.

4. Line 308, p.9 says "We next applied dissipative particle dynamics (DPD) polymer simulations to reconstruct the 3D structures of haploid X chromosomes (Supplementary Fig. 14a)". But Suppl. Fig 14a does not show anything related to the simulations.

5. The new model depicted in Fig 6c is intriguing but I believe conflicts with the higher resolution data from Wang et al (ref 25), which shows a single, well-defined border separating the large TADs (associated with inactive chromatin) and the proximal small TAD within the active chromatin. This new model would also not explain the presence of well-defined small TADs within the active chromatin in this data, which were also observed by microscopy in Mateo et al, Nature 568, 49 (2019).

6. Finally, the authors failed to correct the legend to Suppl Fig 15a and b: the definition of "false negatives" is still literally the same as that of "true positives".

Reviewer #1 (Remarks to the Author):

The authors have addressed the main points that I raised, but there are two comments that still need to be considered.

Point 1. I appreciate that the authors did the requested analysis. “We preferred to retain our initial (<50% and >50%) thresholds in the revised version”. I would disagree and think it is important to include the updated figure in the revised version of the manuscript.

Reply:

We present the figure with the thresholds suggested by the reviewer in the Supplement as Supplementary Figure 12.

Point 5. “the overwhelming majority of signals are from the interior of TADs, and (ii) these neighboring boundaries are located at different distances from the target boundary; hence, their signals do not contribute significantly”. I am not sure if this is addressed in the manuscript/discussed. The authors need to add the numbers/percentages in the manuscript so the reader can judge these assumptions by themselves.

Reply:

In the revised version of the MS, we provide distributions of distances between neighboring boundaries as Supplementary Figure 13. The fact that “overwhelming majority of signals are from the interior of TADs” can be derived from the number of boundaries per genome provided with the Figure. The fact that the neighboring boundaries are located at different distances from the target boundary can be observed directly from the distribution of distances. We note that the peak at 10 Kb denotes that the boundaries are frequently located at the neighboring bins, which contributes to the slight broadening of the peak in epigenetic profiles in Fig. 3f and Supplementary Fig. 11 and 12. We note that besides that, the distributions have the heavy tail that spans the distances up to 150 and more Kb. This confirms that we indeed catch the signal from the neighboring boundaries, but the contribution does not introduce the systematic bias to our analysis. Moreover, with the distributions provided as Suppl. Fig 13, the reader can derive these conclusions by themselves, as suggested by the Reviewer.

In text: “...and analyzed them separately (number of boundaries of each type and distances between neighbouring boundaries within each type are shown in Supplementary Fig. 13)”.

Reviewer #2 (Remarks to the Author):

The authors have addressed all concerns in full and the current manuscript is robust with more detailed arguments and rationales throughout.

The manuscript is suitable for publication in Nat Comm without further modifications.

Reply:

We appreciate this comment from the Reviewer. We note that the Reviewer’s suggestions are a valuable contribution, and we are grateful for the improvement of our work.

Reviewer #4 (Remarks to the Author):

The authors have largely addressed my concerns. In particular, their efforts on the justification of the stated resolution is noteworthy – it lays bare in useful quantitative terms just how many reads per bin in their (and previous) data. It is likely that this will be the standard means of determining “map resolution” in single-nuclei Hi-C studies in the near future. The explanation of the significance of the difference in magnitude of the shared borders compared to the “random” case is also clear.

Reply:

We appreciate this comment from the Reviewer. We note that the Reviewer’s suggestions are a valuable contribution, and we are grateful for the improvement of our work.

There are just a few issues that the authors may wish to address.

1. I am still a little concerned with the modeling. By the authors’ own criteria mentioned in the reply, only 5 of the 20 *Drosophila* models exhibit both TPR and FNR values superior to those calculated from chromosome 13 from Stevens et al. And these 5 are among the 6 whose cells have the lowest number of reads. Also, there is no indication that the model generated of chromosome 13 matches that obtained by Stevens et al., who modeled all chromosomes in the nuclei at once (unlike what was done in this work). Inspection of the A/B compartments within the chromosome territories in Stevens et al seems to indicate A compartments that are not completely surrounded by B compartments, which would better explain AA interactions between chromosomes. Still, admittedly, the models described here reproduce the TAD and A-compartment features in the snHi-C data, even with the discarding of 1/3 to 1/2 of the contacts. If no other changes, perhaps the authors could comment on what they are concluding of these discarded contacts in terms of the generated structure.

That is, by ignoring this data, are they indicating that this is just noise (and not a genuine feature of the physical genome structure from which the data were generated)? Strictly speaking, the models cannot account for 1/3 to 1/2 of the experimentally determined contacts.

Reply:

Indeed, we have applied our protocol to reconstruct the human chromosome 13 based on the data from Stevens et al. As indicated in the MS, for the model of chromosome 13 constructed with our protocol, we gained TPR and FNR values similar to the values obtained using data of Stevens et al. Moreover, we compared reconstructed models of Chromosome 13 for 3 cells using our protocol and protocol of Stevens et al. For that, we calculate the similarity measure between models, as we do in the manuscript. In particular, we calculate the distance matrices of the conformations, then adjust spatial scales through the normalizing to their radius of gyration, and calculate the Difference coefficient $\|M1-M2\| / \|M1 + M2\|$. Despite the different resolutions (50 Kb in our model and 100 Kb for the data deposited in GEO by Stevens et al.), we conclude that the conformations are similar, see Attached Figure 1. Violet points represent similarity rate between models of the same chromosome (chr13 for cell 1 modeled using our and Stevens et al. approaches) and yellow points represent similarity rate between models of chromosome 13 from different cells (i.e. chr 13 for **cell 1** modeled using our approach

Additional Figure 1

and chr 13 for **cell 2** modeled by Stevens et al. approach). We note that datasets from Stevens et al. are currently the gold-standard for modeling of 3D-structures from single-cell Hi-C. Thus, our models in *Drosophila* fit this standard well.

We note that the comparison of general arrangement of A- and B- compartments within the single chromosome in mouse ES cells (models of Stevens) and *Drosophila* (our models) was beyond the aim of our study. Mammals and *Drosophila* differ substantially in terms of mechanisms governing chromatin topology. Thus we do not focus on the reconstruction of whole-nucleus models or models of autosomes in our study. Furthermore, in our cell line and cell cycle stage, autosomes are diploid restricting us from reconstructing other chromosomes for *Drosophila* at the same quality rate. We do agree with the Reviewer that TPR and FNR reach best values for the systems with low number of contacts in our models. This could be explained in the following way. Low number of contacts slightly constrains the spectrum of possible 3D shapes of a polymer. Consequently, it allows the polymer to adopt a configuration where the majority of experimentally captured contacts are realized (high TPR). At the same time, a polymer with a low number of input contacts folds into a “loose” globule which does not facilitate the formation of extra contacts (low FNR).

Of note, we assume that cross-ligation products reflect close spatial proximity. However, this is a trend rather than a strict rule. After partial eviction of histones by SDS treatment of fixed nuclei, some DNA fragments may be extended enough to reach spatially distant regions. The number of such cases (“incorrect links”) is impossible to estimate. In particular, Stevens et al. proposed to solve this problem by discarding the most extended links as noise. When the density of links is relatively low, the polymer can adopt the configuration that accommodates both correct and incorrect links because of the flexibility of the polymer. With the increase of the density of the links, the flexibility of polymer is more and more restricted, and incorrect links interfere with the folding more. This will be manifested in the model by the increase in the number of discarded extended links (springs). Assuming that that percentage of incorrect links is about the same in cells with high and low numbers of captured contacts, one can expect the higher percentage of discarded contacts in the cells with high number of captured contacts.

To conclude, we agree that our modelling approach cannot take into account all the experimentally detected contacts, but it assumes that most of the contacts are correct and removes overstretched links, which cannot equilibrate in the 3D conformation, see Supplementary Fig. 21.

2. I believe that there is still something incorrect about the data presented in Fig 2e. Adding up the total number of windows, I get about 200 for Cell 4 and 50 for Cell 6. With a window width of 100 kb (as stated in the Fig 2e legend), this means about 20 Mb for Cell 4 and 5 Mb for Cell 6. In the reply, the authors stated that there is 25 Mb covered for both cells. This is close to my estimate for Cell 4, but significantly off for Cell 6. So I think that there is something wrong with the numbers for Cell 6.

Reply:

Indeed, in Figure 2e we showed the distributions for a window width of 400 kb, but not of 100 kb (thus, 50 windows correspond to 20 Mb). In the revised version of the MS we indicated this in the figure legend.

3. The last sentence of the Introduction reads “At least 50% of TAD boundaries identified in each individual cell bear active chromatin marks and are highly reproducible between individual cells.” (line 123, p.4). I believe that this was not stated in the Results section: the authors mentioned that 39.5% of all borders were shared between different cells in pairwise comparisons and the stable borders (defined as present in more than 50% of the cells) had high levels of active chromatin marks. But nowhere is it mentioned that “at least 50% of TAD boundaries identified in each cell” bear the active chromatin marks. That is, it is not stated anywhere that more than 50% of the borders in cell 1 (or any cell) have the active marks. And neither is it mentioned that this 50% of borders that bear the active marks are “highly reproducible” between individual cells. Perhaps this apparent discrepancy can be clarified.

Reply:

We thank the reviewer for this concern. Indeed, the above-mentioned sentence is incorrect. In the revised version of the MS the last paragraph of the Introduction is modified in the following way: “These maps allow direct annotation of TADs that appear to be non-hierarchical and are remarkably reproducible between individual cells. TAD boundaries conserved in different cells of the population bear a high level of active chromatin marks supporting the idea that active chromatin might be among determinants of TAD boundaries in *Drosophila*²⁴.”

4. Line 308, p.9 says “We next applied dissipative particle dynamics (DPD) polymer simulations to reconstruct the 3D structures of haploid X chromosomes (Supplementary Fig. 14a)”. But Supp. Fig 14a does not show anything related to the simulations.

Reply:

Indeed, the reference to the figure was wrong. We now correct the typo. Note, that Supplementary Fig. 14 from the previous version of the MS is Supplementary Fig. 16 in the revised version of the MS.

5. The new model depicted in Fig 6c is intriguing but I believe conflicts with the higher resolution data from Wang et al (ref 25), which shows a single, well-defined border separating the large TADs (associated with inactive chromatin) and the proximal small TAD within the active chromatin. This new model would also not explain the presence of well-defined small TADs within the active chromatin in this data, which were also observed by microscopy in Mateo et al, Nature 568, 49 (2019).

Reply:

Indeed, Wang et al. high-resolution Hi-C dataset demonstrates well-defined borders separating large TADs and proximal small TADs within the active chromatin on the sides. The example of this can be found following HiGlass view:

http://higlass.skoltech.ru/l/?d=BfWCvXu_SEW6y5FYK2KNqQ

However, *Drosophila*'s average gene length is around 11.3 Kb, making it impossible for snHi-C with an effective resolution of 10 Kb to reveal small active TADs of chromatin. Small active TADs are beyond the resolution of our approach. Thus, in our model in Figure 6b-c, we do not emphasize the structure of active regions separating large inactive TADs. We note that the absence of any structure at inactive TAD borders denotes ambiguity of folding of these regions with snHi-C, but not the absence of this structure.

We now modify the sizes of inactive/active regions to better correspond to the relative sizes of inactive/active chromatin regions in *Drosophila*. We also add the notion on the ambiguity of active regions into the Figure 6 caption: "*Fine structure of active regions is not observable with snHi-C and thus is denoted in white*". We hope that this modification will resolve the conflict with Wang et al. data and do not introduce the features that we could not observe with the snHi-C approach.

6. Finally, the authors failed to correct the legend to Suppl Fig 15a and b: the definition of "false negatives" is still literally the same as that of "true positives".

Reply:

We indeed did not correct the legend to Supplementary Figure 17a and b (we believe that the Reviewer suggested Supplementary Fig. 17 but not 15). This is now improved: "*True positives for the cell are defined as the number of pairs of bins that interact based on snHi-C and also interact in the models by the criteria above*". Note, that Supplementary Fig. 15 from the previous version of the MS is Supplementary Fig. 17 in the revised version of the MS.

REVIEWERS' COMMENTS

Reviewer #1 (Remarks to the Author):

Authors have addressed all my comments

Reviewer #4 (Remarks to the Author):

The authors have adequately addressed my concerns. Note that the legend to Supplementary Table 3 mentions text that is green, but none of the text in this table is green.

Reviewer #1 (Remarks to the Author):

Authors have addressed all my comments

Reply:

We thank the Reviewer for the positive feedback.

Reviewer #4 (Remarks to the Author):

The authors have adequately addressed my concerns. Note that the legend to Supplementary Table 3 mentions text that is green, but none of the text in this table is green.

Reply:

In the revised version of the MS we have addressed this comment.